# A stromal Integrated Stress Response activates perivascular cancer-associated fibroblasts to drive angiogenesis and tumour progression

Ioannis I. Verginadis[1], Harris Avgousti[1], James Monslow[2], Giorgos Skoufos[3,4], Frank Chinga[1], Kyle Kim[1], Nektaria Maria Leli[1], Ilias V. Karagounis[1], Brett I. Bell[1,5], Anastasia Velalopoulou[1], Carlo Salas Salinas[1], Victoria S. Wu[1], Yang Li[6], Jiangbin Ye[6], David A. Scott[7], Andrei L. Osterman[7], Arjun Sengupta[8,9], Aalim Weljie[8,9], Menggui Huang[1], Duo Zhang[1], Yi Fan[1], Enrico Radaelli[10], John W. Tobias[11], Florian Rambow[12,13], Panagiotis Karras[12,13], Jean-Christophe Marine[12,13], Xiaowei Xu[14], Artemis G. Hatzigeorgiou[3,4,15], Sandra Ryeom[16,17], J. Alan Diehl[18], Serge Y. Fuchs[2], Ellen Puré[2] and Constantinos Koumenis[1] ✉

**Bidirectional signalling between the tumour and stroma shapes tumour aggressiveness and metastasis. ATF4 is a major effector of the Integrated Stress Response, a homeostatic mechanism that couples cell growth and survival to bioenergetic demands. Using conditional knockout ATF4 mice, we show that global, or fibroblast-specific loss of host ATF4, results in deficient vascularization and a pronounced growth delay of syngeneic melanoma and pancreatic tumours. Single-cell transcriptomics of tumours grown in *Atf4*[Δ/Δ] mice uncovered a reduction in activation markers in perivascular cancer-associated fibroblasts (CAFs). *Atf4*[Δ/Δ] fibroblasts displayed significant defects in collagen biosynthesis and deposition and a reduced ability to support angiogenesis. Mechanistically, ATF4 regulates the expression of the *Col1a1* gene and levels of glycine and proline, the major amino acids of collagen. Analyses of human melanoma and pancreatic tumours revealed a strong correlation between ATF4 and collagen levels. Our findings establish stromal ATF4 as a key driver of CAF functionality, malignant progression and metastasis.**

The tumour microenvironment (TME) is a diverse ecosystem comprising multiple malignant cells and untransformed stromal and immune cells. These cells have functions that affect tumour growth and progression to metastasis and shape therapeutic responses[1,2]. Among stromal cells, CAFs constitute a distinct and heterogeneous population and are one of the most active and functionally important components of the TME[3,4]. These CAFs are often co-opted to support multiple hallmarks of cancer[5,6]. CAFs are the primary source of extracellular matrix (ECM) components, including collagens, fibronectin and matrix metalloproteinases[7], which modulate tumour stiffness and facilitate tumour progression[8–10]. CAFs also secrete a plethora of cytokines, chemokines, growth factors and exosomes that further support tumour progression and modulate treatment responses[11–13].

Despite recent advances in prevention and treatment, including immune checkpoint inhibitors, malignant melanoma remains a particularly aggressive and deadly malignancy[14], which is partly attributed to its highly heterogeneous TME. Meanwhile, pancreatic ductal adenocarcinoma has one of the worst outcomes among all malignancies, with a median 5-year survival rate of 7%[15]. In both melanoma and pancreatic ductal adenocarcinoma, this heterogeneous TME coupled with disorganized vasculature, limits the delivery of oxygen and nutrients to malignant cells, which leads to the development of hypoxic and nutritional stress. Malignant cells exhibit altered signalling pathways that enable them to adapt to both cell intrinsic and cell extrinsic stressors within the TME. The Integrated Stress Response (ISR) is an evolutionarily conserved mechanism that promotes cellular adaptation to TME stresses[16–18].

[1]Department of Radiation Oncology, Perelman School of Medicine, University of Pennsylvania, Philadelphia, PA, USA. [2]Department of Biomedical Sciences, School of Veterinary Medicine, University of Pennsylvania, Philadelphia, PA, USA. [3]Department of Electrical and Computer Engineering, University of Thessaly, Volos, Greece. [4]Hellenic Pasteur Institute, Athens, Greece. [5]Department of Radiation Oncology, Albert Einstein College of Medicine, Bronx, NY, USA. [6]Department of Radiation Oncology, Stanford University School of Medicine, Stanford, CA, USA. [7]Cancer Metabolism Core, Sanford–Burnham Prebys Medical Discovery Institute, La Jolla, CA, USA. [8]Department of Systems Pharmacology and Translational Therapeutics, Perelman School of Medicine, University of Pennsylvania, Philadelphia, PA, USA. [9]Institute of Translational Medicine and Therapeutics, Perelman School of Medicine, University of Pennsylvania, Philadelphia, PA, USA. [10]Department of Pathobiology, School of Veterinary Medicine, Comparative Pathology Core, University of Pennsylvania, Philadelphia, PA, USA. [11]Penn Genomic Analysis Core, Department of Genetics, Perelman School of Medicine, University of Pennsylvania, Philadelphia, PA, USA. [12]Laboratory for Molecular Cancer Biology, VIB Center for Cancer Biology, KU Leuven, Leuven, Belgium. [13]Department of Oncology, KU Leuven, Leuven, Belgium. [14]Department of Pathology and Laboratory Medicine, Perelman School of Medicine, University of Pennsylvania, Philadelphia, PA, USA. [15]DIANA-Lab, Department of Computer Science and Biomedical Informatics, University of Thessaly, Lamia, Greece. [16]Department of Cancer Biology, Perelman School of Medicine, University of Pennsylvania, Philadelphia, PA, USA. [17]Department of Surgery, Columbia University Irving Medical Center, New York, NY, USA. [18]Department of Biochemistry, School of Medicine, Case Western Reserve University, Cleveland, OH, USA. ✉e-mail: costas.koumenis@pennmedicine.upenn.edu

ISR kinases, including PKR-like ER kinase (PERK), general control non-derepressible 2 (GCN2), double-stranded RNA-dependent protein kinase (PKR) and haem-regulated eIF2α kinase (HRI), converge on phosphorylation of the α-subunit of the eukaryotic translation initiation factor eIF2 (eIF2α) in response to such stresses. Phosphorylation of eIF2α reduces energy-consuming global translation and promotes efficient translation of stress-responsive genes, including activating transcription factor 4 (ATF4). ATF4 regulates the transcription of genes involved in antioxidant responses, autophagy and amino acid biosynthesis and transport[19,20]. We and others have established a critical tumour cell intrinsic role of ATF4 that culminates in the promotion of primary growth and the establishment of metastases in xenograft, allograft and transgenic models[21,22]. However, the potential roles of the ISR and particularly of ATF4 in host-dependent, tumour-adaptive processes, have not been extensively investigated.

Here, we show that global host ATF4 deletion significantly delays both primary and metastatic tumour growth. Tumours grown in ATF4 knockout (KO) mice exhibit deficiencies in markers of CAF activation, a significant reduction in the expression and biosynthesis of type I collagen and secretion of critical pro-angiogenic cytokines. We also report a significant positive correlation between collagen I and ATF4 levels in human tumours and a negative correlation between levels of collagen I with overall survival in patients with melanoma. These studies uncover a crucial, pro-tumourigenic role of the ISR pathway through CAF-dependent mechanisms and suggest effective modes for therapeutic intervention.

## Results

**Deletion of host ATF4 inhibits tumour growth.** To test the impact of host ATF4 on tumour growth, we generated a tamoxifen-inducible KO mouse model by crossing $Atf4^{fl/fl}$ mice with Rosa26::CreER$^{T2}$ (Fig. 1a and Extended Data Fig. 1a) as previously described[23]. Rosa26::CreER$^{T2}$:$Atf4^{WT/WT}$ and Rosa26::CreER$^{T2}$:$Atf4^{fl/fl}$ mice were treated with tamoxifen (Fig. 1a), which results in almost complete (90–100%) excision of ATF4 as assayed by quantitative PCR with reverse transcription (RT–qPCR) analysis of $Atf4$ mRNA levels in whole liver, lung and spleen homogenates of $Atf4^{Δ/Δ}$ mice (Fig. 1b and Extended Data Fig. 1b). Ablation of ATF4 was well tolerated, causing only a modest and transient decrease in body weight (Fig. 1c). Full necropsy analysis of $Atf4^{Δ/Δ}$ mice at 12 months after ATF4 deletion did not reveal overt pathological aberrations, apart from mild-to-moderate toxicity in the small intestine, spleen and liver (Supplementary Table 1).

Mouse melanoma B16F10 cells were subcutaneously injected into the right flank of $Atf4^{WT/WT}$ and $Atf4^{Δ/Δ}$ mice (Extended Data Fig. 1c). A pronounced delay in tumour growth was observed in $Atf4^{Δ/Δ}$ mice accompanied by a significant increase in their survival compared with $Atf4^{WT/WT}$ littermates (Fig. 1d,e and Extended Data Fig. 1d). These effects were not sex-dependent (Extended Data Fig. 1e). A significant inhibitory phenotype on tumour growth was also observed when host ATF4 was excised following the establishment of palpable B16F10 tumours, which provides further support for the role of ATF4 as a potential therapeutic target (Fig. 1f). To test whether these findings extend to other tumour types, we subcutaneously injected mice with syngeneic MH6419 pancreatic tumour cells, which originated from the $Kras^{LSL-G12D/WT}$;$Trp53^{fl/fl}$;$Pdx1$-Cre (KPC) model of spontaneous pancreatic cancer[24]. Similar to the results from the melanoma growth model, global host ATF4 ablation resulted in a pronounced delay in tumour growth and extension of overall survival (Fig. 1g and Extended Data Fig. 1f,g). As the TME can be highly variable depending on the tumour site, we also orthotopically injected MH6419 pancreatic tumour cells in the tail of the pancreas of $Atf4^{WT/WT}$ and $Atf4^{Δ/Δ}$ mice. Growth of these orthotopic pancreatic tumours was significantly reduced in $Atf4^{Δ/Δ}$ mice compared with $Atf4^{WT/WT}$ littermates 3 weeks after injection

(Fig. 1h,i and Extended Data Fig. 1h–j). In total, these results indicate that host ATF4 contributes substantially to the establishment and growth of syngeneic tumours.

**ATF4 is essential for CAF activation.** To delineate the role of host ATF4 in the sequence of events leading to tumour growth, we performed transcriptional profiling at the single-cell level (single cell RNA sequencing (scRNA-seq)) in small (150 mm³) and large (300 mm³) B16F10 tumours grown in $Atf4^{WT/WT}$ and $Atf4^{Δ/Δ}$ mice (Extended Data Fig. 2a). We acquired single-cell transcriptomes from a total of 7,414 cells from small B16F10 tumours (3,860 cells from $Atf4^{WT/WT}$ mice and 3,554 cells from $Atf4^{Δ/Δ}$ mice) and 28,166 cells from large B16F10 tumours (14,526 cells from $Atf4^{WT/WT}$ mice and 13,640 cells from $Atf4^{Δ/Δ}$ mice) for downstream analysis. Graph-based clustering of cells following uniform manifold approximation and projection (UMAP) identified seven distinct cell types in small tumours (Fig. 2a) and five in large tumours (Extended Data Fig. 2b), with CAFs accounting for 6.12% and 2.58% of the total cells in the small and large tumours, respectively. We then cross-referenced the gene signature of each cluster with known markers of cell populations described in the literature[25,26] (Fig. 2b and Extended Data Fig. 2c) and performed differential gene expression analysis for each cell type to identify potential transcriptome changes between the $Atf4^{WT/WT}$ and $Atf4^{Δ/Δ}$ cohorts. We confirmed host ATF4 deletion by the absence of $Atf4$ mRNA expression in $Atf4^{Δ/Δ}$ mice across all the host clusters, whereas $Atf4$ levels remained unchanged in melanoma clusters compared with the $Atf4^{WT/WT}$ mice (Extended Data Fig. 2d,e). Notably, we observed a substantial decrease in the total number of endothelial cells in the $Atf4^{Δ/Δ}$ grown tumours (Extended Data Fig. 2f,g), which implied that there was deficient angiogenesis. Although there was a decrease in the total number of T cells and natural killer cells (T/NK cells) in $Atf4^{Δ/Δ}$ grown tumours, anti-CD8 treatment caused a small increase in the rate of tumour growth in both $Atf4^{WT/WT}$ and $Atf4^{Δ/Δ}$ mice, which suggests that other mechanisms must account for the dramatic differences seen between these cohorts (Extended Data Fig. 2h). We did, however, observe striking differences in gene expression in a cluster that corresponds to CAFs. In this cluster, we identified 148 differentially expressed genes (Supplementary Table 2), including a significant downregulation of $Col1a1$ and $Col1a2$ in tumours grown in $Atf4^{Δ/Δ}$ mice (Fig. 2c). $Col1a1$ and $Col1a2$ encode the pro-α1(I) and pro-α2(I) chains, respectively, essential components of type I collagen, the most abundant collagen (~90%) in the body and in the ECM[27]. Furthermore, several additional collagen genes were downregulated in tumours grown in $Atf4^{Δ/Δ}$ mice (Supplementary Table 2). In contrast to the small tumours, no difference was observed in the expression levels of $Col1a1$ and $Col1a2$ between the $Atf4^{WT/WT}$ and $Atf4^{Δ/Δ}$ cohorts (Extended Data Fig. 2i), which suggests that there is activation of an alternative mechanism of $Col1$ gene expression to compensate for ATF4 loss during tumour progression. Notably, the expression levels of $Acta2$ (which encodes αSMA) and $Pdgfrb$ (which encodes platelet-derived growth factor receptor-β), which are broadly reported as markers of CAFs[4,6], were nearly absent in the tumours grown in $Atf4^{Δ/Δ}$ mice (Fig. 2c). Similar to the data from small tumours, the expression levels of $Acta2$ in the CAF cluster were significantly reduced in the large tumours grown in $Atf4^{Δ/Δ}$ mice (Extended Data Fig. 2i and Supplementary Table 3). Gene set enrichment analysis using Reactome pathways revealed that genes with CAF-related functions, such as ECM organization, collagen formation and biosynthesis, exhibited higher levels of expression in the CAFs of small tumours grown in $Atf4^{WT/WT}$ mice compared with $Atf4^{Δ/Δ}$ mice (Fig. 2d, left, and Supplementary Table 4). Moreover, CAFs in large tumours grown in $Atf4^{WT/WT}$ mice showed enrichment for mRNAs of the ISR and unfolded protein response (UPR) pathways and smooth muscle contraction (Extended Data Fig. 2j) compared with those in $Atf4^{Δ/Δ}$ mice.

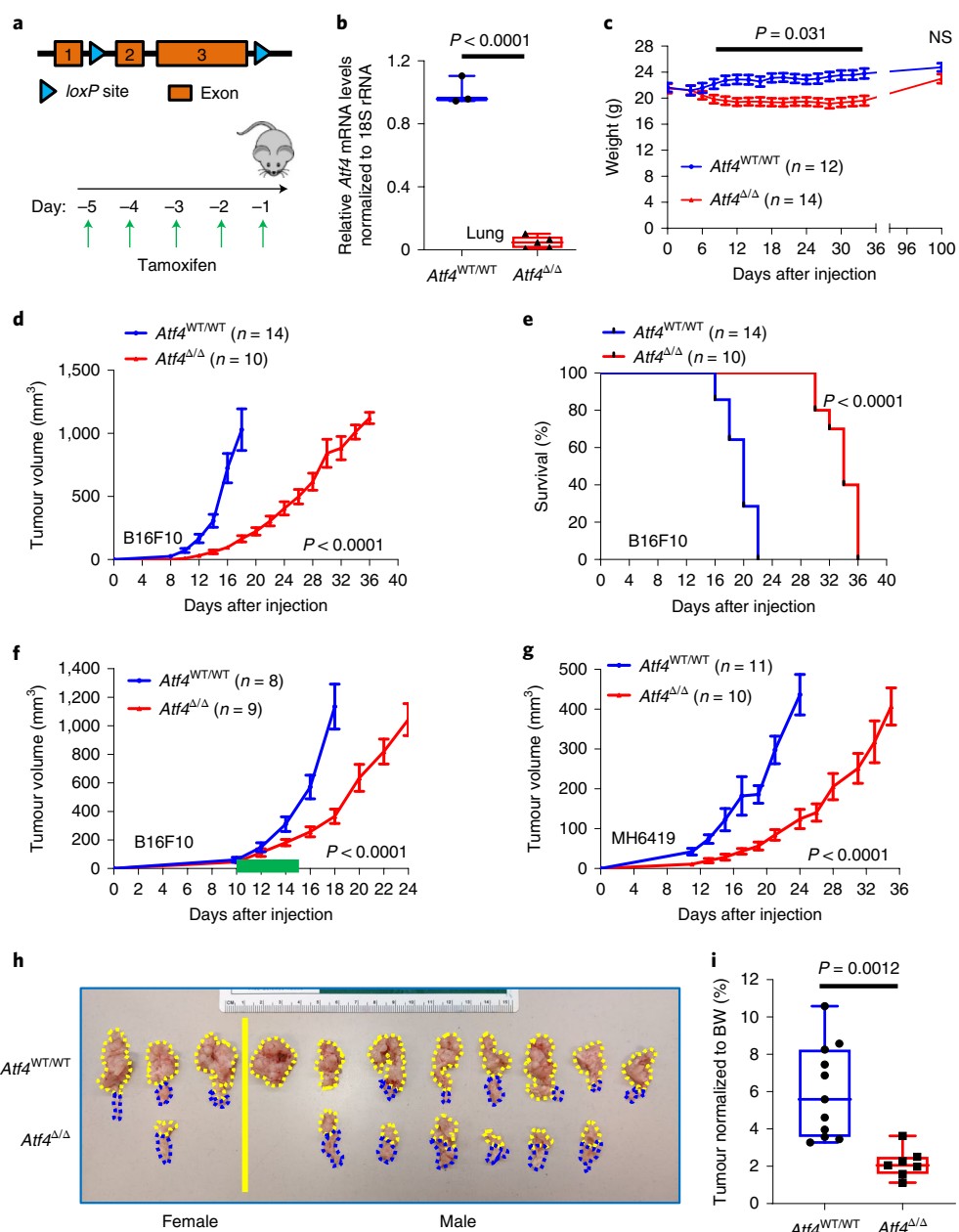

**Fig. 1 | Host ATF4 deletion inhibits tumour growth and extends survival. a**, Top: *loxP* sites flank exons 2 and 3 of the *Atf4* gene. Bottom: schematic of the tamoxifen treatment schedule. Tamoxifen (200 mg per kg body weight (BW) was given for 5 consecutive days by oral gavage. **b**, Box and whisker plot of the RT–qPCR results of *Atf4* mRNA levels in whole lung (*n* = 3–5 biologically independent samples per group). Unpaired two-sample *t*-test. **c**, Reversible BW loss after ATF4 excision. Two-way analysis of variance (ANOVA) analysis (until day 34). Values represent the mean ± s.e.m., unpaired *t*-test. NS, not significant. **d**, Tumour growth curves of *Atf4*^WT/WT^ and *Atf4*^Δ/Δ^ mice following the injection of 5 × 10⁵ B16F10 cells. Values represent the mean ± s.e.m., two-way ANOVA analysis (until day 18). **e**, Kaplan–Meier survival analysis of the mice from **d** (log-rank (Mantel–Cox) test). **f**, Growth curves after injection of B16F10 cells into *Atf4*^WT/WT^ and *Atf4*^Δ/Δ^ mice with the tamoxifen administered after the tumours reached around 100 mm³ and continued for 5 days (dark green line on *x* axis). Values represent the mean ± s.e.m., two-way ANOVA analysis (until day 18). **g**, Tumour growth curves of *Atf4*^WT/WT^ and *Atf4*^Δ/Δ^ mice following injection of 5 × 10⁵ MH6419 cells. Values represent the mean ± s.e.m., two-way ANOVA analysis (until day 24). **h**, Images from pancreas collected 3 weeks after injection of 5 × 10⁴ MH6419 cells orthotopically in the tail of the pancreas of *Atf4*^WT/WT^ (*n* = 11 biologically independent samples) and *Atf4*^Δ/Δ^ (*n* = 7 biologically independent samples). The yellow and blue dotted lines indicate the tumour and normal areas of pancreas, respectively. **i**, Box and whisker plot display the percentage tumour normalized to BW. Unpaired two-sample *t*-test. *n* in figures represent biologically independent samples.

By contrast, genes expressed at higher levels in *Atf4*^Δ/Δ^ CAFs were associated mainly with ISR activation, including response of GCN2 to amino acid deficiency (Fig. 2d, right, and Supplementary Table 5), a finding consistent with the effects of ATF4 loss in these cells. Using previously described gene signatures[28] (Supplementary Table 6),

we identified three distinct cell subclusters in small tumours: vascular CAFs (vCAFs), matrix CAFs (mCAFs) and cyclin/melanoma CAFs (cCAFs/melCAFs) (Fig. 2e and Extended Data Fig. 2k). By contrast, in large tumours, we identified four distinct cell subclusters, with melCAFs separated from the cCAFs subcluster and

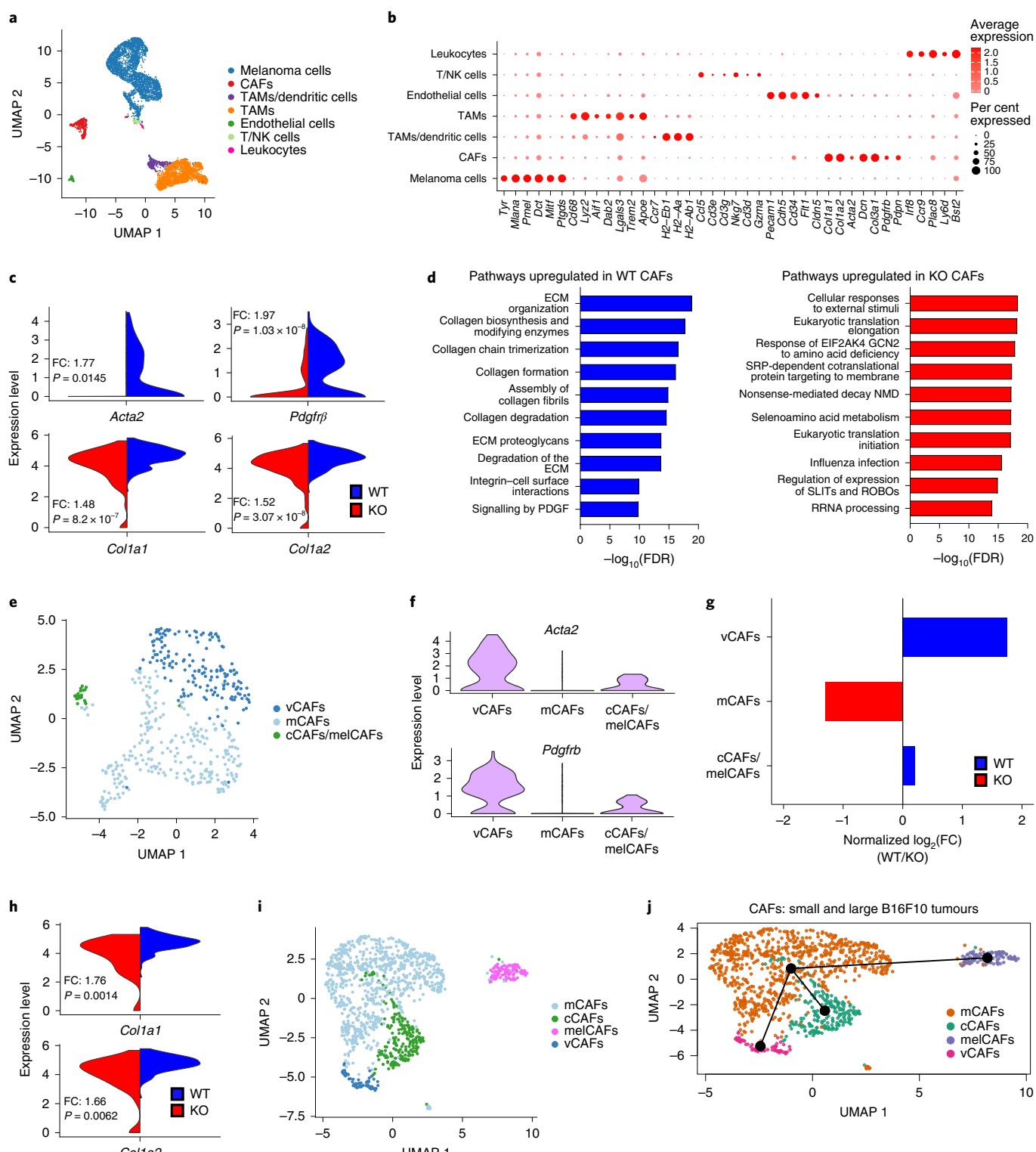

**Fig. 2 | Single-cell transcriptomic analysis reveals ATF4-dependent changes in CAFs in B16F10 tumours. a**, UMAP plot of cells from two biologically independent samples pooled from small B16F10 tumours (150 mm³) from each genotype. Different cell type clusters are colour coded. TAMs, tumour-associated macrophages. **b**, Dot plot displaying selected gene markers across all clusters. The colour intensity represents the average expression and the size of dots indicates the percentage of cells expressing each gene. **c**, Violin plots showing the expression of *Acta2*, *Pdgfrb*, *Col1a1* and *Col1a2* at the CAF cluster identified in B16F10 tumours. The *y* axis shows the mean expression level. Red (KO) and blue (WT) represent $Atf4^{\Delta/\Delta}$ and $Atf4^{WT/WT}$, respectively. FC, fold change. **d**, Bar plot displaying the negative $\log_{10}$(false discovery rate (FDR)) of the ten most significantly upregulated gene ontology terms enriched in WT (left) or KO (right) CAFs. **e**, UMAP plot after reclustering of the CAF cell type in the dataset from **a**. **f**, Violin plots of the expression levels of the indicated CAF markers in the CAF subclusters. **g**, Bar plot of the normalized $\log_2$(FC) (WT/KO) of CAF subclusters in tumours grown in each genotype. **h**, Violin plots of *Col1a1* and *Col1a2* expression in the vCAF subcluster. **i**, UMAP plot of reclustered CAFs from merged small and large B16F10 tumours. **j**, Slingshot-based pseudo-time ordering suggests that mCAFs move along a differentiation trajectory to become vCAFs, cCAFs and melCAFs.

forming a distinct subcluster (Extended Data Fig. 3a–c). The vCAFs are considered essential for vascular development and angiogenesis and were characterized by the highest levels of *Acta2* and *Pdgfrb* expression among all CAF subclusters (Fig. 2f and Extended Data Fig. 3d). Notably, the vCAFs were substantially reduced in *Atf4*[Δ/Δ] mice (Fig. 2g and Extended Data Fig. 3e), and *Col1a1* and *Col1a2* were significantly downregulated only in the vCAF subcluster in the small tumours in *Atf4*[Δ/Δ] mice (Fig. 2h).

Interestingly, vCAFs remained a distinct subcluster during the transition from small to large sized melanoma tumours, thereby underlying the importance of this subcluster in shaping the TME (Fig. 2i,j). Identifying the origin of CAFs poses a challenge owing to their heterogeneity and remarkable plasticity[6,29]. To that end, we compiled a list of specific gene markers for vCAFs[28] and mural cells[30] (that is, vascular smooth muscle cells (vSMCs) and pericytes) (Supplementary Table 7) and analysed the signal strength of both gene signatures. The signal strength of the vCAF gene signature was much stronger in small melanoma tumours than the mural cell signature (Extended Data Fig. 3f), whereas a smaller difference was observed in large tumours (Extended Data Fig. 3f). This suggests that during the transition from small to large sized melanoma tumours, the vCAF subcluster transitions from a fibroblast-origin phenotype to a more mural-like one.

Collectively, these results suggest that host ATF4 deletion impairs the functionality of CAFs at different stages of tumour development that results in a tumour-inhibiting phenotype.

**Host ATF4 loss results in abnormal tumour vascularization.** To further explore the tumour growth differences between the two genotypes, we stained large size B16F10 and MH6419 tumours of equal volume (approximately 300 mm³) from *Atf4*[WT/WT] and *Atf4*[Δ/Δ] mice for CD31 expression, a pan-endothelial marker. Microvessel length and microvascular density were significantly reduced in the *Atf4*[Δ/Δ] mice, which indicates that there was abnormal vascularization in both tumour types (Fig. 3a,b and Extended Data Fig. 4a–c). Similar results were obtained in B16F10 tumours (about 1,000 mm³), which showed a persistent vascularization defect after ATF4 loss (Extended Data Fig. 4d,e). To elucidate this defect on vascularization, we stained B16F10 tumours from *Atf4*[WT/WT] and *Atf4*[Δ/Δ] mice with Ki-67 (a marker of proliferation) and CD31, and TUNEL (a marker of apoptosis) and CD31. Endothelial cells in *Atf4*[Δ/Δ] mice displayed markedly lower proliferation rates compared with those in *Atf4*[WT/WT] mice (Extended Data Fig. 4f,g), whereas no significant change in apoptosis was observed between the two genotypes (Extended Data Fig. 4h,i). Perfusion studies revealed reduced vascular permeability in *Atf4*[Δ/Δ] mice, which provides further support for an important role of ATF4 in tumour vascularization (Extended

Data Fig. 4j,k). Abnormal blood vessels lead to reduced levels of nutrients and oxygen to tumour tissue, which results in intratumoural necrotic areas. Indeed, B16F10 tumours from *Atf4*[Δ/Δ] mice presented a higher percentage of necrotic areas compared with *Atf4*[WT/WT] tumours (Extended Data Fig. 4l–n). Additionally, B16F10 tumours grown in *Atf4*[Δ/Δ] mice exhibited substantially higher levels of apoptosis and a lower fraction of proliferative tumour cells (Extended Data Fig. 4o–r), which is consistent with the markedly lower rates of tumour growth observed in *Atf4*[Δ/Δ] mice. Staining of B16F10 and MH6419 (both subcutaneous and orthotopic models) tumours for αSMA revealed that αSMA expression was primarily restricted in the perivascular area and showed dramatic reductions in tumours grown in *Atf4*[Δ/Δ] mice compared with levels observed in *Atf4*[WT/WT] mice (Fig. 3c,d and Extended Data Fig. 5a–f). Levels of additional CAF markers such as fibroblast activation protein (FAP) in MH6419 tumours and PDGFRβ in B16F10 tumours were also significantly reduced in *Atf4*[Δ/Δ] mice (Fig. 3e,f and Extended Data Fig. 5g–i). These findings further corroborate the results from the scRNA-seq analysis, in which the expression levels of *Acta2* and *Pdgfrb* in the CAF cluster were significantly reduced in the tumours grown in *Atf4*[Δ/Δ] mice. Moreover, CAFs in B16F10 tumours grown in *Atf4*[Δ/Δ] mice presented lower proliferation rates compared with the higher proliferation rates observed in *Atf4*[WT/WT] littermates (Extended Data Fig. 5j,k). Additionally, no significant change in apoptosis levels was observed between the two genotypes (Extended Data Fig. 5l,m). The expression of markers of other cell types (that is, mural cells)[30], which also contribute to blood vessel functionality such as neural/glial antigen 2 (NG2; pericytes), was not appreciably altered in the melanoma tumours from *Atf4*[Δ/Δ] mice (Extended Data Fig. 5n,o). A primary function of fibroblasts is the synthesis and maintenance of ECM. Interestingly, collagen levels were significantly reduced in tumours grown in *Atf4*[Δ/Δ] mice (Fig. 3g,h and Extended Data Fig. 5p,q), and this reduction was primarily confined to the perivascular area (Fig. 3i,j). Finally, immunofluorescence staining of human melanoma tissues revealed that in addition to tumour cells, ATF4 is highly expressed in CAFs (αSMA) that localized to the perivascular area (CD34) (Fig. 3k).

We then sought to investigate whether the absence of activated fibroblasts is related to the tumour-inhibitory effects observed in *Atf4*[Δ/Δ] mice. As the syngeneic B16F10 tumours are grown subcutaneously, we isolated dermal fibroblasts from tumour-naive *Atf4*[WT/WT] (DFB[WT/WT]) and *Atf4*[Δ/Δ] (DFB[Δ/Δ]) mice (Extended Data Fig. 6a). These were co-injected with B16F10 cells (3:1 ratio) into the flanks of *Atf4*[WT/WT] or *Atf4*[Δ/Δ] mice (Fig. 3l). DFB[WT/WT] injected into *Atf4*[Δ/Δ] mice markedly reversed the tumour growth inhibition observed in the *Atf4*[Δ/Δ] + DFB[Δ/Δ] group, whereas DFB[Δ/Δ] injected into *Atf4*[WT/WT] mice caused a delay in tumour growth compared with the

**Fig. 3 | ATF4 loss is associated with abnormal tumour vascularization and reduced ECM component deposition that culminates in a tumour-inhibiting phenotype. a**, Representative immunofluorescence (IF) images from B16F10 tumours (~300 mm³) stained for CD31 (green). Original magnification, ×10 or ×28 (insets). **b**, Box and whisker plot of the microvessel length from **a** (*n* = 4 biologically independent samples per group). **c**, Representative IF images from B16F10 tumours stained for αSMA (red) and CD31 (green). Magnification, ×20. **d**, Box and whisker plot of the percentage αSMA-positive area from **c** (*n* = 5–6 biologically independent samples per group). **e**, Representative IF images from MH6419 tumours stained for FAP (green). Magnification, ×10. **f**, Box and whisker plot of the percentage FAP-positive area from **e** (*n* = 4 biologically independent samples per group). **g**, Representative IF images from B16F10 tumour sections stained for collagen (green). **h**, Box and whisker plot of the percentage positive collagen area from **g** (*n* = 7 biologically independent samples per group). **i**, Representative IF image from MH6419 tumours stained for CD31 (red) and collagen (green). Magnification, ×10. **j**, Box and whisker plot of the percentage collagen positive area from **i** (*n* = 4–5 biologically independent samples per group). **k**, Representative IF image from human melanoma tissues stained for CD34 (red), αSMA (green) and ATF4 (white). Arrows denote the αSMA-positive cells with high ATF4 expression located in the perivascular area. Asterisks denote high ATF4 expression in tumour cells. Right: cropped image from ×20 original magnification. **l**, Schematic of the co-engraftment strategy to examine the ATF4-dependent tumour-promoting role of fibroblasts in the TME. **m**, Tumour growth curves of mice of the indicated genotype (*Atf4*[WT/WT] or *Atf4*[Δ/Δ]) co-engrafted with DFB from the indicated ATF4 genotypes in ratios as described in **l** (*n* = 5–6 biologically independent samples per group). Values represent the mean ± s.e.m., two-way ANOVA (until day 17 for *Atf4*[WT/WT] groups and day 24 for *Atf4*[Δ/Δ] groups). **n**, Tumour growth curves of mice with fibroblast/osteoblast-specific ATF4 excision (*Col1a1*Cre;*Atf4*[wt/wt] (*n* = 11 biologically independent samples) and *Col1a1*Cre;*Atf4*[Δ/Δ] (*n* = 7 biologically independent samples)) following subcutaneous injection of 5 × 10⁵ B16F10 cells. Values represent the mean ± s.e.m., two-way ANOVA (until day 14). Unpaired two-sample *t*-test in all box and whisker plots. Scale bars, 50 μm (**k**), 100 μm (**a**, **c**, **e** and **i**) and 1 mm (**g**).

*Atf4*[WT/WT] + DFB[WT/WT] group (Fig. 3m). This delay was not due to loss of *Atf4*-deficient DFBs after injection, as tracking fluorescently labelled DFB[WT/WT] and DFB[Δ/Δ] (Extended Data Fig. 6b) signal intensity from day 0 (day of injection) to day 9 (when it is generally accepted that neovascularization is established) (Extended Data Fig. 6c) revealed that both cell lines displayed similar rates of decline

over time (Extended Data Fig. 6d). These results implicate the fibroblast compartment in the tumour growth deficiency phenotype of the *Atf4*[Δ/Δ] mice; however, we could not exclude the contribution of other cellular compartments. To test this, we excised ATF4 in a tissue-specific manner by crossing *Atf4*[fl/fl] with *Col1a1*::CreER[T2] mice (Extended Data Fig. 6e). The activity of the *Col1a1* promoter

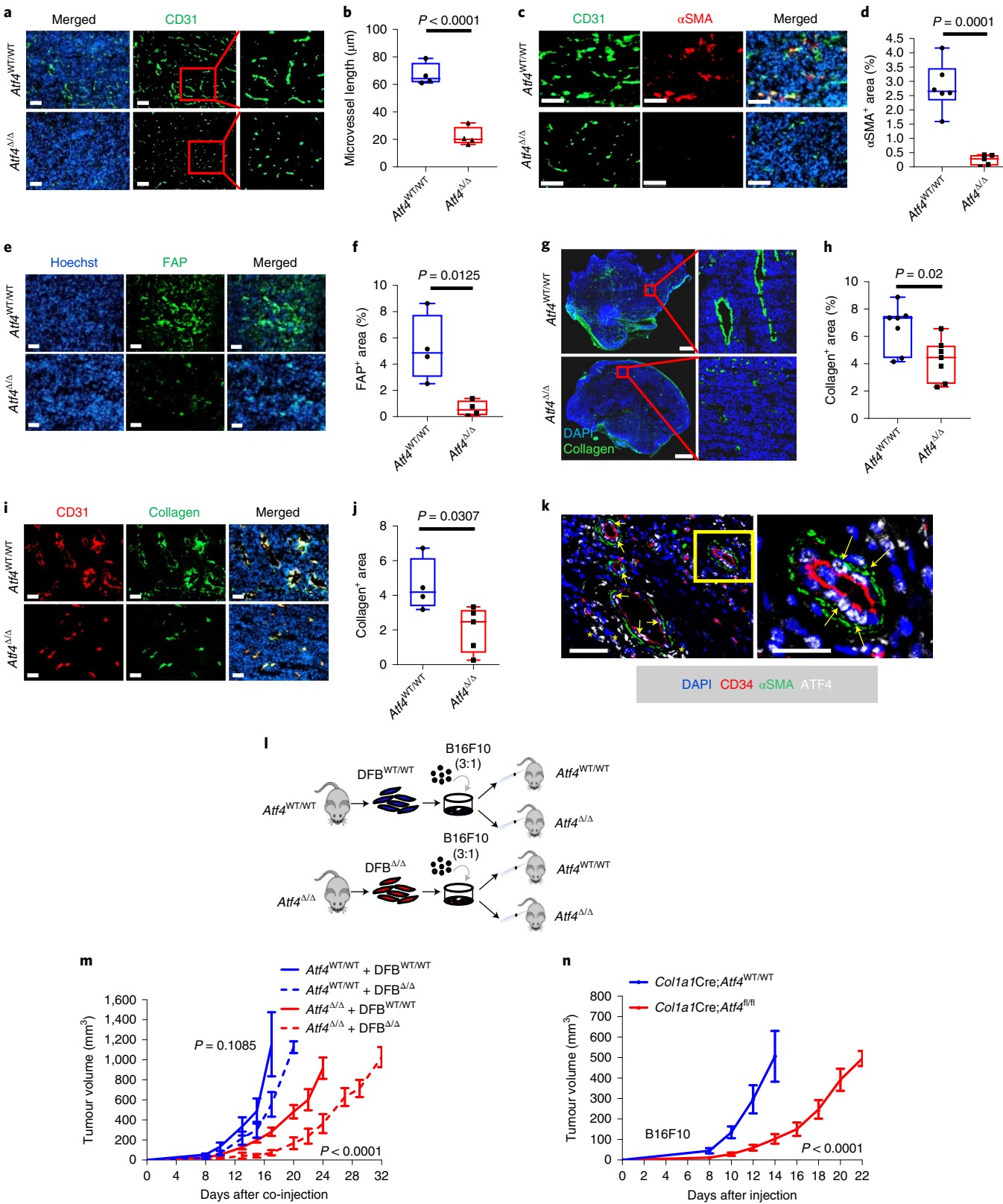

is restricted to the fibroblast and osteoblast cell compartment[31,32]. Deletion of ATF4 in these tissues following tamoxifen treatment did not cause any weight loss or any other overt phenotypes (Extended Data Fig. 6f). *Col1a1*-driven-specific ATF4 deletion caused a significant growth delay in B16F10 tumours, similar to that observed in *Atf4*$^{\Delta/\Delta}$ mice (Fig. 3n). This phenotype was accompanied by reduced microvascular density (Extended Data Fig. 6g,h). Notably, we did not observe any ectopic activity by expression of *Col1a1*;Cre recombinase in tumour endothelial cells (Extended Data Fig. 6i). Thus, these results lend support to the notion that ATF4 deficiency in fibroblasts generates an inhibitory TME through abnormal angiogenesis and reduced collagen deposition.

**ATF4 regulates the collagen biosynthesis pathway.** The single-cell transcriptomic analysis revealed an impact of ATF4 on fibroblast activation status and collagen mRNA levels. To cross-validate some of the scRNA-seq findings, we performed genome-wide microarray analysis on isolated lung fibroblasts from *Atf4*$^{WT/WT}$ (LFB$^{WT/WT}$) and *Atf4*$^{\Delta/\Delta}$ (LFB$^{\Delta/\Delta}$) mice. We identified more than 3,000 genes that were differentially expressed, with a substantial reduction in expression of collagen-associated (that is, *Col1a1* and *Co1a2*, among others) and fibroblast activation (that is, *Pdgfrb*) genes in LFB$^{\Delta/\Delta}$ (Fig. 4a), results that were confirmed by RT–qPCR analysis (Extended Data Fig. 7a). Similarly, *Col1a1* levels were significantly reduced in DFB$^{\Delta/\Delta}$ (Extended Data Fig. 7b). ECM organization and degradation and collagen biosynthesis pathways were the most impaired in LFB$^{\Delta/\Delta}$, as validated using gene set enrichment analysis on the 100 most downregulated genes in *Atf4*$^{\Delta/\Delta}$ mice (Extended Data Fig. 7c and Supplementary Table 8). The biosynthesis of collagen is a highly coordinated process, involving mRNA synthesis and translation into pro-collagen, hydroxylation, glycosylation and crosslink formation (Extended Data Fig. 7d). Because both in vitro and in vivo RNA-seq analysis showed downregulation of *Col1a1* expression in the absence of ATF4, we reasoned that ATF4 directly regulates its expression. Analysis of mouse chromatin immunoprecipitation with sequencing (ChIP-seq) data[33] revealed potential binding sites of ATF4 inside intron 5 of *Col1a1* as alternative transcription start sites (Fig. 4b), and ChIP with RT–qPCR validated the ChIP-seq results (Fig. 4c). We hypothesized that the severe phenotype of reduced collagen levels in tumours grown in ATF4-deficient mice could also involve additional steps in the pathway. The biosynthesis of COL1 protein requires adequate levels of glycine, proline and/or hydroxyproline, which account for 70–100% of its polypeptide chain. To delineate the impact of ATF4 on glycine and proline biosynthesis pathways (Extended Data Fig. 7e,f), we carried out RT–qPCR on LFB$^{WT/WT}$ and LFB$^{\Delta/\Delta}$ (refs. [34,35]). We found substantially reduced levels of enzymes involved in glycine (*Psat1*, *Shmt1* and *Shmt2*)[34] and proline (*Aldh18a1* and *Pycr1*)[35] biosynthesis in LFB$^{\Delta/\Delta}$ compared with LFB$^{WT/WT}$ (Fig. 4d). Using NMR

spectroscopy, we also observed that intracellular levels of both amino acids were significantly reduced in ATF4-deficient cells (Fig. 4e). To further corroborate these findings, we measured the metabolic flux from serine to glycine and glutamine to proline by labelling the cells with serine-$^{13}$C$_3$ (M + 3) and glutamine-$^{13}$C$_5$$^{15}$N$_2$ (M + 7), respectively. Notably, quantitative liquid chromatography and electrospray ionization tandem mass spectrometry (LC–ESI-MS/MS) analysis revealed that although the labelling fractions of M + 3 serine and M + 7 glutamine were similar in LFB$^{WT/WT}$ and LFB$^{\Delta/\Delta}$ cells (Extended Data Fig. 7g,h), there was a significant reduction in both labelled glycine (M + 2/$^{13}$C$_2$) and proline (M + 5/$^{13}$C$_5$ and M + 6/$^{13}$C$_5$$^{15}$N) in LFB$^{\Delta/\Delta}$, which indicates that this reduction was not due to a downregulation in precursor uptake but due to a reduction in ATF4-dependent metabolic flux from serine to glycine and glutamine to proline (Fig. 4f). As expected, this metabolic defect in ATF4-deficient cell lines was accompanied by nearly undetectable intracellular pro-collagen levels (Fig. 4g and Extended Data Fig. 7i). The deficiencies in the synthesis of both mRNA and protein from low mRNA levels translated into a near-complete inability of ATF4-deficient fibroblasts to deposit collagen on gelatin-coated plates (Fig. 4h,i and Extended Data Fig. 7j,k). Notably, re-expression of a mouse ATF4 homologue in LFB$^{\Delta/\Delta}$ resulted in the detection of intracellular pro-collagen levels similar to the levels found in LFB$^{WT/WT}$ (Fig. 4j). Moreover, the high demands in collagen production after TGF-β1 stimulation (secreted in the TME) cause endoplasmic reticulum stress followed by the phosphorylation and activation of PERK, which is absent after siRNA-mediated silencing of *Col1a1* expression (si*Col1a1*) (Fig. 4k and Extended Data Fig. 7l). However, knockdown of *Col1a1* increased the levels of eIF2α phosphorylation followed by an increase in ATF4 expression. This can be explained by a compensatory activation of GCN2 kinase following the reduction in PERK activation. Collectively, these results demonstrate that ATF4 is required to maintain a functional phenotype in fibroblasts through the regulation of multiple steps of the collagen biosynthesis pathway.

**ATF4-deficient fibroblasts lack pro-angiogenic activity.** To further investigate the abnormal vascularization phenotype observed in B16F10 and MH6419 tumours, we performed ex vivo tumour vasculature imaging, using confocal/multiphoton microscopy on excised B16F10 tumours from *Atf4*$^{WT/WT}$ and *Atf4*$^{\Delta/\Delta}$ mice. Fewer blood vessels were found, with less sprouting in the B16F10 tumours grown in *Atf4*$^{\Delta/\Delta}$ mice compared with the tumours grown in *Atf4*$^{WT/WT}$ littermates (Fig. 5a,b and Supplementary Videos 1–4). To delineate the mechanism of the vascularization defect, primary lung endothelial cells were isolated from healthy *Atf4*$^{WT/WT}$ (EC$^{WT/WT}$) mice (Extended Data Fig. 8a,b) and tested for their ability to form endothelial tubes on Matrigel-coated plates. Endothelial cells were stimulated with conditioned medium (CM) derived

**Fig. 4 | ATF4-dependent *Col1a1* expression and multistep regulation of the collagen biosynthesis pathway contribute to fibroblast functionality.**
**a**, Volcano plot from the genome-wide gene expression microarray on LFBs. **b**, Predicted binding site of ATF4 on intron 5 of *Col1a1*. **c**, ATF4 ChIP followed by RT–qPCR at the *Col1a1* locus and *Eif4ebp1* (positive control) (representative from two biologically independent replicates; n = 3–4 technical replicates). NEG, PCR amplification of a site with no predicted ATF4 binding sites, located at intron 6 of *Col1a1*. **d**, Box and whisker plot of RT–qPCR of *Atf4*, *Psat1*, *Shmt1* and *Shmt2* (left) and *Atf4*, *Aldh18a1* and *Pycr1* (right) in LFBs (n = 5–6 biologically independent samples per group). **e**, Box and whisker plot of the NMR spectrometry analysis of intracellular glycine and proline levels (μM per cell) in LFB$^{WT/WT}$ and LFB$^{\Delta/\Delta}$ cells (n = 4 biologically independent samples per group). **f**, LC–ESI-MS/MS analysis to measure the metabolic flux from serine to glycine and glutamine to proline in LFB$^{WT/WT}$ and LFB$^{\Delta/\Delta}$ cells (n = 3 biologically independent samples per group). Values represent the mean ± s.e.m. The letters indicate a significant change from the LFB$^{WT/WT}$ at each isotopologue: $^a$P < 0.01, $^b$P < 0.001. **g**, Proteins were detected by immunoblotting in untreated LFBs. β-actin was used as a loading control. **h**, Representative images of collagen deposition from LFB$^{WT/WT}$ and LFB$^{\Delta/\Delta}$ using second harmonic generation (SHG) microscopy. Magnification, ×10. Scale bar, 100 μm. **i**, Box and whisker plot of the fluorescent signal from **h**. Each dot represents quantitative value from a ×10 field. **j**, Re-expression of a mouse ATF4 homologue from an adenoviral vector (AdmATF4) in LFB$^{\Delta/\Delta}$ cells restores collagen I levels. Proteins were detected by immunoblotting. β-actin was used as a loading control. **k**, LFB$^{WT/WT}$ were treated with TGF-β1 for 6 h and proteins were detected by immunoblotting. β-actin was used as a loading control. siNT, small interfering non-targeting RNA. Numbers below blots represent relative band intensities, normalized to T-eIF2a and β-actin. Unpaired two-sample *t*-test in all box and whisker plots.

from LFB^WT/WT and LFB^Δ/Δ (Extended Data Fig. 8a). In concordance with the ex vivo data, the CM from LFB^Δ/Δ caused a marked reduction in the number of tubes and junctions of EC^WT/WT compared with those treated with CM from LFB^WT/WT, which indicated a possible deficiency in the LFB^Δ/Δ secretome (Fig. 5c,d and Extended Data Fig. 8c). Analysis of CM from ATF4-proficient and ATF4-deficient LFB as well as from LFB^Δ/Δ expressing the mouse ATF4 homologue revealed that the levels of the pro-angiogenic vascular endothelial growth factor (VEGF), stromal-cell-derived factor-1 (SDF-1; also known as CXCL12), insulin-like growth factor

binding protein-2 (IGFBP-2) and IGFBP-9 were all significantly reduced in the CM from LFB^Δ/Δ, and these levels were restored in LFB^Δ/Δ with re-expressed ATF4 (Fig. 5e,f). To further probe these findings in vivo, we ran the same angiogenesis array on tumour lysates from equal volume B16F10 tumours from $Atf4$^WT/WT and $Atf4$^Δ/Δ mice. There were no pronounced differences in the VEGF and CXCL12 secreted levels between the tumours of different ATF4 host status (Fig. 5g). This is probably due to the fact that the fibroblasts are not the only source of the VEGF and CXCL12 secretion into the TME[36–38]. However, staining of B16F10 tumours from

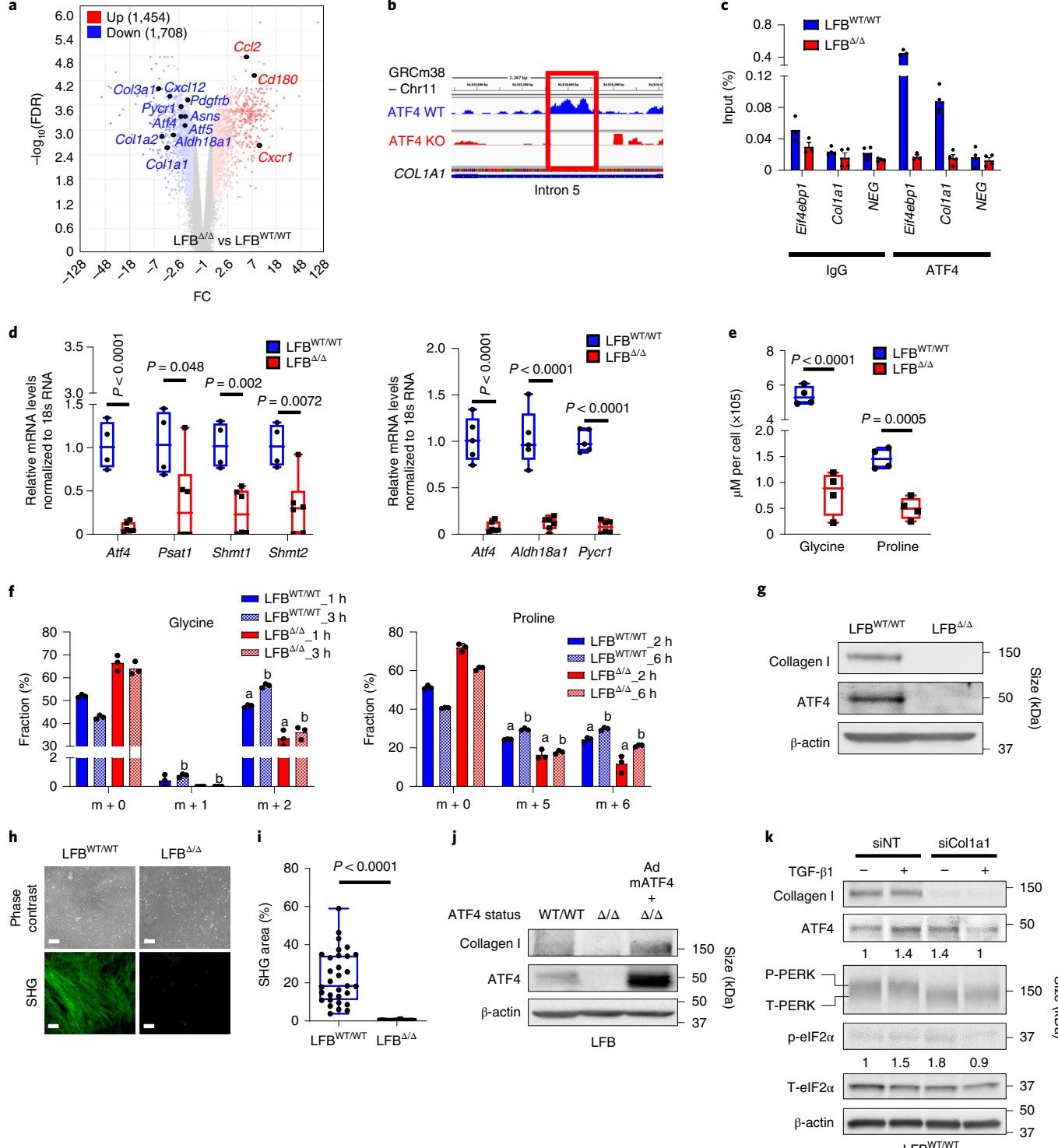

$Atf4^{WT/WT}$ and $Atf4^{\Delta/\Delta}$ mice for VEGF and CD31 and CXCL12 and CD31 showed that although there was no difference in the total signal of areas positive for VEGF and CXCL12 (Extended Data Fig. 8d,e), the levels of both angiogenic factors were lower in the perivascular areas of tumours grown in $Atf4^{\Delta/\Delta}$ mice (Fig. 5h–k). Furthermore, the levels of both VEGF (and CD31) and CXCL1 (and CD31) angiogenic factors were significantly reduced in the perivascular areas of tumours grown in $Col1a1$Cre;$Atf4^{\Delta/\Delta}$ mice compared with $Col1a1$Cre;$Atf4^{WT/WT}$ mice (Extended Data Fig. 8f–i), which indicates that the ATF4-deficient perivascular CAFs present a defective source of secreted angiogenic factors. Some levels of VEGF and CXCL12 are still present in the tumours grown in fibroblast-specific $Atf4$ KOs and therefore we cannot exclude some additional contribution from other sources. It is well established that TGFβ–SMAD3 pathway is active in CAFs, which in turn secrete VEGF, among other cytokines, to boost angiogenesis[39]. Interestingly, expression of both p-SMAD3 and T-SMAD3 were downregulated in LFB$^{\Delta/\Delta}$ after TGF-β1 treatment (Fig. 5l). Taken together, these results suggest that ATF4 loss in fibroblasts impairs their pro-angiogenic activity through a defective secretome, which leads to abnormal angiogenesis and significant attenuation of tumour growth.

**Host ATF4 ablation inhibits lung metastasis.** As activated CAFs also play a crucial role in the establishment of the metastatic niche[29,40,41], we speculated that ATF4 deficiency could also have an inhibitory effect on lung metastasis. B16F10 melanoma tumours, similar to human melanoma, metastasize to multiple sites, but primarily to the lung[42]. We first examined the impact of host ATF4 deletion in the pre-metastatic niche by analysing gene expression changes in lungs from $Atf4^{WT/WT}$ and $Atf4^{\Delta/\Delta}$ mice at 4 weeks after tamoxifen treatment. Genome-wide microarray analysis identified more than 170 genes as differentially expressed, with 21 genes significantly downregulated in $Atf4^{\Delta/\Delta}$ lungs, including $Col1a1$ (Fig. 6a and Extended Data Fig. 8j). Importantly, pathway analysis of the most dysregulated genes revealed defects in collagen formation, ECM organization and integrin cell surface interaction pathways (Fig. 6b and Supplementary Table 9), consistent with the LFB gene expression data. MS analysis of lung tissue extracts revealed pronounced reductions in glycine and proline levels in $Atf4^{\Delta/\Delta}$ mice compared with their $Atf4^{WT/WT}$ littermates (Fig. 6c). These results indicate that loss of host ATF4 might cause an unfavourable metastatic niche, possibly through the regulation of fibroblast functionality. Tail vein injection of B16F10 cells (Fig. 6d) resulted in efficient lung colonization in $Atf4^{WT/WT}$ mice at 3 weeks. By contrast, both the number and the area of lung metastases were significantly reduced in $Atf4^{\Delta/\Delta}$ mice compared with $Atf4^{WT/WT}$ mice (Fig. 6e,f and Extended Data Fig. 8k,l). To elucidate whether this phenotype is due to reduced seeding or growth of B16F10 cells in the lung, we repeated the lung colonization experiment but initiated tamoxifen treatment at 3 days

after tail vein injection of B16F10 cells. Notably, both the number of lung metastases and the percentage of tumour area in lungs were significantly reduced in $Atf4^{\Delta/\Delta}$ mice (Extended Data Fig. 8m–p). This result indicates that ATF4 ablation renders the metastatic niche less permissive for both initial seeding and subsequent growth of melanoma cells. Moreover, in a more physiologically relevant model of metastasis, in which equal volume (approximately 300 mm³) of B16F10 tumours in both genotypes were surgically excised and lungs were examined at 4 weeks after excision (Fig. 6g), the results were even more striking: 6 out of 9 lungs from $Atf4^{\Delta/\Delta}$ mice lacked any detectable metastases, whereas the other 3 presented with only a small single metastatic nodule. By contrast, a significantly higher number of metastases was observed in all the lungs of $Atf4^{WT/WT}$ littermates (Fig. 6h,i). Together, these results indicate that host ATF4 acts as a driving factor in the development of the metastatic niche and efficient metastatic process in B16F10 melanoma tumours.

**Association between ATF4 levels and stromagenesis in patients.** To investigate the relevance of our findings in human malignancies, we analysed the expression of $Col1a1$, $Acta2$ and multiple other genes in relation to ATF4 activity in different cohorts of patients with skin cutaneous melanoma (SKCM) and patients with pancreatic adenocarcinoma (PAAD) from The Cancer Genome Atlas (TCGA) database. As ATF4 is primarily regulated at the translational level[19,23,33], we used ATF4 transcriptional targets (ISR target genes) that we and others have previously reported[19,23] as a surrogate for ATF4 activation (Supplementary Table 10). $COL1A1$, $COL1A2$, $ACTA2$, $PDGFRB$ and $FAP$ displayed a significant positive correlation with this gene dataset in melanoma tumours (SKCM), whereas in pancreatic tumours (PAAD), that correlation was even stronger owing to their highly desmoplastic TME (Fig. 7a and Extended Data Fig. 9a). By contrast, no correlation was found using a list of 32 randomly chosen genes (Extended Data Fig. 9b and Supplementary Table 11). To further probe this relationship, human malignant melanoma and high-density pancreatic cancer tissue arrays were stained for collagen (COL1) and ATF4 by immunohistochemistry. Indeed, a positive correlation was found in the melanoma tissue array (Fig. 7b–d and Extended Data Fig. 9c), which was stronger in the metastatic group compared to the primary tumour group (Fig. 7d, bottom, and Extended Data Fig. 9d), which corroborates the results from the mouse melanoma metastasis model (Fig. 6). A significant positive correlation was also observed in the pancreatic tissue array (Extended Data Fig. 9e–g). We also noted that this correlation was stronger in patients with grade 2 compared with grade 3 pancreatic cancer, which suggests that ATF4 may exert a stronger regulatory role on collagen deposition at earlier disease stages (Extended Data Fig. 9h). Notably, high expression of $COL1A1$ also correlated with poor prognosis in patients with melanoma (Fig. 7e). Together, these findings suggest that ATF4-dependent activation of CAFs dictates

**Fig. 5 | ATF4-deficient fibroblasts fail to support endothelial tube formation and secrete reduced levels of specific angiogenic cytokines.**
**a**, Representative images of vasculature from B16F10 tumours grown in $Atf4^{WT/WT}$ and $Atf4^{\Delta/\Delta}$ mice. **b**, Box and whisker plot of the number of sprouts per field from **a** ($n = 3$ biologically independent samples per group). **c**, EC$^{WT/WT}$ were treated with CM collected from LFB$^{WT/WT}$ or LFB$^{\Delta/\Delta}$ for 24 h and plated for tube formation assay and analysed 4 h after plating. Magnification, ×19. **d**, Box and whisker plots of the number of tubes and number of junctions per field from **c**. **e**, CM collected from LFB$^{WT/WT}$, LFB$^{\Delta/\Delta}$ and LFB$^{\Delta/\Delta}$ + AdmATF4 cells was used for analysis of pro-angiogenic cytokines using antibody arrays. Green boxes indicate the reference spots. Red boxes refer to the analysed proteins (VEGF, CXCL12, IGFBP-2 and IGFBP-9). **f**, Membranes were subjected to immunoblotting and protein levels were quantified from **e**. Values represent the mean ± s.e.m., unpaired two-sample $t$-test. **g**, Tumour lysates from equal volume B16F10 tumours collected from two $Atf4^{WT/WT}$ and two $Atf4^{\Delta/\Delta}$ mice were analysed for pro-angiogenic cytokines using the same antibody array as in **e**. **h**, Representative IF images from B16F10 tumours stained for VEGF (red) and CD31 (green). Magnification, ×20. Right: cropped images from ×20 original magnification. **i**, Box and whisker plot of the percentage VEGF⁺CD31⁺ colocalization area from **h** ($n = 5$ biologically independent samples per group). **j**, Representative IF images from B16F10 tumours stained for CXCL12 (red) and CD31 (green). Magnification, ×20. Right: cropped images from ×20 original magnification. **k**, Box and whisker plot of the percentage CXCL12⁺CD31⁺ colocalization area from **j** ($n = 5$ biologically independent samples per group). **l**, Proteins were detected by immunoblotting in untreated or TGF-β1-treated LFB$^{WT/WT}$ or LFB$^{\Delta/\Delta}$ (6 h). β-tubulin was used as a loading control. Numbers below blots represent relative band intensities, normalized to β-tubulin. Unpaired two-sample $t$-test in all box and whisker plots. Scale bars, 100 μm (**a**), 50 μm (**h** and **j**) and 5 μm (**c**).

early ECM organization and CAF-instructed angiogenesis to support the growth of primary tumours and the metastatic phenotype (Extended Data Fig. 10).

## Discussion

Given the cardinal features of CAFs in the TME[4,29,43], a better understanding of their transitory roles during tumour evolution and the mechanisms underlying these genotypic and phenotypic changes is crucial for developing effective therapeutic approaches. In this study, we uncovered an essential role for the master ISR effector ATF4 in shaping CAF functionality to dictate ECM organization and angiogenesis to support a tumour-promoting phenotype in experimental models of melanoma and pancreatic cancer (Extended Data Fig. 10).

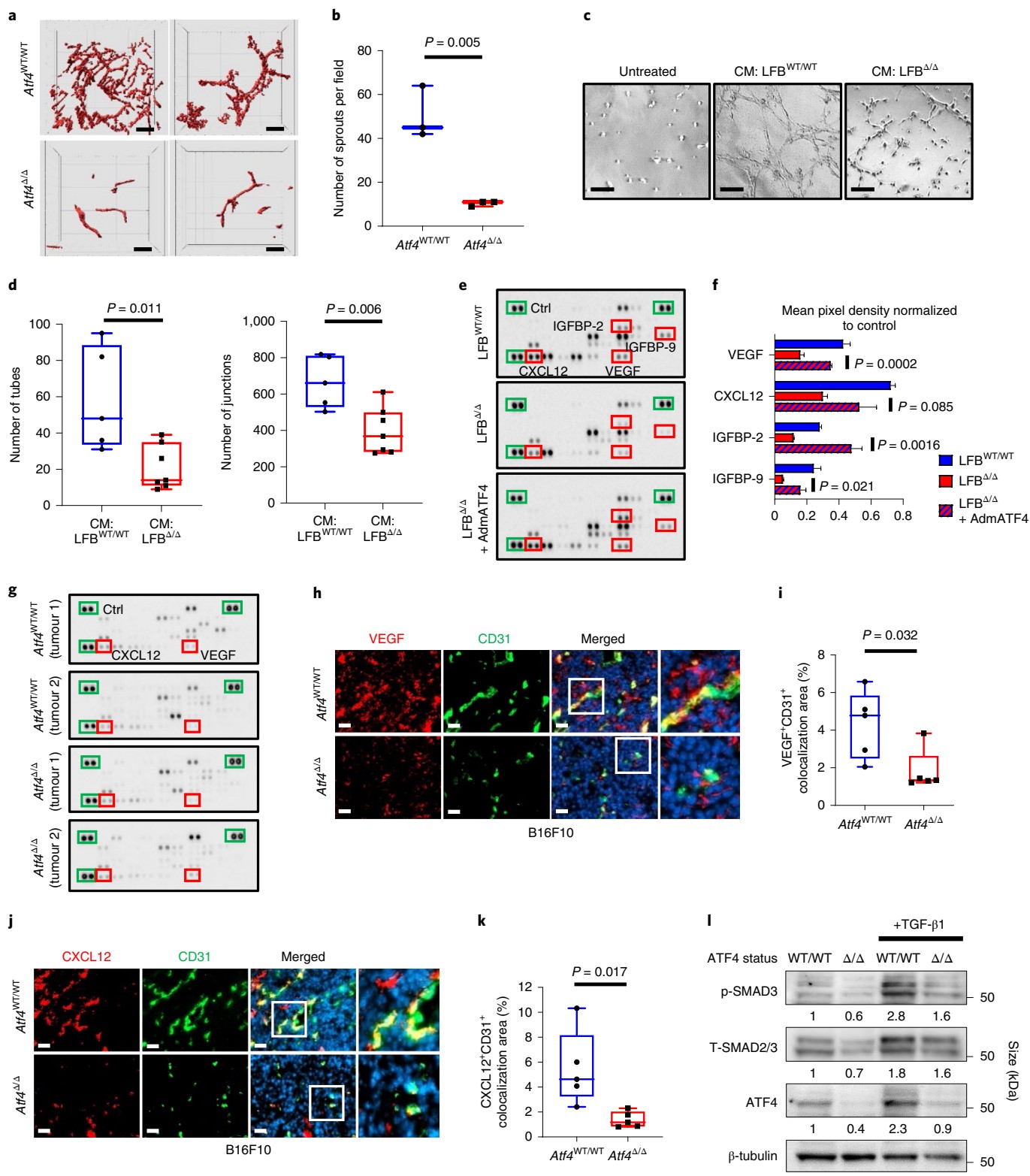

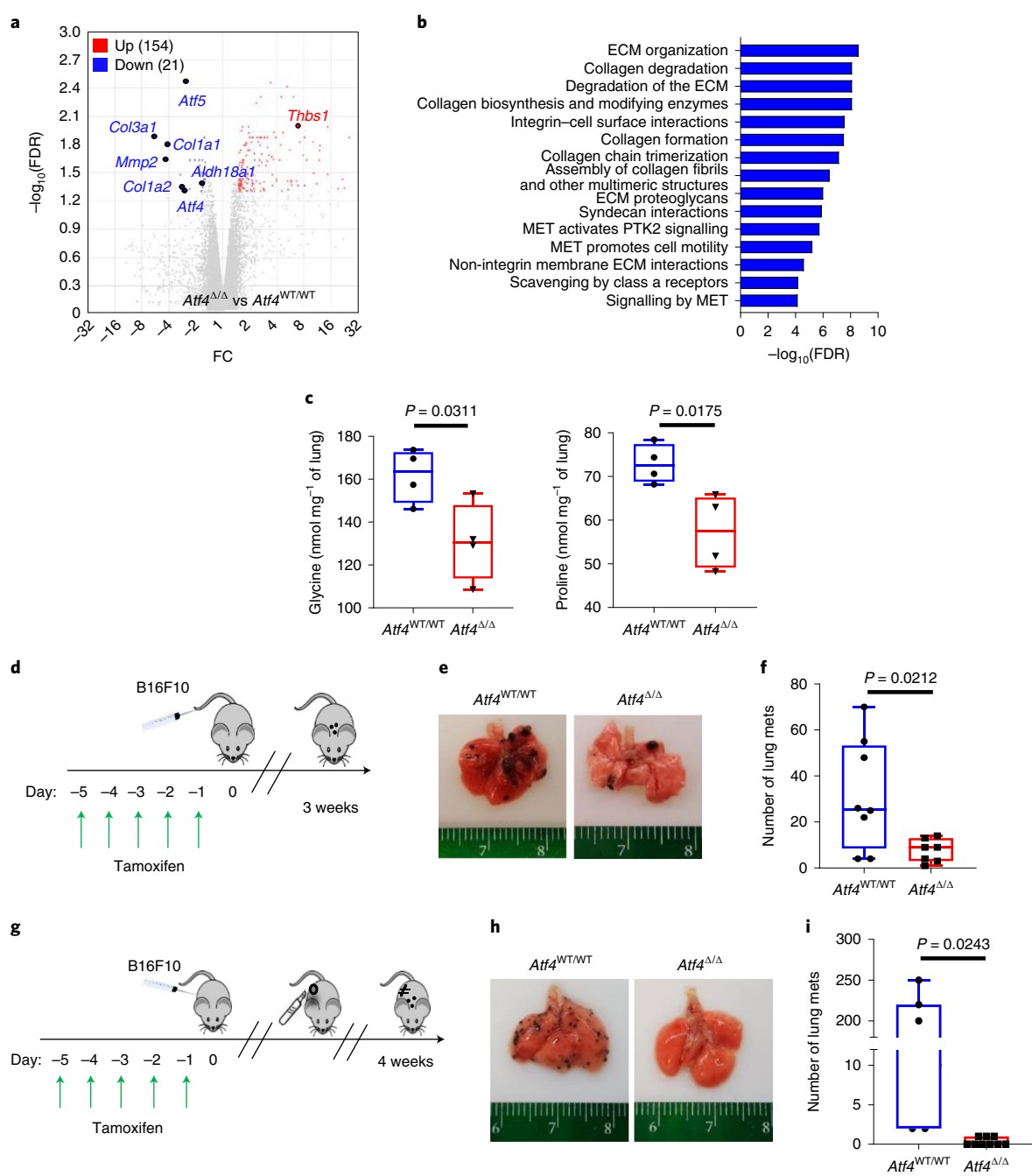

**Fig. 6 | Host ATF4 ablation severely impairs lung colonization and metastasis of melanoma cells. a**, Volcano plot from genome-wide gene expression microarray on lungs from *Atf4*<sup>WT/WT</sup> and *Atf4*<sup>Δ/Δ</sup> mice at 4 weeks after tamoxifen treatment (*Atf4*<sup>Δ/Δ</sup> versus *Atf4*<sup>WT/WT</sup>). **b**, Bar plot displaying the 15 most significantly enriched gene ontology terms in lungs from *Atf4*<sup>WT/WT</sup> compared with *Atf4*<sup>Δ/Δ</sup> mice from **a**. **c**, Box and whisker plot of the quantitative MS analysis in *Atf4*<sup>WT/WT</sup> and *Atf4*<sup>Δ/Δ</sup> lungs (nmol mg$^{-1}$ of lung) for glycine and proline ($n = 4$ biologically independent samples per group). **d**, Schematic of the lung colonization experiment. Mice were injected with $1.5 \times 10^5$ B16F10 cells in the tail vein, and lungs were collected 3 weeks later. **e**, Representative images from *Atf4*<sup>WT/WT</sup> and *Atf4*<sup>Δ/Δ</sup> lungs. **f**, Box and whisker plot of the number of macroscopic lung metastases (mets) ($n = 7$–8 biologically independent samples per group). **g**, Schematic of the process to analyse metastatic activity. Mice were subcutaneously injected with $5 \times 10^5$ B16F10 cells, and the primary tumours were surgically excised when they reached about 300 mm$^3$. The mice were sutured and followed-up for a period of 4 weeks. **h**, Representative images from lungs, collected 4 weeks after tumour excision. **i**, Box and whisker plot of the macroscopic lung metastases ($n = 7$–9 biologically independent samples per group). Unpaired two-sample *t*-test in all box and whisker plots.

Surprisingly, global ATF4 ablation results only in transient decreases in haematocrit and body weight, which completely resolved by 14–16 weeks of age. This phenotype is profoundly milder compared with that following embryonic ATF4 deletion[44], which is characterized by high levels of lethality, microphthalmia,

bone deformities and haematopoietic deficiencies[44–46]. These results suggest that the role of ATF4 in these processes may be more critical during embryonic development. Moreover, tamoxifen-induced deletion of ATF4, albeit efficient, is not complete, ostensibly allowing for repopulation of rapidly replicating cellular compartments

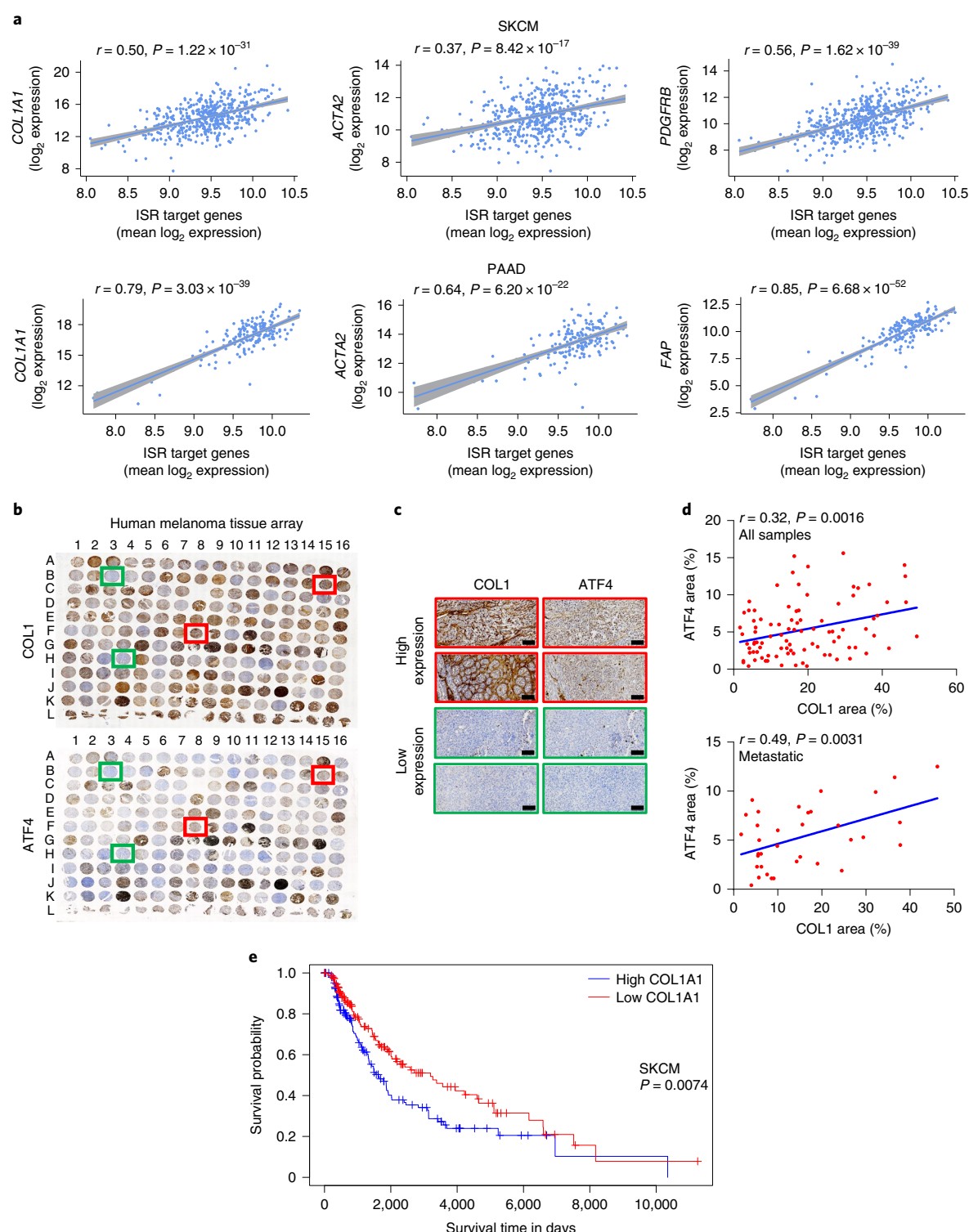

**Fig. 7 | High ATF4 levels or ATF4-dependent gene expression correlate with increased COL1 expression or deposition in human tumours.** The ISR gene signature comprising 32 genes was used as a surrogate for ATF4 activation. **a**, Pearson's correlation between the ISR target signature and *COL1A1*, *ACTA2*, *PDGFRB* and *FAP* in SKCM and PAAD. The linear regression lines along with 95% confidence intervals (shaded regions) are shown. **b**, Human melanoma tissue arrays containing sections from 176 tumours and 16 healthy controls were stained for COL1 (top) and ATF4 (bottom) proteins. Damaged or tissues expressing high melanin levels were excluded from the quantification. Red and green boxes indicate representative high and low expression levels of COL1 and ATF4, respectively. **c**, Representative images from human melanoma tissue arrays stained for COL1 and ATF4 proteins Scale bars, 100 μm. **d**, Pearson's correlation for all the samples (top) and metastatic samples (bottom) between the percentage ATF4 area and percentage COL1 area. **e**, Kaplan–Meier plot of survival time of patients with SKCM with high ($n = 151$ biologically independent samples) or low ($n = 151$ biologically independent samples) COL1A1 expression. log-rank (Mantel–Cox) test.

with ATF4-proficient progenitor cells that escaped *Atf4* excision. Activated fibroblasts with increased collagen synthesis and secretion have been shown to play a pivotal role in wound healing. It is intriguing that in the tumour growth/resection experiments using the global ATF4 KO mice, we did not observe deficiencies with wound closure following surgical resection. It is possible, however, that some defects at the microvascular level may result from ATF4 ablation, a notion that requires further investigation using well-established wound-healing models.

Unbiased scRNA-seq analysis of small and large melanoma tumours showed that CAF activation in *Atf4*$^{Δ/Δ}$ mice was impaired; this was based on the expression levels of *Acta2* and *Pdgfrb*, which are the most commonly used CAF markers[4,47]. The levels of αSMA, PDGFRβ and FAP in melanoma and pancreatic tumours were nearly undetectable in *Atf4*$^{Δ/Δ}$ mice, which indicates that ATF4 is essential for CAF activation within the TME. However, accumulating evidence suggests that CAFs can emerge not only from resident fibroblasts[48] but also from bone-marrow-derived mesenchymal stem cells[49,50], adipocytes[51] and pericytes[52], which is indicative of the remarkable plasticity of this component of the TME. In this regard, more constrained deletion of ATF4 in the fibroblast/osteoblast compartment resulted in a similar tumour growth profile as in the global ATF4 KO mice. Notably, co-injection of fibroblasts from *Atf4*$^{WT/WT}$ mice led to substantial recovery of tumour growth rates in *Atf4*$^{Δ/Δ}$ mice. Although the levels of NG2 expression in melanoma tumours remained relatively unchanged, we cannot exclude the possibility of a contribution from the pericyte compartment on some CAFs.

Our studies also provide a putative mechanism underlying the pro-CAF activation role of ATF4. Collagen biosynthesis is a highly coordinated, multistep process that includes mRNA synthesis and processing, translation into the pro-collagen peptide, hydroxylation and glycosylation[27,53]. A structural prerequisite motif for the assembly of the pro-collagen polypeptide chain is a (glycine-proline-X)$_n$ repeat, which indicates the high demand for these amino acids[27]. It is well established that ATF4 acts as an important transcriptional regulator of genes involved in amino acid biosynthesis and transport[34,54]. In this regard, we found reduced mRNA levels of both *Col1a1* and *Col1a2* genes, as well as in enzymes involved in glycine (*Psat1*, *Shmt1* and *Shmt2*)[34] and proline (*Aldh18a1* and *Pycr1*)[35] biosynthesis in LFB$^{Δ/Δ}$ cells, leading to significantly reduced intracellular levels of both amino acids in ATF4-deficient fibroblasts. Continuous supplementation of non-essential amino acids in the medium was not sufficient to rescue the defect in collagen synthesis in fibroblasts lacking ATF4 (data not shown), as previously been demonstrated in ATF4-deficient osteoblasts[55]. This suggests that transcriptional regulation of *Col1* gene expression may be the dominant defect in ATF4-deficient fibroblasts. Mechanistically, we validated an ATF4 binding site at intron 5 of the *Col1a1* gene. Intriguingly, in humans, this site has been identified as the second most active binding region, with more than 20 transcription factor binding elements, deeming this locus a regulatory hotspot[56,57]. Our data also indicate that TGF-β1 treatment of LFB$^{WT/WT}$ caused high ATF4 expression through PERK activation and eIF2α phosphorylation, which was ameliorated by silencing of *Col1a1* expression. However, the remaining ATF4 expression and p-eIF2α upregulation after si*Col1a1* treatment suggests that another ISR kinase (probably GCN2) may be activated.

There is overwhelming evidence related to the heterogeneity and plasticity of CAFs[6,43,47]. Among our key findings, we identified vCAFs as a spatially distinct CAF subcluster characterized by the highest levels of αSMA and PDGFRβ, which were reduced in *Atf4*$^{Δ/Δ}$ mice. CAFs have been ascribed key roles in supporting angiogenesis through the release of VEGFA, FGF2 and CXCL12 (CAF secretome)[13,58,59] or through the exertion of mechanical forces[60] within the tumour milieu. Interestingly, CM from ATF4-replete LFBs exhibited higher levels of VEGF and CXCL12, which have been shown to drive angiogenesis[13,61]. Our analysis of VEGF and CXCL12 levels in tumours from *Atf4*$^{WT/WT}$ and *Atf4*$^{Δ/Δ}$ mice revealed that the levels of these cytokines were decreased in the perivascular areas of tumours grown in *Atf4*$^{Δ/Δ}$ mice, and vCAFs appear to be the main source of angiogenic factors in the perivascular areas. Finally, we showed impaired TGFβ–SMAD3 signalling in TGFβ-treated LFBs lacking ATF4, thereby suggesting a potential involvement of this pathway in the defective secretome of the LFB$^{Δ/Δ}$.

Collectively, our work highlights the paramount importance of ATF4 in regulating the functionality and activation of CAFs through collagen I synthesis and TGFβ–SMAD3 pathways. Importantly, analysis of patients with cutaneous melanoma and patients with pancreatic adenocarcinoma showed that ATF4-dependent transcriptional signatures correlated with collagen I, CAF markers and overall survival. Overall, the lower toxicity profile following transient ATF4 deletion in mice, coupled with the demonstrated pro-tumourigenic role of ATF4 in a tumour-intrinsic manner, further supports the notion that a clinically useful therapeutic window may exist for ATF4 inhibition as an attractive antitumour modality.

## Online content

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

## Methods

This research complies with all relevant ethical regulations of the University of Pennsylvania, including the Institutional Review Board and Institutional Animal Care and Use Committee (IACUC) committees.

**Plasmids and other reagents.** Lists of reagents, assays and adenovirus used in this study are provided in Supplementary Tables 12 and 13.

**Antibodies.** A list of antibodies is provided in Supplementary Table 14.

**Cell culture.** All cell lines are listed in Supplementary Table 15. B16F10 cells (American Type Culture Collection (ATCC), CRL-6475) and MH6419 cells (provided by B. Stanger, University of Pennsylvania[24]) were cultured in RPMI-1640 supplemented with 10% FBS in the presence of 5% $CO_2$ at 37 °C. Isolated EC$^{WT/WT}$ cells were cultured in EC medium. Isolated LFB$^{WT/WT}$, LFB$^{\Delta/\Delta}$, DFB$^{WT/WT}$ and DFB$^{\Delta/\Delta}$ cells were cultured in phenol-free DMEM/F12. LFB$^{WT/WT}$ and LFB$^{\Delta/\Delta}$ were treated with 2 ng ml$^{-1}$ of TGF-β1, and cell lysates were collected at 6 h. Also, LFB$^{WT/WT}$ cells were transfected with small interfering non-targeting RNA (siNT) or si*Col1a1* for 48 h, then treated with 2 ng ml$^{-1}$ of TGF-β1 and cell lysates were collected at 6 h. All cell lines were determined to be free of mycoplasma, with repeated testing at the Cell Center Facility, University of Pennsylvania.

**LFB and DFB isolation.** Lungs and skin were isolated from 8–10-week old *Atf4*$^{WT/WT}$ and *Atf4*$^{\Delta/\Delta}$ mice (males and females). Tissues from 4 mice were minced and digested in 20 ml of mixed collagenase lysis buffer (1 mg ml$^{-1}$ of collagenase type I (Worthington, LS004214) and 1 mg ml$^{-1}$ collagenase type II (Worthington, LS004176)) dissolved in phenol-free DMEM/F12 without FBS and penicillin–streptomycin under continuous rotation on a rocker at 37 °C for 40–50 min. An equal volume of phenol-free DMEM/F12 supplemented with 10% FBS, 100 U ml$^{-1}$ penicillin and 100 mg ml$^{-1}$ streptomycin were added to the lysed tissues to quench collagenase and then passed through 70-μm and 40-μm cell strainers (Falcon, 352350 and 352340, respectively). Cells were spun at 300*g* for 5 min, and the pellet was resuspended in 10 ml of complete FB medium (phenol-free DMEM/F12 supplemented with 10% FBS, 100 U ml$^{-1}$ penicillin, 100 mg ml$^{-1}$ streptomycin, 1 μm of 4-hydroxytamoxifen (4-HT) (Sigma-Aldrich, H7904), 1× non-essential amino acids (NEAA) (Gibco, 11140-050) and 55 μM β-mercaptoethanol (β-ME) (Millipore, ES-007-E)). This is considered as passage zero (p0). All cell lines were also treated with 50 μg ml$^{-1}$ of gentamycin (VWR, E737) until they reached p2.

**Lung endothelial cell isolation.** Lungs were isolated from 8–10-week old *Atf4*$^{WT/WT}$ mice (males and females). Tissues from 8 mice were minced and digested in 10 ml of collagenase lysis buffer (5 mg ml$^{-1}$ of collagenase type II (Worthington, LS004176)) dissolved in phenol red-free EC medium (ScienCell, 1001-prf) without FBS and penicillin–streptomycin under continuous rotation on a rocker at 37 °C for 35–45 min. An equal volume of phenol-free EC medium supplemented with 10% FBS, 100 U ml$^{-1}$ penicillin and 100 mg ml$^{-1}$ streptomycin were added to the lysed tissues to quench collagenase and then passed through 70-μm and 40-μm cell strainers. Cells were spun at 300*g* for 5 min, the pellet was resuspended in 10 ml of complete EC medium (phenol-free EC medium supplemented with 10% FBS, 100 U ml$^{-1}$ penicillin, 100 mg ml$^{-1}$ streptomycin, 1 μM of 4-HT, 1× NEAA and 55 μM β-ME) and cells were plated in 10-cm plates and incubated at 37 °C for 1 h. Non-adherent cells were collected after 3–5 washes with HBSS, spun at 300*g* for 5 min and washed again once with HBSS supplemented with 0.5% Fraction V BSA (Gibco, 15260-037). Cells were incubated with beads, and CD31+ cells were isolated (positive selection) according to the manufacturer's instructions (Miltenyi Biotec, 130-097-418). The purity was evaluated by flow cytometry (Supplementary Fig. 2). Endothelial cells were plated on 0.1% gelatin-coated plates. This is considered as p0.

**Tumour endothelial cell isolation.** Tumour tissues (equal volume from *Atf4*$^{WT/WT}$ and *Atf4*$^{\Delta/\Delta}$ mice, approximately 300 mm³) were collected, minced and digested in 10 ml of collagenase lysis buffer (as described in the 'Lung endothelial cell isolation' method) under continuous rotation on a rocker at 37 °C for 35–45 min. An equal volume of phenol-free EC medium was added to the lysed tissues to quench collagenase and then passed through a 40-μm cell strainer. Cells were spun at 300*g* for 5 min, the pellet was resuspended in 2 ml of ACK and left at room temperature for 3 min. Next, 10 ml of PBS was added, and cells were spun at 300*g* for 5 min. The pellet was incubated with beads, and CD31+ cells were isolated (positive selection) according to the manufacturer's instructions (Miltenyi Biotec, 130-097-418) and processed for RNA isolation.

**Western blot analysis.** Cells were collected in ice-cold PBS and proteins were extracted using 1× RIPA buffer supplemented with protease (Sigma, B8640) and phosphatase inhibitors (Sigma, P5726 and P0044). Proteins were separated by 4–15% SDS–PAGE, transferred to polyvinylidene fluoride (PVDF) membranes and blocked with 5% nonfat milk or BSA in 1× PBS with Tween-20 (0.1%). Primary antibodies against ATF4 (Cell Signaling Technology, 11815, RRID: AB_2616025), collagen type I (Millipore-Sigma, AB765P, RRID: AB_92259), p-SMAD3 (Cell Signaling Technology, 9520, RRID: AB_2193207), SMAD2 and SMAD3 (Cell Signaling Technology, 8685, RRID: AB_10889933), PERK (Cell Signaling

Technology, 3192, RRID: AB_2095847), β-actin (Cell Signaling Technology, 3700, RRID: AB_2242334), p-eIF2α (Cell Signaling Technology, 3398, RRID: AB_2096481), eIF2α (Cell Signaling Technology, 9722, RRID: AB_2230924) and β-tubulin (Cell Signaling Technology, 2146, RRID: AB_2210545) were added at 1:1,000 and incubated overnight at 4 °C. Membranes were washed, and secondary antibodies (ThermoFisher, 31460, RRID: AB_228341 or 31430, RRID: AB_228307) were added at 1:2,000. ECL (Thermo Scientific, 32106 and GE Healthcare, RPN2232) was added, and membranes were exposed to autoradiography films or ChemiDoc (Bio-Rad).

**Immunofluorescence.** Tumour tissues (equal volume from *Atf4*$^{WT/WT}$ and *Atf4*$^{\Delta/\Delta}$ mice, approximately 300 mm³) were cut in 8–10-μm thick (or 100-μm-thick for confocal microscopy) sections, coded and stored at −80 °C. Slides were thawed at room temperature and subsequently fixed with 2% paraformaldehyde for 20 min. After three washes with TBS, tissues were blocked with 8% BSA and 1% donkey serum in TBS-T (0.025% Triton X-100) at room temperature for 1 h. The primary antibodies against CD31 (1:50, BD Biosciences, 550274 or 1:50, Abcam, ab28364 or 1:20, Novus, NB100-2284), ACTA2 (1:100, Sigma, C6198), FAP (1:100, R&D systems, AF3715), PDGFRβ (1:100, Abcam, ab69506), NG2 (1:20, ThermoFisher, MA5-24247), collagen type I (1:400, Southern Biotech, 1310-01), Ki-67 (1:50, Novus, NB500-170), VEGF (1:50, ThermoFisher, MA5-13182) and CXCL12 (1:50, R&D Systems, MAB-350) were incubated overnight at 4 °C. After three washes with TBS-T, the secondary antibodies (ThermoFisher, A-11005, A-21206, A-11015, A-11006 and A-11055) were added at 1:200 and incubated for 1 h at room temperature in a humidified chamber. After 3 × 5 min rinses with TBS, tissues were stained with 1 μg ml$^{-1}$ Hoechst (Invitrogen, H3570) for 30 min at room temperature, washed with TBS and coverslips were mounted with antifade mounting medium. To analyse the levels of apoptosis in the TME, tumour tissues were stained using an In Situ Cell Death Detection kit (Millipore-Sigma, 11684795910 and 12156792910, TUNEL) according to the manufacturer's instructions.

**Immunohistochemistry.** For haematoxylin and eosin (H&E) staining, 5-μm thick paraffin sections were mounted on Superfrost Plus slides and stained using a Gemini AS Automated Slide Stainer. Slides were finally mounted with a resinous mounting medium (Thermo Scientific ClearVue coverslipper).

**Immunohistochemical staining of human tissue arrays for ATF4 and COL1.** Immunohistochemical staining was used to separately analyse ATF4 and COL1 content in human tissue arrays. Formalin-fixed paraffin-embedded (FFPE) tissue arrays were first deparaffinized and rehydrated following standard procedures. Antigen retrieval was then performed by submerging the slides in simmering (low-rolling boil) 0.05% Tween-20/10 mM sodium citrate (pH 6) for 20 min. After cooling for 5 min at room temperature, sections were consecutively treated to block endogenous peroxidase (3% $H_2O_2$ for 15 min) then with 10% normal serum blocking solution (depending on host of the secondary antibody, in 1% BSA/PBS for 15 min). Sections were then incubated with primary antibodies against ATF4 (1:200, Abcam, ab31390) or COL1 (1:400, Southern Biotech, 1310-01) or IgG isotype (1:200, Jackson ImmunoResearch, 005-000-003 or 1:400, Jackson ImmunoResearch, 011-000-003) control in blocking solution overnight at 4 °C. Sections were then incubated with HRP–IgG secondary antibodies (1:400, Jackson ImmunoResearch, 705-035-147 or 1:400, Jackson ImmunoResearch, 111-035-144) diluted in 1% BSA/PBS for 1 h at room temperature. Sections were then equilibrated in sterile $H_2O$ for 5 min then developed using a DAB Substrate kit (Dako, Agilent). Samples were counterstained with haematoxylin, dehydrated and mounted in Cytoseal-60.

**Multiplex immunostaining of human melanoma tissues.** Immunofluorescence multiplex staining was used to determine the colocalization of ATF4 with ACTA2 (αSMA)$^+$ or CD34$^+$ cells. FFPE tissue sections were first rehydrated, subjected to antigen retrieval, peroxidase and serum blocking as described above. Sections were then incubated with primary antibodies against ATF4 (1:200, Abcam, ab31390) or CD34 (1:200, BioLegend, 826401) or IgG isotype controls for ATF4 (1:200, Jackson ImmunoResearch, 011-000-003) and CD34 (1:200, Southern Biotech, 0102-01) in blocking solution overnight at 4 °C. Sections were then treated with an immunofluorescent-labelled secondary antibody against CD34 (1:400, ThermoFisher, A-11005) and a poly-HRP-secondary antibody then consecutive fluorescent-tyramide (ThermoFisher, B40926) against ATF4 (1 h at room temperature). Samples were then stained with an immunofluorescent-labelled primary antibody for ACTA2 (1:500, Millipore-Sigma, F3777) or IgG isotype control (1:500, BD Biosciences, 553456) for 1 h at room temperature. Finally, sections were counterstained for nuclei with 4,6-diamidino-2-phenylindole (DAPI) then aqueous mounted in SlowFade Gold.

**RT–qPCR.** Total RNA was isolated using a Macherey–Nagel kit and complementary DNA was synthesized using a High Capacity RNA-to-cDNA kit according to the manufacturer's instructions. RT–qPCR was performed with Power SYBR green PCR master mix. For data analysis, the QuantStudio 6 Flex Real-Time PCR System (Applied Biosystems) was used. Relative gene expression levels were defined using the DDCt method, and normalization was performed to 18S rRNA. All primers used in this study are described in Supplementary Table 16.

**ChIP.** ChIP assays were performed as previously described[62]. In brief, chromatin crosslinking was carried out with a 3 ml formaldehyde solution for 10 min at room temperature to 30 ml medium. Crosslinking was quenched with 1.65 ml of a 2.5 M glycine solution for 5 min at room temperature. Cells were collected, the cell suspension was centrifuged (4 °C, 5 min, 1,350$g$), and the cell pellet was washed 3 times with PBS. For nuclear extraction, the cell pellet was snap-frozen in liquid nitrogen and thawed at room temperature three times followed by a 10 min incubation at 4 °C in 10 ml ice-cold hypotonic buffer (with rocking). The cell pellet was collected by centrifugation (4 °C, 5 min, 1,350$g$) followed by resuspension in 10 ml ice-cold lysis buffer. Intact nuclei were collected by centrifugation (4 °C, 5 min, 1,350$g$) followed by a wash with 10 ml ice-cold wash buffer for 10 min at room temperature (with rocking). The nuclei were collected by centrifugation (4 °C, 5 min, 1,350$g$) and the pellet was resuspended in 1 ml ice-cold sonication-lysis buffer. Chromatin was sonicated using a Covaris 200 instrument at settings of Temp 5–9, PP200, DF 10, CB 200, for 720 s, clarified by centrifugation (4 °C, 10 min, 18,000$g$). The supernatant was used for immunoprecipitation.

For each ChIP, 30 µl of the magnetic bead slurry was washed 3 times with 1 ml blocking solution against a magnet. The beads were saturated with 0.17 µg anti-ATF4 (rabbit) antibody (according to the manufacturer's instructions) and the corresponding amount of normal rabbit control IgG in 0.5 ml of blocking solution by overnight rotation at 4 °C. Beads washed twice with 1 ml block solution followed by 2 washes with 1 ml FA lysis buffer. ChIP was carried out by mixing 10 µg of sonicated chromatin with the antibody–beads mixture and the FA lysis buffer in a total volume of 0.5 ml and incubated overnight at 4 °C by rotation. The ChIP–beads mixture was washed 5 times with 1 ml ice-cold RIPA wash buffer, and the bound chromatin was eluted in 200 µl TES at 65 °C. The crosslinking was reversed by adding NaCl at a final concentration of 200 mM and by a further incubation of the samples at 65 °C for 18 h. The next two steps included a 1.5 h incubation at 37 °C using 0.2 mg ml$^{-1}$ RNase A and a 1 h incubation at 55 °C using 0.2 mg ml$^{-1}$ proteinase K. DNA was purified using phenol–chloroform extraction followed by ethanol precipitation with overnight incubation at −20 °C. The DNA was resuspended in 100 µl water and DNA enrichment was measured by RT–qPCR. ChIP–qPCR enrichment data were obtained from two biological replicates and presented as mean ± s.e.m. The primer sequences are listed in Supplementary Table 16.

**CM.** Fibroblasts were plated and cultured until 80% confluency. After three washes with PBS, phenol-free DMEM/F12, without any supplement, was added and kept for 24 h. The medium was spun at 300$g$ for 5 min, filtered through a 0.22-µm filter and stored at −80 °C until further use.

**Fibroblast-derived matrices (FDMs).** Fibroblasts were plated on 0.2% gelatin (crosslinked with 1% glutaraldehyde and 1 M ethanolamine)-coated plates (MatTek, P35G-1.5-14-C) and incubated overnight (day 0). Fresh medium supplemented with 75 µg ml$^{-1}$ ascorbic acid was added on days 1, 3, 5 and 7. At day 8, cells were lysed carefully (0.5% Triton X-100 + 20 mM NH$_4$OH), PBS was added and stored at 4 °C overnight to avoid disturbing the matrix. Matrix-coated plates were washed with PBS, and fresh PBS (supplemented with 100 U ml$^{-1}$ penicillin and 100 mg ml$^{-1}$ streptomycin) was added and sealed with Parafilm for up to 2–3 weeks at 4 °C. ECM deposition was first visualized using a Nikon TiE inverted microscope using phase-contrast optics. Fibrillar collagen deposition was detected using second-harmonic generation (SHG) microscopy using a Leica SP8 2-photon microscope with the laser tuned to a wavelength of 900 nm. The SHG 'backwards scatter' signal was imaged (five random fields of view per sample), and the fibrillar collagen area and intensity were quantified using NIS Elements software (v.4.60.00).

**Tube formation assay.** Growth factor reduced (GFR) Matrigel (120 µl per chamber; Corning, 356231) was added to 8-chamber culture slides, incubated at 37 °C for 20 min and washed with PBS. EC$^{WT/WT}$ were pretreated with conditioned medium from LFB$^{WT/WT}$ or LFB$^{Δ/Δ}$ for 24 h, trypsinized and seeded on Matrigel-coated slides at a density of 2 × 10$^4$ cells per chamber. Cells were imaged at 4 h after plating using an Axiovert 40CFL inverted microscope (Zeiss) equipped with AxioCam MRM CCD camera (Zeiss).

**Protein angiogenesis array.** CM was added to the membranes of the proteome profiler mouse angiogenesis array and processed according to the manufacturer's instructions (R&D systems).

**Flow cytometry.** Endothelial cells were stained using a Live/Dead Fixable Aqua Dead Cell Stain kit (Invitrogen, L34957) for live/dead cell discrimination (Supplementary Fig. 1). Cell surface staining against CD31 (1:50, BioLegend, 102508) was performed for 30 min at 4 °C. All data acquisition was done using a FACSCanto II (BD Biosciences) and analysis with FlowJo v.10.

**Gas chromatography–MS.** Samples (up to 50 mg, cut and reweighed as needed) were added to 0.5 ml of 10 mM pH 7.4 HEPES, 1 mM EDTA, 0.1% Triton X100 and 2 mm ʟ-norvaline (internal standard) in Omni Bead Ruptor tubes, and lysed with the setting 5.5 for 30 s. The equivalent of 2.5 mg tissue (25–114 µl or 50–228 nmol norvaline) was transferred to 0.5-ml Eppendorf tubes (duplicates

for 2 lung samples L7, L10 and 1 skin sample S3), and dried for 1 h in a Speedvac. Next, 100 µl 6 N HCl was added and incubated at 105 °C for 15.5 h. Tissue samples (5 µl (lung)) were dried by Speedvac (25 min, 2.5–22.8 nmol norvaline). Sets of standards to run in parallel were also dried. Samples were derivatized with 60 µl 1:1 mix pyridine:MTBSTFA for 60 min at 80 °C. After derivatization, samples were transferred to gas chromatography–MS vials and left at room temperature for 4 days before running the analysis (this was to stabilize the norvaline signal).

**NMR spectrometry.** Cell or tissue samples were extracted using a biphasic extraction protocol. LC–MS-grade methanol and chloroform were purchased from ThermoFisher. Next, 0.22-µm filtered milli-Q water was used for extraction purpose. About 50 mg of tissue and/or 10$^6$ cells were extracted using 500 µl 2:2:1 methanol:chloroform:water. The cell samples were sonicated using a sonicator bath and the tissue samples were homogenized using steel beads in a TissueLyser II system (Eppendorf). The samples were further centrifuged at 16,200$g$, 4 °C. The upper fraction containing the polar metabolites were carefully collected and dried using a vacuum centrifuge (Eppendorf).

The dried samples were dissolved in 200 µl phosphate buffer (pH ~7.1) containing sodium-2,2-dimethyl2-silapentane-5-sulfonate (DSS; Cambridge Isotope) and 10% D$_2$O for field frequency lock purpose (Cambridge Isotope). The samples were transferred to NMR tubes (3 mm i.d., Bruker Biospin).

NMR spectra were acquired using an Avance III HD 700 MHz NMR spectrometer (Bruker Biospin) fitted with a 3 mm NMR triple resonance inverse probe and SampleJet system for automated high-throughput spectral acquisition. All spectra were acquired at 298 K. The pulse program of the acquired NMR spectra took the shape of the first transient of a two-dimensional NOESY and generally of the form RD-90-t-90-tm-90-ACQ. Where RD = relaxation delay, t = small time delay between pulses, tm = mixing time and ACQ = acquisition[63]. Continuous irradiation of water during RD and tm was used to suppress the water signal. The spectra were acquired using 1-s interscan delay, 0.1 s mixing time, 76,000 data points and 14 ppm spectral width with a variable number of scans depending on the starting sample mass. The FIDs were zero-filled to 128 K; 0.1 Hz of linear broadening was applied followed by Fourier transformation.

NMR spectra were imported into Chenomx v.8.0 for quantitative targeted profiling[64]. The processor module was used to phase and baseline correct the spectra followed by internal standard calibration and deletion of the water region. The processed spectra were then imported to the profiler module for the targeted profiling of selected metabolites. Quantified data from this process were exported for further analysis.

**Metabolic tracing study.** For serine-$^{13}$C$_3$ labelling experiments, cells were cultured in RPMI medium lacking glucose, serine and glycine (TEKnova) supplemented with 2 g per litre glucose and 0.03 g per litre serine-$^{13}$C$_3$ (Sigma Aldrich) for 1 and 3 h before collection. Cells were washed twice with ice-cold PBS before extraction with 600 µl of 80:20 acetonitrile:water over ice for 15 min. Cells were scraped off plates to be collected with supernatants, sonicated for 30 s, then spun down at 15,000 r.p.m. for 15 min. A total of 200 µl of supernatant was taken out for LC–MS/MS analysis immediately. Quantitative LC–ESI-MS/MS analysis was performed using an Agilent 1290 UHPLC system equipped with an Agilent 6545 Q-TOF mass spectrometer. A hydrophilic interaction chromatography method (HILIC) with a ZIC-pHILIC column (150 × 2.1 mm i.d., 5 µm; Merck) was used for compound separation at 35 °C with a flow rate of 0.3 ml min$^{-1}$. The mobile phase A consisted of 20 mM ammonium bicarbonate in water and mobile phase B was acetonitrile. The following gradient elution was used: 0–1.5 min, 80% B; 1.5–7 min, 80% B → 40% B; 7–8.5 min, 40% B; 8.5–8.7 min, 40% → 80% B; and 8.7–10 min, 80% B. The overall runtime was 10 min, and the injection volume was 6 µl. Agilent Q-TOF was operated in negative mode and the relevant parameters were as follows: ion spray voltage, 3,500 V; nozzle voltage, 1,000 V; fragmentor voltage, 125 V; drying gas flow, 11 litres per min; capillary temperature, 300 °C; drying gas temperature, 320 °C; and nebulizer pressure, 40 psi. A full scan range was set at 50–1,200 ($m/z$). The reference masses were 119.0363 and 980.0164. The acquisition rate was 2 spectra per s. Data processing was performed with Agilent Profinder B.08.00 (Agilent Technologies). The mass tolerance was set to ±15 ppm and retention time tolerance was ±0.2 min.

For glutamine-$^{13}$C$_5$,$^{15}$N$_2$ labelling experiments, cells were cultured in DMEM lacking glutamine (Gibco) supplemented with 2 mM glutamine-$^{13}$C$_5$,$^{15}$N$_2$ (Cambridge Isotopes) for 2 and 6 h before collection. Metabolite extraction was performed as described above. Quantitative LC–ESI-MS/MS analysis was performed using an Agilent 1290 UHPLC system equipped with an Agilent 6545 Q-TOF mass spectrometer. A HILIC with an Atlantis Silica HILIC Column (100 × 2.1 mm i.d., 3 µm; Waters) was used for amino acid separation at 35 °C with a flow rate of 0.3 ml minl$^{-1}$. The mobile phase A consisted of 10 mM ammonium formate and 0.1% fomic acid in water and mobile phase B was acetonitrile. The following gradient elution was used: 0–1 min, 85% B; 1–4 min, 85% B → 65% B; 4–4.5 min, 65% B → 50% B; 4.5–5.5 min, 50% → 40% B; 5.5–6 min, 40% B → 25% B; 6–7 min, 25% B; and 7–7.5 min, 25% B → 85% B. After the gradient, the column was re-equilibrated at 85% B for 2.5 min. The overall runtime was 10 min, and the injection volume was 3 µl. Agilent Q-TOF was operated in positive mode, and the relevant parameters were as follows: ion spray voltage, 3,000 V; nozzle voltage,

500 V; fragmentor voltage, 125 V; drying gas flow, 11 litres per min; capillary temperature, 300 °C, drying gas temperature, 320 °C; and nebulizer pressure, 40 psi. A full scan range was set at 50–1,200 ($m/z$). The reference masses were 121.0509 and 922.0098. The acquisition rate was 2 spectra per s. Data processing was performed with MAVEN (http://genomics-pubs.princeton.edu/mzroll/index.php).

**Genome-wide gene expression microarray analysis.** Microarray services were provided by the UPENN Molecular Profiling Facility, including quality control tests of the total RNA samples by Agilent Bioanalyzer and Nanodrop spectrophotometry. All protocols were conducted as described in the Affymetrix WT Plus Reagent kit manual and the Affymetrix GeneChip Expression Analysis technical manual. In brief, 250 ng of total RNA was converted to the first-strand cDNA using reverse transcriptase primed by poly(T) and random oligomers that incorporated the T7 promoter sequence. Second-strand cDNA synthesis was followed by in vitro transcription with T7 RNA polymerase for linear amplification of each transcript, and the resulting cRNA was converted to cDNA, fragmented, assessed by Bioanalyzer, and biotinylated by terminal transferase end labelling. A total of 5.5 μg of labelled cDNA were added to Affymetrix hybridization cocktails, heated at 99 °C for 5 min and hybridized for 16 h at 45 °C to Clariom D Mouse Arrays using a GeneChip Hybridization oven 645. The microarrays were then washed at low (6× SSPE) and high (100 mM MES, 0.1 M NaCl) stringency and stained with streptavidin–phycoerythrin. Fluorescence was amplified by adding biotinylated anti-streptavidin and an additional aliquot of streptavidin–phycoerythrin stain. A GeneChip 3000 7G scanner was used to collect fluorescence signals. Affymetrix Command Console and Expression Console were used to quantitate expression levels for targeted genes; default values provided by Affymetrix were applied to all analysis parameters. The Gene Expression Omnibus (GEO) accession number is GSE159020.

**scRNA-seq.** Cells derived from equal volume tumours (small, 150 mm³; large, 300 mm³) from $Atf4^{WT/WT}$ and $Atf4^{\Delta/\Delta}$ mice were loaded into a 10X Genomics Chromium Single-Cell controller following the manufacturer's instructions using a 10×3′ RNA-Seq V2 kit. Illumina sequencing libraries were prepared then sequenced either on three lanes of a HiSeq 4000 (28 bp × 98 bp) or a NovaSeq 6000 (28 bp × 91 bp). Samples were sequenced to a median depth of $14,188 \pm 932.6$ reads per cell with a median $2,291 \pm 334$ median gene count detected per cell. The fraction of reads mapping confidently to the transcriptome was $60 \pm 4.6\%$. The per cent of reads from mitochondrial genes had a median of $6 \pm 0.41\%$. Initial data processing was performed with Cell Ranger v.3.0.1. The GEO accession number is GSE159996.

**Mouse necropsy.** Mouse necropsy was performed according to the Comparative Pathology Core's standardized approach for rodent studies.

**Correlation and survival analysis.** Gene expression profiles (RSEM-normalized gene expression values) were obtained from TCGA[65] for the following cancer types: SKCM and PAAD. Expression profiles were downloaded from Broad GDAC Firehose. For each dataset, Pearson's correlation coefficients between $\log_2$-transformed expression values of *COL1A1*, *COL1A2*, *ACTA2*, *PDGFRB* and *FAP* and $\log_2$-transformed mean expression values of 32 ISR-target genes were estimated. As a baseline, Pearson's correlation coefficients were also estimated between *COL1A1*, *ACTA2*, *PDGFRB*, and 32 genes that were randomly chosen using RSAT[66].

Subsequently, patients in each cancer type were divided into two groups according to *COL1A1* expression: low *COL1A1* expression (below first quartile) and high *COL1A1* expression (above third quartile). Survival analysis using Kaplan–Meier and the log-rank test between *COL1A1*-low and *COL1A1*-high groups, was performed using the R package survival and OncoLnc[67]. Correlation analysis and Kaplan–Meier plots were produced using the R package ggplot2.

**In vivo mouse studies.** All animal experiments were approved by the University Laboratory Animal Resources (ULAR) and IACUC of the University of Pennsylvania regulations (animal protocol 805191). Mice were housed in 12:12 light–dark cycles with temperatures of ~18–23 °C and 40–60% humidity. The maximal tumour volume permitted by ULAR and IACUC is $2 \times 10^3$ mm³ and was not exceeded. Only one tumour in one mouse exceeded this limit (Extended Data Fig. 1d) due to doubling of tumour size over a weekend. The mouse was euthanized immediately. Both males and females (C57BL/6 background) 9–10 weeks old were used in all in vivo experimental procedures. Mice were housed in pathogen-free conditions. ATF4 excision was achieved by oral gavage of tamoxifen (200 mg per kg body weight) for 5 consecutive days. For tumour growth studies, $5 \times 10^5$ B16F10 or MH6419 cells were subcutaneously injected into the flanks of $Atf4^{WT/WT}$ and $Atf4^{\Delta/\Delta}$ mice. For the orthotopic pancreatic tumour model, $5 \times 10^4$ MH6419 cells were orthotopically injected into the tail of the pancreas. For the CD8-depletion studies, before injection of B16F10 cells and then every 4 days after, mice received a dose of 200 μg per mouse of either anti-mouse monoclonal CD8-blocking antibody (InVivoMAb anti-CD8α, BioXCell, BP0061) or rat IgG2b isotype control (InVivoMAb rat IgG2b, BioXCell, BE0090) by intraperitoneal injection. For co-injection studies, $5 \times 10^4$ B16F10 cells mixed with $1.5 \times 10^5$ fibroblasts

($Atf4^{WT/WT}$ or $Atf4^{\Delta/\Delta}$) cells were injected into the flanks of $Atf4^{WT/WT}$ and $Atf4^{\Delta/\Delta}$ mice. For scRNA-seq studies, $5 \times 10^5$ B16F10 cells were injected into the flanks of $Atf4^{WT/WT}$ and $Atf4^{\Delta/\Delta}$ mice. For lung colonization studies, $1.5 \times 10^5$ B16F10 cells were injected in the tail vein of $Atf4^{WT/WT}$ and $Atf4^{\Delta/\Delta}$ mice. For lung metastasis studies, $5 \times 10^5$ B16F10 cells were injected into the flanks of $Atf4^{WT/WT}$ and $Atf4^{\Delta/\Delta}$ mice. Tumours that reached ~300 mm³ were surgically removed, and 4 weeks later mice were euthanized, the lungs were collected and stored in 10% formalin. For in vivo tracking studies, cytoplasmic membranes of both $DFB^{WT/WT}$ and $DFB^{\Delta/\Delta}$ were labelled using the non-cytotoxic Vybrant DiD Cell-Labeling dye immediately before subcutaneous injection. For ex vivo imaging of tumour vasculature, $5 \times 10^5$ B16F10 cells were injected into the flanks of $Atf4^{WT/WT}$ and $Atf4^{\Delta/\Delta}$ mice. Qtracker 705 was injected by the tail vein, mice euthanized, and tumours were collected 20 min after injection. The primers for genotyping used in this study are described in Supplementary Table 16, and the mouse models used in this study are described in Supplementary Table 17.

**In vivo tracking of DiD-labelled DFBs by IVIS imaging.** Mice previously subcutaneously injected in the right flank with DiD-labelled $DFB^{WT/WT}$ and $DFB^{\Delta/\Delta}$ were anaesthetized with isoflurane. Epifluorescence images of DiD were captured using an IVIS Spectrum (Caliper Life Sciences). The excitation and emission filter that was used to capture the DiD fluorescent signal was 648/670 nm. All images were acquired under the same field size and using the autoexposure parameters of the instrument (pixel binning of 8 and f/Stop of 1) to maximize the signal sensitivity. Any changes to the acquisition parameters that control sensitivity are automatically normalized by the Living Image software, without requiring any input by the user. For image analysis, equally sized regions of interest were drawn around the injection site of each mouse, across all time points (day 0 to day 9) and the total flux (photons s⁻¹) was quantified using Living Image Software 4 7.3 (Perkin Elmer).

**Ex vivo imaging of tumour vasculature by confocal/multiphoton microscopy.** Mice previously injected by the tail vein with Qtracker 705 were euthanized, and tumours were collected 20 min after injection. Tumours were washed with PBS and glued to the bottom of a 60-mm petri dish. Samples were covered with cold PBS and imaged. Images of tumour vasculature were acquired on a Leica SP8 MP confocal/multiphoton microscope system with a ×25 (0.95 NA) water-immersion objective. Qtracker was excited by two-photon excitation from a Coherent Chameleon Vision II laser tuned to 910 nm. A 665–705 nm emission filter and an external Hybrid detector were used for signal detection. Z-stacks were acquired at 1.5-μm intervals at 400 Hz with a line averaging of 3.

**Patient samples.** A malignant melanoma with normal skin tissue array (90 males aged $54.8 \pm 12.8$ years and 86 females aged $53.3 \pm 12.9$ years) (Supplementary Table 18) and a pancreas cancer tissue array with adjacent normal pancreas tissue (5 males aged $59.4 \pm 9.6$ years and 5 females aged $61.8 \pm 6.6$ years) (Supplementary Table 19) were purchased from US Biomax. Ethical considerations and protocols that are used in tissue collection of these tissue arrays are mentioned on the Biomax webpage (https://www.biomax.us/FAQs) under FAQ10: "All tissue is collected under the highest ethical standards with the donor being informed completely and with their consent. We make sure we follow standard medical care and protect the donors' privacy. All human tissues are collected under HIPPA approved protocols. All samples have been tested negative for HIV and Hepatitis B or their counterparts in animals and approved for commercial product development". FFPE human melanoma tumours were obtained from patients that underwent resection at the Hospital of the University of Pennsylvania after signing informed consent in accordance with IRB protocol number 703001 (Supplementary Table 20).

**Quantification and statistical analysis.** *scRNA-seq data analysis.* scRNA-seq data have been deposited in the GEO database[68] at NCBI. Illumina basecall files were converted to FASTQ format and aligned to the reference GRCm38 genome using CellRanger (v.3.1)[69] with default parameters. Gene expression matrices were computed using ENSEMBL gene annotation. Count matrices were then processed using the R environment (R v.3.6.0, RStudio v.1.1.442) and Seurat (v.3.2)[70,71]. Before proceeding to further analysis, four $Atf4^{WT/WT}$ and four $Atf4^{\Delta/\Delta}$ B16F10 large tumour samples were merged into one $Atf4^{WT/WT}$ and one $Atf4^{\Delta/\Delta}$ using the Seurat merge function. Cells expressing fewer than 200 or more than 5,000 unique genes and cells with more than 7.5% mitochondrial content were discarded. Genes expressed in fewer than 3 cells were also discarded. Library size normalization was performed using the Seurat NormalizeData function with the method LogNormalize and a scaling factor of 10,000 (default). The 2,000 most variable genes across datasets were identified using the FindVariableFeatures function. Subsequently, pre-processed Seurat objects from both conditions (WT/WT and Δ/Δ) were used to create an integrated assay using FindIntegrationAnchors and IntegrateData functions. A linear transformation was applied to the datasets using ScaleData function with the following configuration: vars.to.regress = 'percent.mt'. To conduct unsupervised clustering of single-cells, linear (principal component analysis (PCA)) and nonlinear (UMAP) dimensionality reduction techniques were combined. The first 20 and 15 principal components (PCs) were used for the

large and small tumours, respectively. Graph-based clustering was performed by using the K-nearest neighbour (KNN) graph and the Louvain algorithm. Initial clustering revealed 16 and 12 clusters in large and small tumours, respectively. UMAP was used to visualize clustering results. The FindConservedMarkers function was utilized to identify cell-type marker genes that were conserved across both conditions using the following parameters: grouping.var = 'condition', only. pos = T, logfc.threshold = 0.25.

Expression levels of a number of canonical marker genes led to the conduction of subclustering in some of the initial clusters in both large and small tumours. Specifically, in small tumours, subclustering of cluster 6 revealed 7 subclusters, from which subcluster 2 was expressing numerous T/NK cell marker genes and was therefore separated from cluster 6 and projected back to the original small tumours UMAP. In large tumours, subclustering of cluster 10 revealed 8 subclusters, from which subclusters 4 and 5 were expressing numerous T/NK cell and dendritic cell marker genes, respectively. Both subclusters 4 and 5 were separated from cluster 10 and projected back to the original large tumours UMAP. In addition, cluster 14 was removed in silico due to the high mitochondrial content in most of its cells, which led to the conclusion that it was a group of dead cells. Furthermore, a number of neighbouring clusters that shared highly similar expression profiles were merged (cluster 0 with 13, cluster 2 with 4, cluster 3 with 7 and cluster 6 with 9). After these processes, small tumours and large tumours comprised 12 and 16 clusters, respectively. Cell-type identification was carried out by using numerous known marker genes described in the literature and PanglaoDB[72]. To identify differentially expressed genes for each of the cell types between $Atf4^{WT/WT}$ and $Atf4^{\Delta/\Delta}$ samples, the function FindMarkers with the Wilcoxon rank-sum test was used. Specifically, for the cluster comprising CAFs, subclustering was performed and revealed 3 and 5 subclusters in small and large tumours, respectively. Differentially expressed genes in CAFs were separated in upregulated and downregulated genes, and pathway analysis was performed with the R package hypeR[73] using a hypergeometric enrichment test and the Reactome[74] gene set. Finally, small and large tumours were integrated in a common assay, the Seurat object was processed similarly as above and trajectory inference in the CAF population was conducted with Slingshot[75].

*Microarray data analysis.* Cel files were imported into TAC (Transcriptome Analysis Console v.4, ThermoFisher), where they were RMA-normalized using default parameters. Inter-sample comparisons were visualized with PCA to confirm the absence of technical outliers. Contrasts comparing (experimental group) versus (control group) were performed. Genes that had false discovery rate values of ≤0.05 and a fold-change greater than 1.5 in either direction were considered significant.

*ChIP-seq data analysis.* The ChIP-seq datasets GSM873426 and GSM873427 were retrieved from the GEO[68] repository, corresponding to three $Atf4^{WT/WT}$ and three $Atf4^{\Delta/\Delta}$ mouse ChIP-seq experiments. Raw datasets were quality checked and pre-processed using FastQC[76] and Cutadapt[77]. Alignment was conducted using bowtie2 (ref. [78]) against the GRCm38 version of the mouse genome. Genome-wide peaks were derived using MACS2 (ref. [79]) (using default parameters) with both immunoprecipitated $Atf4\ WT$ and $Atf4\ KO$ samples. Best candidate motifs were found by using the genome sequences from the peak regions and the de novo motif identification method from MEME suite[80]. Filtering of the peak regions was conducted, and peaks lying inside the $Col1a1$ gene body and 3,000 nucleotides upstream of the transcription start site of $Col1a1$ were kept. FIMO from MEME-suite[80] was utilized and scanning for motif occurrences inside the filtered peak regions was performed. Three strong potential binding sites emerged, all of them residing inside intron 5 of $Col1a1$ gene.

*RT–qPCR gene expression analysis.* For RT–qPCR, relative gene expression levels were defined using the DDCt method and normalization was performed to 18S rRNA.

*Digital imaging and quantification.* Digital images were captured on a Nikon TiE inverted microscope (Nikon Instruments), a Zeiss Observer.Z1 inverted microscope (Zeiss) or a Zeiss LSM 710 confocal microscope (Zeiss). The motorized stage and automated image-stitching parameters were used to obtain complete images of each tumour/tissue array. Immunohistochemical stains were captured using brightfield optics and a Nikon DSRi2 colour camera at ×20 magnification. Multi-channel immunofluorescent samples were captured on a CCD Hammamatsu Camera at ×10 magnification and an AxioCam MRm (Zeiss) at ×10 and ×20 magnification, using independently controlled excitation and emission wavelength filter-wheels for each specific fluorophore. NIS Elements software v.4.60.00 (Nikon) and Zeiss Zen 3.0 software (Zeiss) were used for image quantification and pseudo-colouring of immunofluorescent images where appropriate. Also, ImageJ Fiji[81] was used for image quantification. Parameters in the tube formation assay were analysed in ImageJ Fiji using a code for the morphometric analysis of 'Endothelial Tube Formation Assay'[82]. Imaris 9.8.1 imaging software was used to analyse images capture by confocal/multiphoton microscope.

**Statistical analysis.** Parameters such as sample size, the number of independent experiments (mean ± s.e.m.) and statistical significance are reported in figures and figure legends. Results were considered statistically significant when $P < 0.05$ by

the appropriate test (analysis of variance, log-rank, *t*-test, Pearson's correlation). Student's *t*-test and hypergeometric test were utilized for comparisons in experiments with two sample groups. Survival data were summarized with Kaplan–Meier methods and tested using log-rank test. In box and whisker plots, the box represents the median and the quartiles, and the whisker expresses the minimum up to maximum value and plots each individual value as a point superimposed on the graph. Statistical analyses were performed with Excel 365 and GraphPad Prism v.8.

**Reporting summary.** Further information on research design is available in the Nature Research Reporting Summary linked to this article.

## Data availability
The scRNA–seq and microarray data that support the findings of this study have been deposited in the GEO under accession codes GSE159996 and GSE159020. The human SKCM and PAAD data were derived from Broad GDAC Firehose: https://gdac.broadinstitute.org/. Publicly available ChIP-seq datasets used in this study were retrieved from the GEO repository and can be found under the accession codes GSM873426 and GSM873427.

All other data and scripts supporting the findings of this study are available from the corresponding author on reasonable request. Source data are provided with this paper.

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

## Acknowledgements
This work was supported by NIH grants P01 CA165997, and 1R01CA268597 to C.K. and P01 CA217805 to E.P. I.I.V. was supported in part by the National Center for Advancing Translational Sciences of the National Institutes of Health under award number UL1TR001878 and the Institute for Translational Medicine and Therapeutics' (ITMAT)

Transdisciplinary Program in Translational Medicine and Therapeutics. The content is solely the responsibility of the authors and does not necessarily represent the official views of the NIH. A.G.H. was supported by the Hellenic Foundation for Research and Innovation (HFRI) under the 'First Call for HFRI Research Projects to support Faculty members and Researchers and the procurement of high-cost research equipment grant' (project number 2563). This study was also supported by an American Cancer Society Research Scholar Grant (RSG-20-036-01) and a Stanford Maternal and Child Health Research Institute Research Scholar Award to J.Y. G.S. is supported by the Operational Programme 'Human Resources Development, Education and Lifelong Learning' in the context of the project 'Strengthening Human Resources Research Potential via Doctorate Research' (MIS-5000432), implemented by the State Scholarships Foundation (IKY), in the form of a PhD Scholarship. H.A. and V.S.W. were partially supported by a summer undergraduate training grant (SUPERS, 5R25-CA140116-10). We thank the members of the Koumenis Lab for helpful suggestions and for critically reading the manuscript; B. Stanger for providing the pancreatic cancer cell line; and staff at the University of Pennsylvania Veterinary Comparative Pathology Core, the Molecular Profiling Facility, the Next-Generation Sequencing Core, the Penn Vet Imaging Core (PVIC) and the Microscopy Core for their valuable assistance with this project.

## Author contributions

I.I.V. designed and conducted the experiments and acquired, analysed and interpreted the data. H.A., N.M.L., I.V.K., B.I.B., F.C., K.K., A.V., C.S.S., V.S.W., Y.L., J.Y., D.A.S., A.L.O., M.H., D.Z., Y.F., A.S. and A.W. conducted the experiments and acquired the data. J.M. and E.P. performed immunofluorescence and immunohistochemistry and analysed and interpreted the results. F.R., P.K. and J.-C.M. aided in the conceptual design of scRNA-seq experiments and interpreted the results. G.S. and A.G.H. performed the scRNA-seq analysis, analysed TCGA data and interpreted the results. E.R. performed mouse necropsy. J.W.T. analysed the data from the genome-wide microarray analysis. X.X. provided the human melanoma tissues and contributed to the interpretation of the resultant data. S.R. provided guidance and support on the tube formation assays. J.A.D. and S.Y.F. aided in the conceptual design of metastasis experiments and provided key reagents. All authors read and edited the manuscript. I.I.V. and C.K. conceptualized and designed the research study and wrote the manuscript.

## Competing interests

C.K. is the scientific founder and holds equity position in Veltion Therapeutics. The remaining authors declare no competing interests.

## Additional information

**Extended data** is available for this paper at https://doi.org/10.1038/s41556-022-00918-8.

**Correspondence and requests for materials** should be addressed to Constantinos Koumenis.

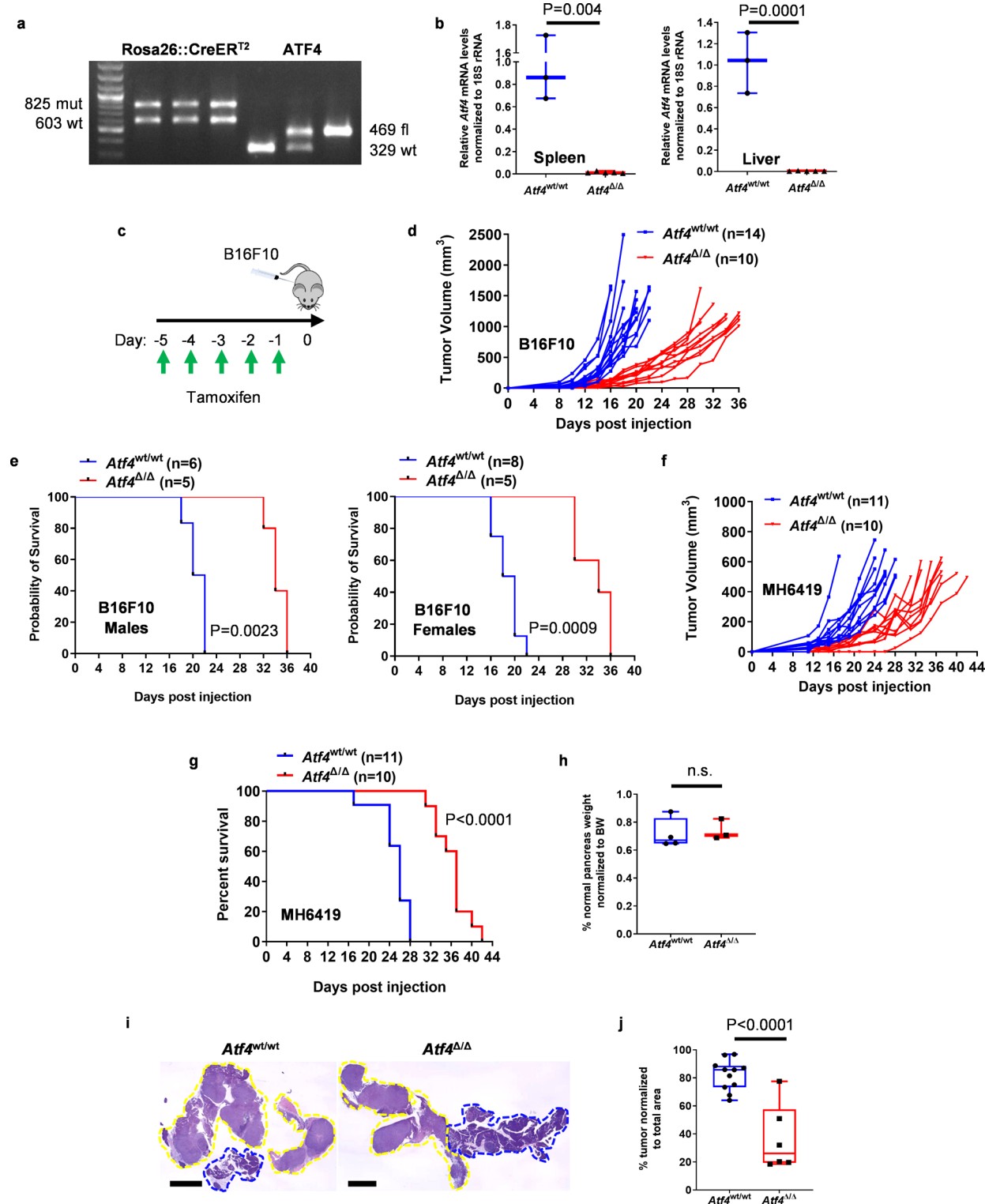

**Extended Data Fig. 1 | Host ATF4 deletion causes a transient weight loss and gender-independent increase in survival of B16F10 cells-injected mice.**
**a**, Genotyping of mice for Rosa26::CreER[T2] and ATF4 status. **b**, Box and whisker plot display the RT-qPCR of *Atf4* in spleen and liver (n = 3–5 biologically independent samples per group). Unpaired two-sample t-test. **c**, Schema for injection of tumour cells post tamoxifen treatment. **d**, Tumour growth curves of single mouse plotted from B16F10 cells-injected mice. **e**, Kaplan-Meier survival analysis by gender from d. Log-rank (Mantel-Cox) test. **f**, Tumour growth curves of single mouse plotted from MH6419 cells-injected mice. **g**, Kaplan-Meier survival analysis of the mice following 5×10[5] MH6419 cell injection. Log-rank (Mantel-Cox) test. **h**, Box and whisker plot display % normal pancreas weight normalized to BW. **i**, Representative images of pancreas from orthotopic pancreatic tumour model, stained for H&E. The yellow and blue dotted lines indicate the tumour and normal areas of pancreas, respectively. **j**, Box and whisker plot of the % tumour area normalized to total area. Unpaired two-sample t-test.

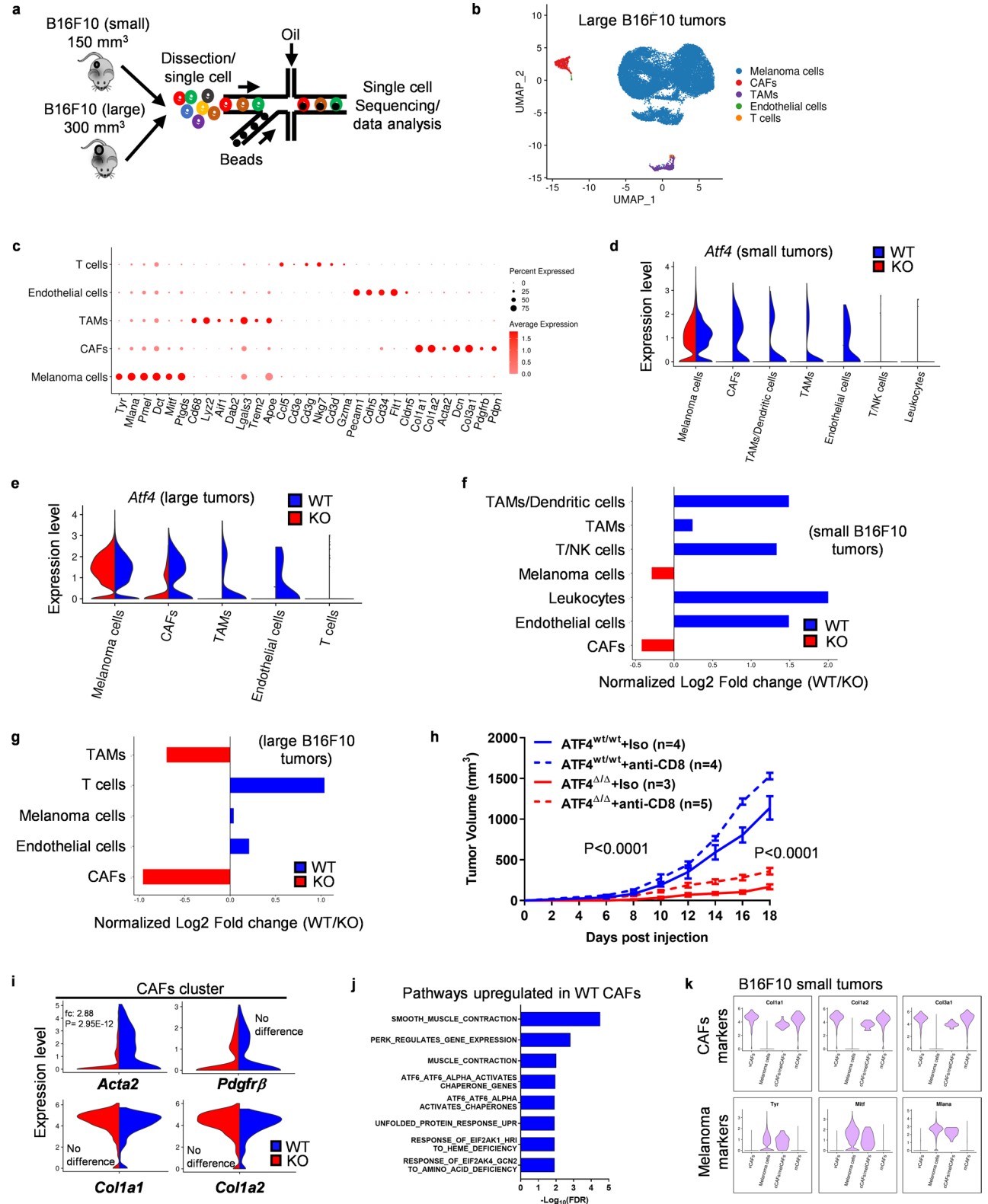

**Extended Data Fig. 2 | See next page for caption.**

**Extended Data Fig. 2 | scRNA-seq in small and large size B16F10 tumours. a**, Schematic for harvesting B16F10 tumours at small (150 mm$^3$) and large (300 mm$^3$) volumes and processing for scRNA-seq. **b**, UMAP plot of cells from 4 pooled large B16F10 tumours (300 mm$^3$) from each genotype. **c**, Dot plot displaying selected gene markers across all clusters. The colour intensity represents the average expression while the size of dots indicates the percentage of cells expressing each gene. **d**, Violin plots showing the expression of ATF4 across all clusters in small sized tumours and **e**, in large sized tumours. **f**, Bar plot displaying the normalized Log2 fold change of cell types in each cluster in small sized tumours and **g**, in large sized tumours. **h**, Tumour growth curves of *Atf4*^wt/wt and *Atf4*^Δ/Δ mice treated with Isotype or anti-CD8 antibody following 5×10$^5$ B16F10 cell injection (n = 3–5 biologically independent samples per group). Values represent mean ± SEM. Two-way ANOVA analysis (until day 18). **i**, Violin plots showing the expression of the indicative markers at the CAFs cluster. **j**, Bar plot displaying the eight most significant Reactome pathways enriched in *Atf4*^wt/wt CAFs using the genes upregulated in *Atf4*^wt/wt.
**k**, Violin plots showing the expression of the indicative CAF and melanoma markers at the CAFs subclusters and melanoma cells (small B16F10 tumours).

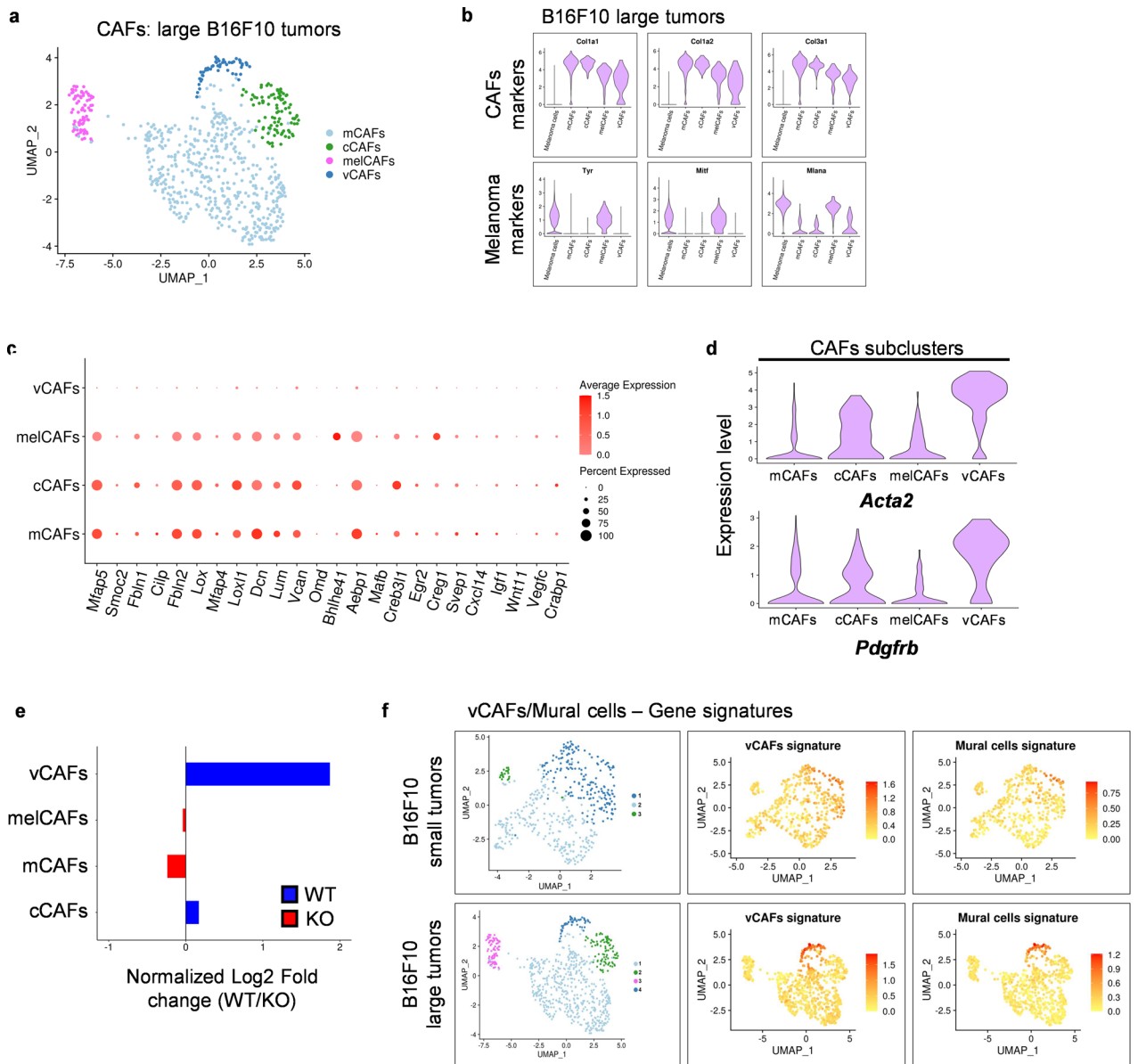

**Extended Data Fig. 3 | Gene signature of CAFs and mural cells. a**, UMAP plot after re-clustering of the CAF cell type in the data set from Extended Data Fig. 2b. **b**, Violin plots showing the expression of the indicative CAF and melanoma markers at the CAFs subclusters and melanoma cells (large B16F10 tumours). **c**, Dot plot displaying selected gene markers across CAFs subclusters. **d**, Violin plots showing the expression of *Acta2* and *Pdgfrβ* in CAFs subclusters. **e**, Bar plot displaying the normalized Log2 fold change (WT/KO) of CAFs subclusters in each genotype. **f**, UMAP plots of B16F10 small (top row) and large (bottom row) tumours coupled with the expression signal of the vCAFs- and mural cells-specific gene signatures.

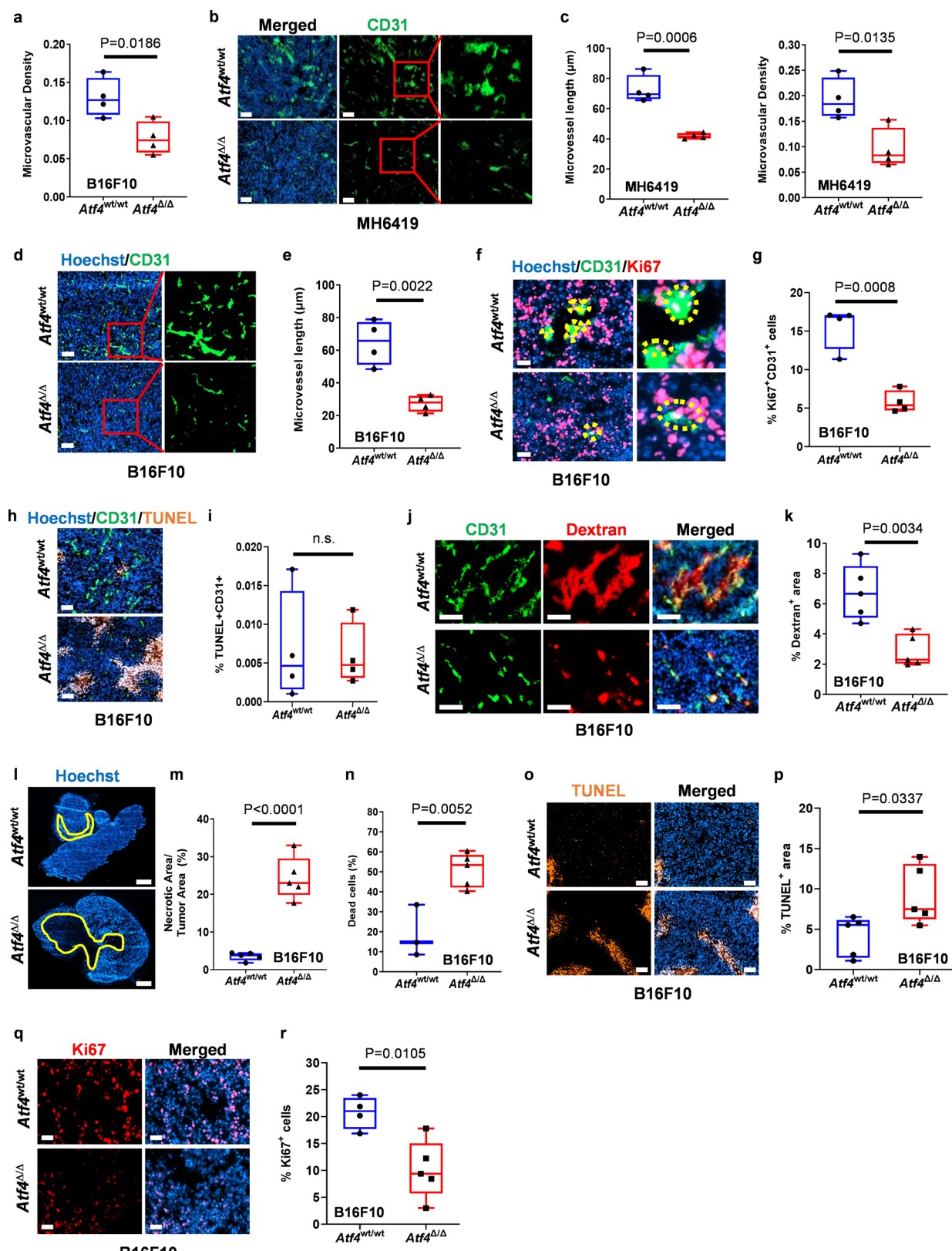

**Extended Data Fig. 4 | See next page for caption.**

**Extended Data Fig. 4 | ATF4 loss is associated with abnormal vascularization and extensive intratumoural necrosis. a**, Box and whisker plot display of the microvascular density in sections from B16F10 tumours (n = 4 biologically independent samples per group). **b**, Representative IF images from MH6419 tumours (appr. 300 mm³) stained for CD31 (green). Original magnification, 10x, 28x (insets). **c**, Box and whisker plots display the microvessel length (in μm) (left) and microvascular density (n = biologically independent samples 4 per group) (right) from b. **d**, Representative IF images from B16F10 tumours (appr. 1000 mm³) stained for CD31 (green). Original magnification, 10x, 28x (insets). **e**, Box and whisker plot of the microvessel length (in μm) from d (n = 4 biologically independent samples per group). **f**, Representative IF images from B16F10 tumours stained for CD31 (green) and Ki67 (red). The yellow dotted circles indicate the proliferative endothelial cells. Magnification, 20x. **g**, Box and whisker plot of %Ki67⁺CD31⁺ cells from f. **h**, Representative IF images from B16F10 tumours stained for CD31 (green) and TUNEL (orange). Magnification, 10x. **i**, Box and whisker plot the %TUNEL⁺CD31⁺ cells from h (n = 4 biologically independent samples per group). **j**, Representative IF images from mice i.v.-injected with Texas Red-Dextran post-ATF4 deletion. Magnification, 20x. **k**, Box and whisker plot of % Dextran positive area from j (n = 5 biologically independent samples per group). **l**, Representative IF images from B16F10 tumours stained for Hoechst only. The yellow highlighted area indicates tumour necrosis. **m**, Box and whisker plot display the ratio of tumour necrotic area over the total tumour area from l (n = 5 biologically independent samples per group). **n**, Flow cytometric analysis and box and whisker plot of the % dead cells in B16F10 tumours (n = 3–5 biologically independent samples). **o**, Representative IF images from B16F10 tumours stained for TUNEL, as a marker of apoptosis. Magnification, 10x. **p**, Box and whisker plot displays the %TUNEL positive area from o (n = 5 biologically independent samples per group). **q**, Representative IF images from B16F10 tumours stained for Ki67, as a marker of proliferation. Magnification, 10x. **r**, Box and whisker plot the %Ki67 positive cells from q (n = 4–5 biologically independent samples per group). Unpaired two-sample t-test in all box and whisker plots. Scale bars, 100 μm (b, d, f, h, j, o and q), 1 mm (l).

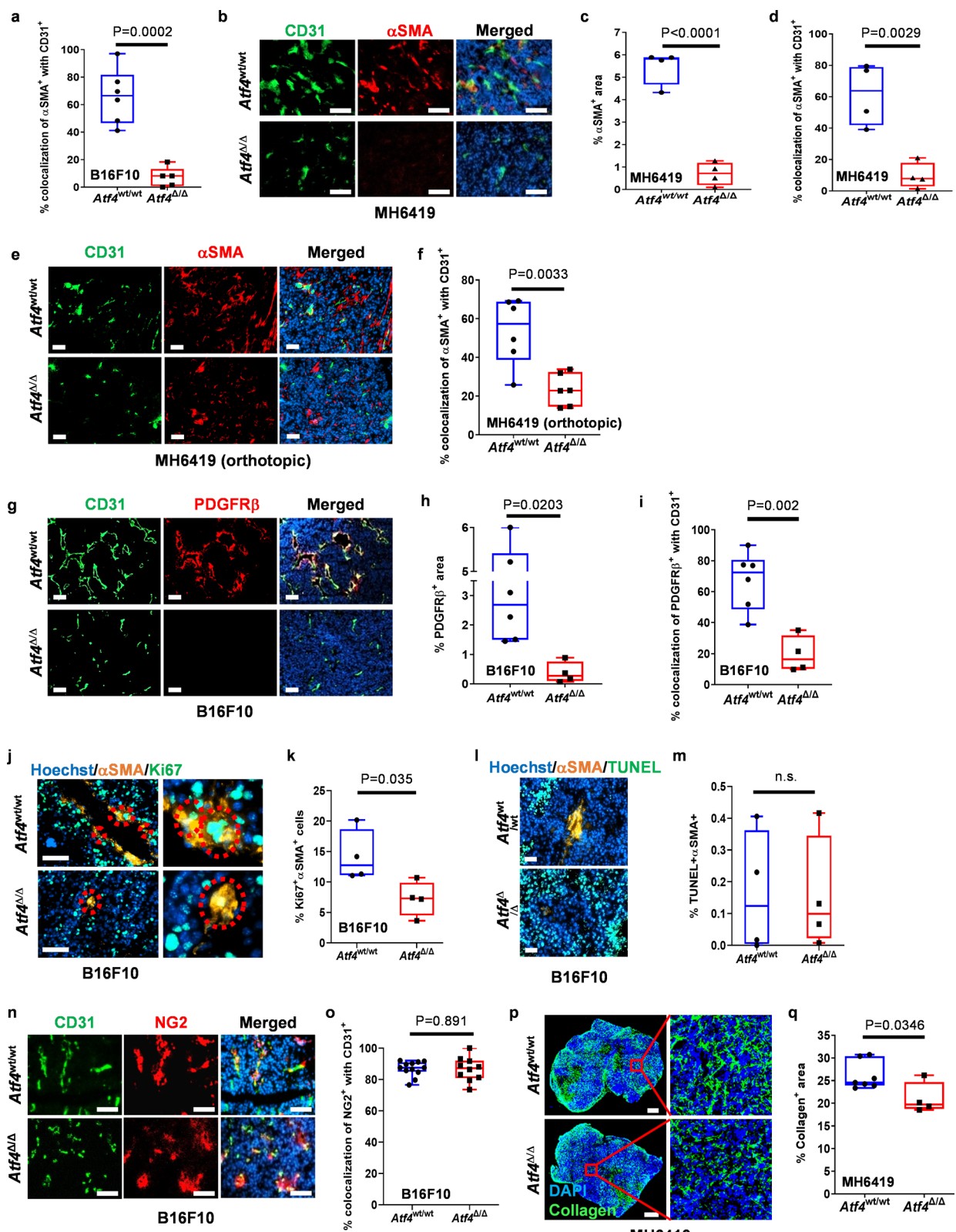

**Extended Data Fig. 5 | See next page for caption.**

**Extended Data Fig. 5 | ATF4 ablation is associated with reduced levels of CAFs markers, proliferation and collagen deposition in the TME. a**, Box and whisker plot of the % colocalization of aSMA+ with CD31+ in B16F10 tumours (n = 5–6 biologically independent samples per group). **b**, Representative IF images from MH6419 tumours stained for CD31 (green) and αSMA (red). Magnification, 20x. **c**, Box and whisker plots of the % αSMA positive area and **d**, of the % colocalization of aSMA+ with CD31+from b (n = 4 biologically independent samples per group). **e**, Representative IF images from orthotopic MH6419 pancreatic tumours stained for CD31 (green) and αSMA (red). Magnification, 10x. **f**, Box and whisker plot of the % colocalization of aSMA+ with CD31+ from e. **g**, Representative IF images from B16F10 tumours stained for CD31 (green) and PDGFRβ (red). Magnification, 10x. **h**, Box and whisker plot of the % PDGFRβ positive area and **i**, of the % colocalization of PDGFRβ+ with CD31+ from g (n = 4–6 biologically independent samples per group). **j**, Representative IF images from B16F10 tumours stained for αSMA (orange) and Ki67 (green). Magnification, 20x. **k**, Box and whisker plot of %Ki67+αSMA+ cells from j. **l**, Representative IF images from B16F10 tumours stained for αSMA (orange) and TUNEL (green). Magnification, 10x. **m**, Box and whisker plot the %TUNEL+αSMA + cells from l (n = 4 biologically independent samples per group). **n**, Representative IF images from B16F10 tumours stained for CD31 (green) and NG2 (red). Magnification, 20x. **o**, Box and whisker plot of % colocalization of NG2+ with CD31+ from n (n = 10–12 biologically independent samples per group). **p**. Representative images from MH6419 tumour sections stained with an antibody against collagen (green). **q**, Box and whisker plot of the % of positive collagen area from p (n = 4–7 biologically independent samples per group). Unpaired two-sample t-test in all box and whisker plots. Scale bars, 100 μm (b, e, g, j, l and n) and 1 mm (p).

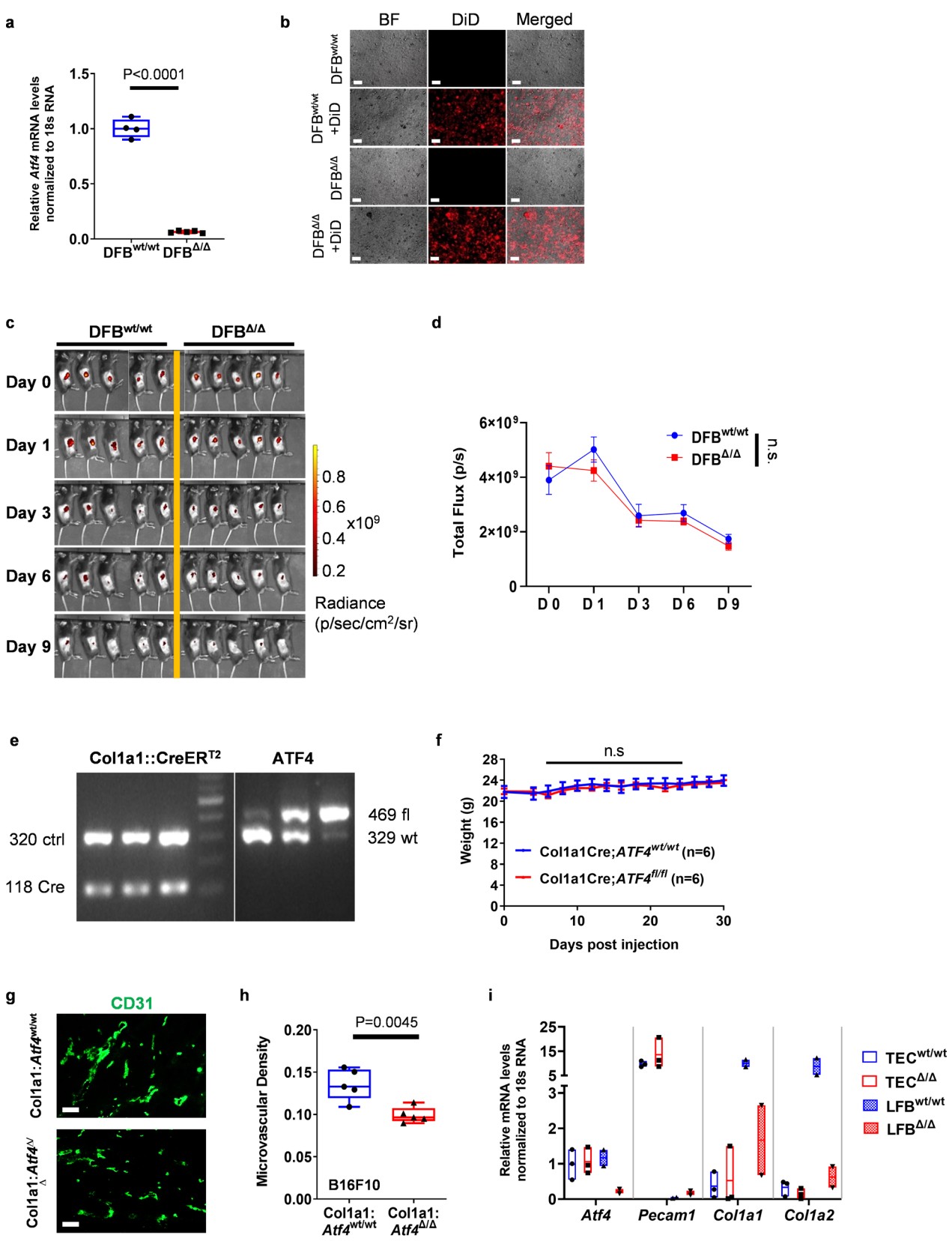

**Extended Data Fig. 6 | See next page for caption.**

**Extended Data Fig. 6 | Fibroblast specific ATF4 deletion causes abnormal vascularization. a**, Box and whisker plot display RT-qPCR of *Atf4* in DFB cells (n = 4–5 biologically independent samples per group). Unpaired two-sample t-test. **b**, Representative images of labelled DFBs with the Vybrant™ DiD. Magnification, 10x. **c**, *In vivo* tracking of the DFB + DiD cells injected to the flank of the mice by IVIS imaging. **d**, Quantification of the fluorescent signal from c. Values represent mean ± SEM, unpaired two-sample t-test. **e**, Genotyping of mice for *Col1a1*::CreER$^{T2}$ and ATF4 status. **f**, No BW changes post ATF4 excision in *Col1a1*Cre;*Atf4*$^{wt/wt}$ and *Col1a1*Cre;*Atf4*$^{\Delta/\Delta}$ mice (n = 6 biologically independent samples per group); n.s. = not significant. Values represent mean ± SEM, two-way ANOVA analysis. **g**, Representative IF images from B16F10 tumours grown in *Col1a1*Cre;*Atf4*$^{wt/wt}$ and *Col1a1*Cre;*Atf4*$^{\Delta/\Delta}$ mice stained for CD31 (green). Magnification, 10x. **h**, Box and whisker plot display the microvascular density from g (n = 5 biologically independent samples per group). Unpaired two-sample t-test. **i**, Floating bars display the RT-qPCR of *Atf4, Pecam1, Col1a1* and *Col1a2* in isolated tumour endothelial cells (TEC) and LFBs (n = 2–3 biologically independent samples per group). Light gray lines are used to separate the groups for each gene. Scale bars, 100 µm (b and g).

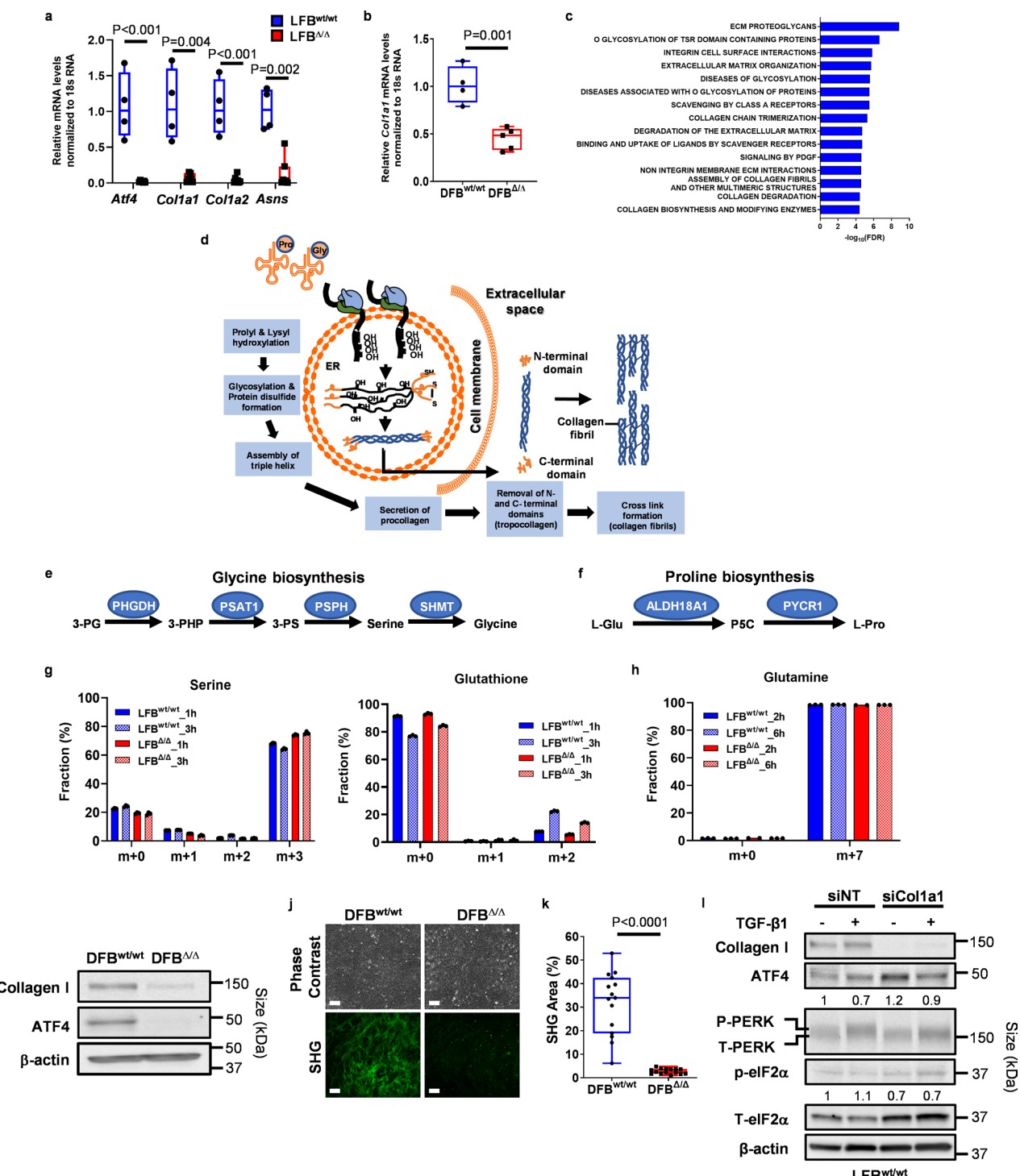

**Extended Data Fig. 7 | ATF4 loss reduces intracellular and secreted collagen in DFBs. a**, Box and whisker plot display RT-qPCR of *Atf4*, *Col1a1*, *Col1a2* and *Asns* in LFBs (n = 4–7 biologically independent samples per group). **b**, Box and whisker display RT-qPCR of *Col1a1* in DFBs (n = 4–5 biologically independent samples per group). **c**, Bar plot displaying the negative log10 FDR of the fifteen most significantly upregulated gene ontology terms enriched in LFB^WT/WT compared to LFB^Δ/Δ cells. **d**, Schematic of the collagen synthesis pathway. **e**, Schematic of the enzymes involved in glycine and **f**, proline biosynthesis pathways. **g** and **h**, LC-ESI-MS/MS analysis to measure the precursor serine-$^{13}C_3$ and glutamine-$^{13}C_5$$^{15}N_2$ and the metabolic flux from serine to glutathione in LFB^WT/WT and LFB^Δ/Δ cells, respectively (n = 3 biologically independent samples per group). Values represent mean + SEM. # and & indicate a statistically significant change from the LFB^WT/WT at each isotopologue (# p < 0.001, & p < 0.01). **i**, Proteins were detected by immunoblotting in untreated DFBs. β-actin was used as a loading control. **j**, Representative images of collagen deposition from DFB^WT/WT and DFB^Δ/Δ using second harmonic generation (SHG) microscopy. Magnification, 10x. Scale bar, 100 μm. **k**, Box and whisker plot display the fluorescent signal from j. Each dot represents quantitative value from a 10x field. **l**, LFB^WT/WT were treated with TGF-β1 for 24 h and proteins were detected by immunoblotting. β-actin was used as a loading control. Numbers below blots represent relative band intensities, normalized to T-eIF2a and β-actin. Unpaired two-sample t-test in all box and whisker plots.

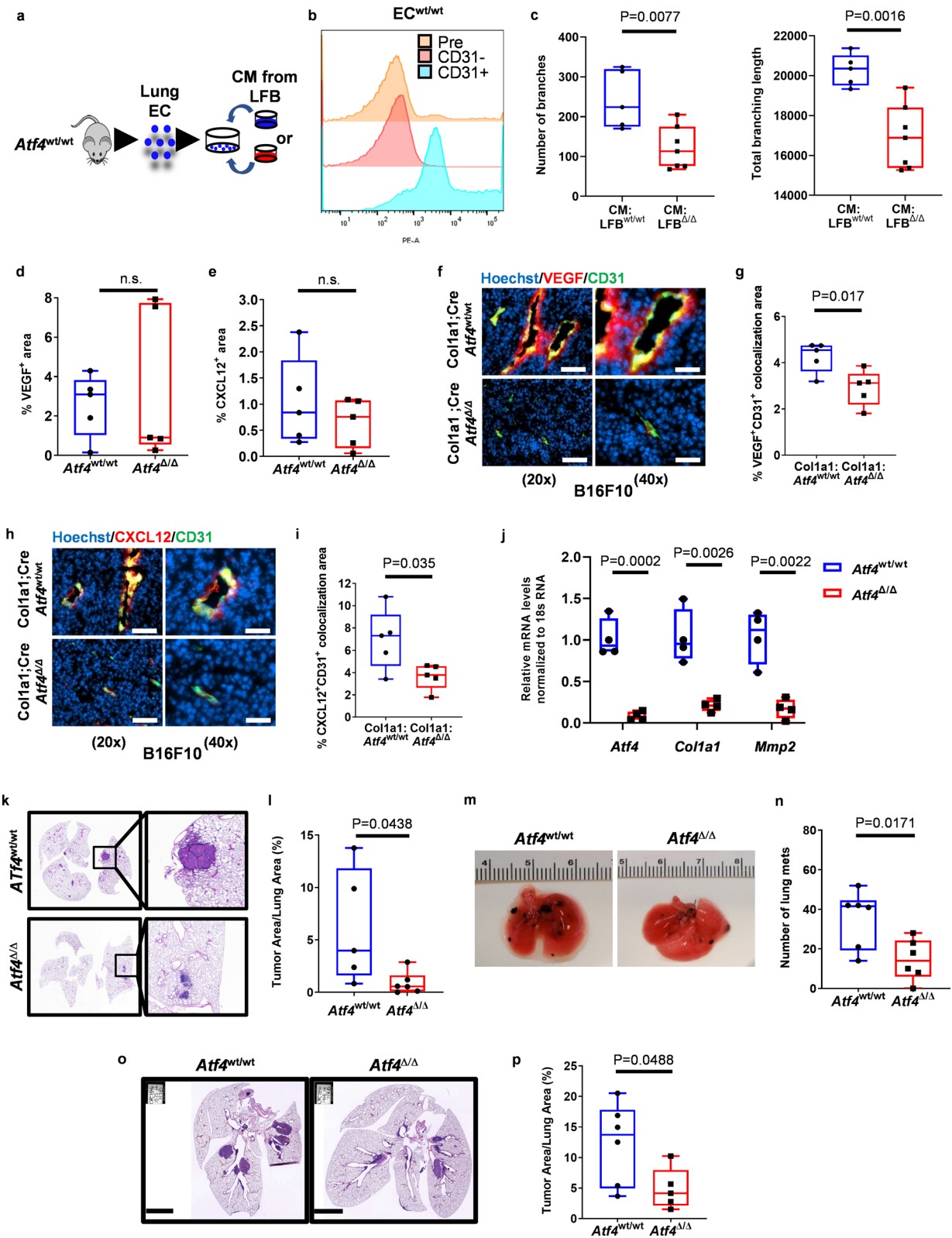

**Extended Data Fig. 8 | See next page for caption.**

**Extended Data Fig. 8 | Reduced sprouting and tube formation of ECs treated with CM from LFB$^{\Delta\Delta}$. a**, Schematic of in vitro analysis of angiogenic activity of fibroblast-derived conditioned medium (CM). **b**, Flow cytometry to confirm the purity of the isolated lung ECs (pre = pre magnetic beads separation; CD31-, the negative fraction of beads; CD31+, the positive fraction of beads). **c**, Box and whisker plots of number of branches and total branching length from the tube formation experiment. **d**, Box and whisker plot of %VEGF$^+$ area and **e**, of %CXCL12$^+$ area. **f**, Representative IF images from B16F10 tumours grown in *Col1a1*Cre;*Atf4*$^{wt/wt}$ and *Col1a1*Cre;*Atf4*$^{\Delta/\Delta}$ mice stained for VEGF (red) and CD31 (green). Magnification, 20x and 40x. **g**, Box and whisker plot display the %VEGF$^+$CD31$^+$ colocalization area from f (n = 5 biologically independent samples per group). Scale bar, 100 μm for 20x and 50 μm for 40x. **h**, Representative IF images from B16F10 tumours grown in *Col1a1*Cre;*Atf4*$^{wt/wt}$ and *Col1a1*Cre;*Atf4*$^{\Delta/\Delta}$ stained for CXCL12 (red) and CD31 (green). Magnification, 20x and 40x. **i**, Box and whisker plot display the %CXCL12$^+$CD31$^+$ colocalization area from h (n = 5 biologically independent samples per group). Scale bar, 100 μm for 20x and 50 μm for 40x. **j**, Box and whisker plot display RT-qPCR of indicative markers in *Atf4*$^{wt/wt}$ and *Atf4*$^{\Delta/\Delta}$ lungs (n = 4 biologically independent samples per group). **k**, Representative images of serial sections of lungs stained for H&E. **l**, Box and whisker plot of the percentage of lung tumour area from f (n = 5-6 biologically independent samples per group). **m**, Representative images from lungs, harvested at 3 weeks post injection (with tamoxifen treatment to start 3 days post injection). **n**, Box and whisker plot of the number of lung metastases. **o**, Representative images of serial sections of lungs stained for H&E. Scale bar, 3 mm. **p**, Box and whisker plot of the percentage of lung tumour area from k (n = 5–6 biologically independent samples per group). Unpaired two-sample t-test in all box and whisker plots.

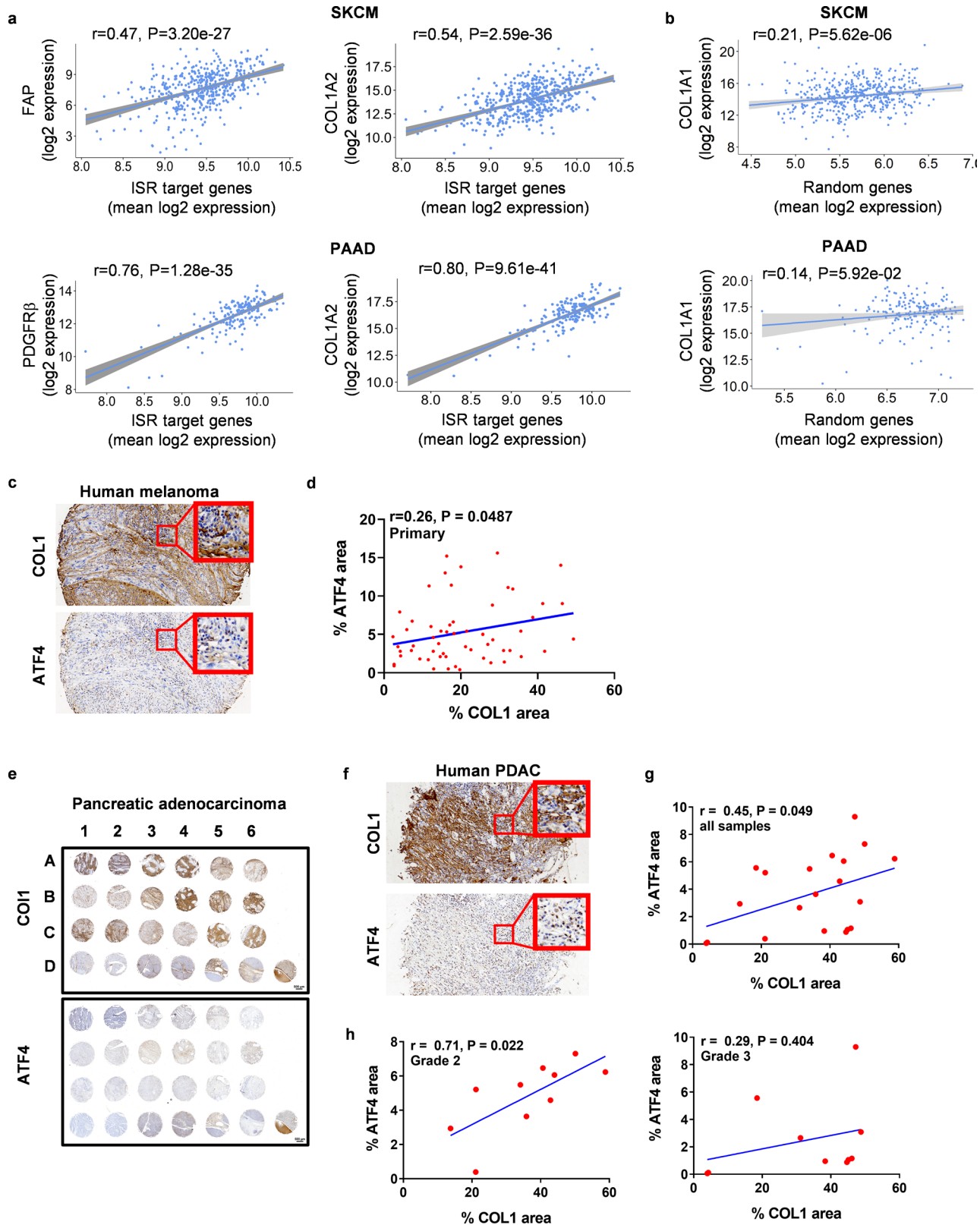

**Extended Data Fig. 9 | See next page for caption.**

**Extended Data Fig. 9 | High ATF4 levels correlate with increased COL1 expression on human melanoma and pancreatic tumours. a**, Pearson correlation between the ISR target signature and COL1A2, FAP and PDGFRβ in Skin Cutaneous Melanoma (SKCM) and Pancreatic adenocarcinoma (PAAD). The linear regression lines along with 95% confidence intervals (shaded regions) are shown. **b**, Pearson correlation between randomly selected genes and COL1A1 in Skin Cutaneous Melanoma (SKCM) and Pancreatic adenocarcinoma (PAAD). The linear regression lines along with 95% confidence intervals (shaded regions) are shown. **c**, Representative images from immunohistochemical staining for COL1 and ATF4 of human melanoma tissues **d**, Pearson correlation between the % ATF4 area and % COL1 area in primary group from the human melanoma tissue array. **e**, Pancreatic adenocarcinoma tissue array containing sections from 24 tumours were stained for COL1 (upper panel) and ATF4 (lower panel) proteins. **f**, Representative images from immunohistochemical staining for COL1 and ATF4 of human pancreatic adenocarcinoma tissues. **g**, Pearson correlation between the % ATF4 area and % COL1 area in all samples from e and **h**, in grade 2 (left) and grade 3 (right) groups.

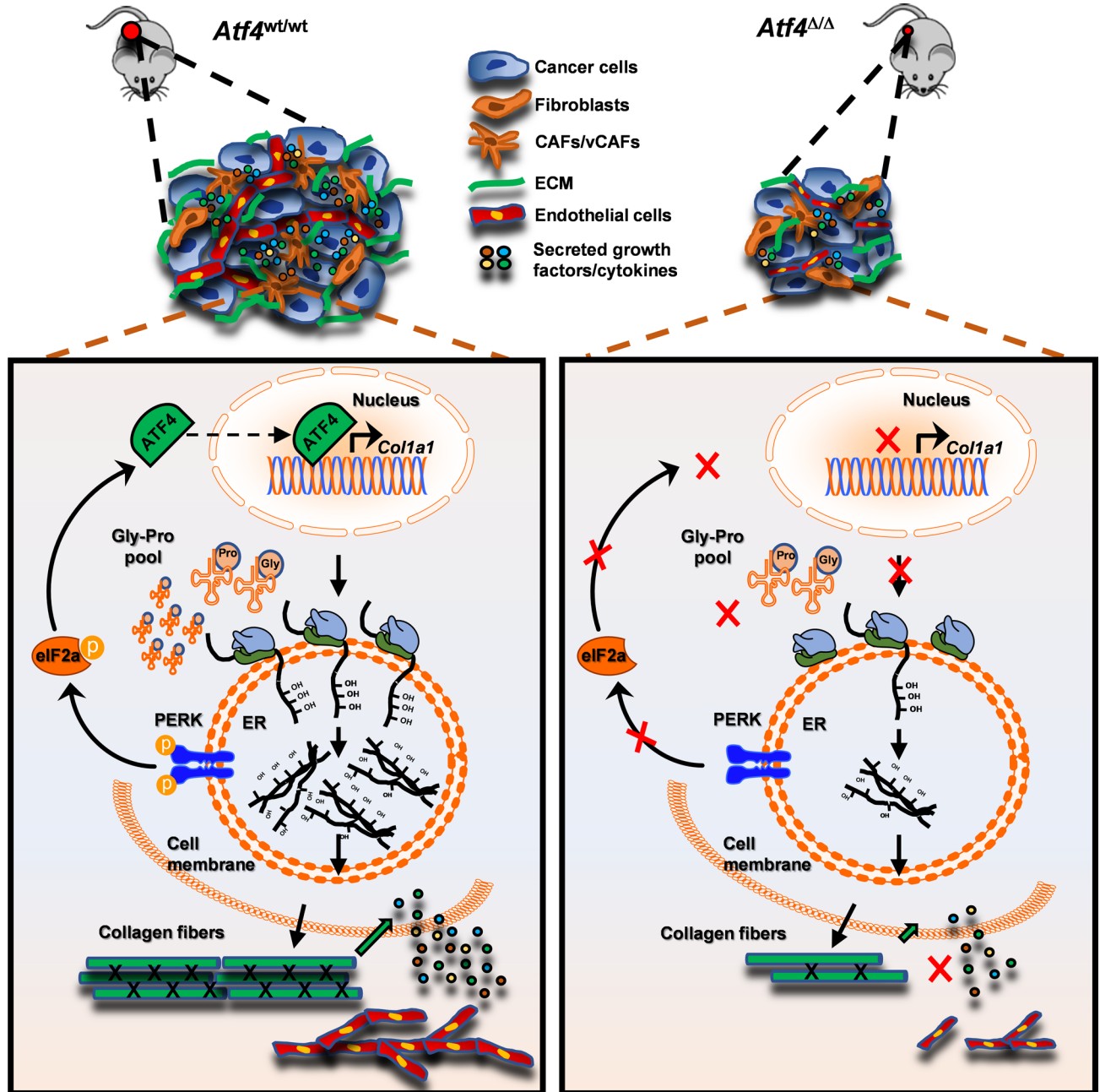

**Extended Data Fig. 10 | Proposed model.** Working model for host ATF4's role in tumour progression and metastasis. ATF4 is essential for the CAF activation via direct regulation of *Col1a1* expression and by impacting multiple additional steps in the collagen synthesis pathway, including Glycine (Gly) and Proline (Pro) pools. The resulting abrogation of Collagen I (and potentially additional collagen isoforms) in ATF4-deficient FBs leads in dramatic reduction in secreted extracellular matrix collagen, which in turn results in defective CAF activation and reduced levels of angiogenic cytokine signaling to endothelial cells. The resulting defective angiogenesis leads to reduced support for primary and metastatic tumour growth.

# Reporting Summary

## Statistics

For all statistical analyses, confirm that the following items are present in the figure legend, table legend, main text, or Methods section.

| n/a | Confirmed | |
|---|---|---|
| ☐ | ☒ | The exact sample size (*n*) for each experimental group/condition, given as a discrete number and unit of measurement |
| ☐ | ☒ | A statement on whether measurements were taken from distinct samples or whether the same sample was measured repeatedly |
| ☐ | ☒ | The statistical test(s) used AND whether they are one- or two-sided<br>*Only common tests should be described solely by name; describe more complex techniques in the Methods section.* |
| ☒ | ☐ | A description of all covariates tested |
| ☒ | ☐ | A description of any assumptions or corrections, such as tests of normality and adjustment for multiple comparisons |
| ☐ | ☒ | A full description of the statistical parameters including central tendency (e.g. means) or other basic estimates (e.g. regression coefficient) AND variation (e.g. standard deviation) or associated estimates of uncertainty (e.g. confidence intervals) |
| ☐ | ☒ | For null hypothesis testing, the test statistic (e.g. *F*, *t*, *r*) with confidence intervals, effect sizes, degrees of freedom and *P* value noted<br>*Give P values as exact values whenever suitable.* |
| ☒ | ☐ | For Bayesian analysis, information on the choice of priors and Markov chain Monte Carlo settings |
| ☒ | ☐ | For hierarchical and complex designs, identification of the appropriate level for tests and full reporting of outcomes |
| ☐ | ☒ | Estimates of effect sizes (e.g. Cohen's *d*, Pearson's *r*), indicating how they were calculated |

*Our web collection on statistics for biologists contains articles on many of the points above.*

## Software and code

Policy information about availability of computer code

| Data collection | QuantStudio 6 Flex, ChemiDoc, MAVEN, Cell Ranger V3.0.1, R package ggplot2, OncoLnc, BD FFACSCanto II, NIS Elements v.4.60.00, Zen 3.1 |
|---|---|
| Data analysis | Excel 365, GraphPad Prism version 8, Image J (FIJI), ChemiDoc, MAVEN, Cell Ranger V3.0.1, R package ggplot2, OncoLnc, RSAT, FlowJo v.10, NIS Elements v.4.60.00, Zen 3.1, FastQC, Cutadapt, Imaris 9.8.1 |

For manuscripts utilizing custom algorithms or software that are central to the research but not yet described in published literature, software must be made available to editors and reviewers. We strongly encourage code deposition in a community repository (e.g. GitHub). See the Nature Portfolio guidelines for submitting code & software for further information.

## Data

Policy information about availability of data

All manuscripts must include a data availability statement. This statement should provide the following information, where applicable:

- Accession codes, unique identifiers, or web links for publicly available datasets
- A description of any restrictions on data availability
- For clinical datasets or third party data, please ensure that the statement adheres to our policy

scRNA–seq and microarray data that support the findings of this study have been deposited in the Gene Expression Omnibus (GEO) under accession codes GSE159996 and GSE159020. The human skin cutaneous melanoma and pancreatic adenocarcinoma data were derived from Broad GDAC Firehose: https://gdac.broadinstitute.org/. Publicly available ChIP-Seq datasets used in this study were retrieved from GEO repository and can be found under the accession codes GSM873426 and GSM873427.
All other data and scripts supporting the findings of this study are available from the corresponding author on reasonable request.

# Field-specific reporting

Please select the one below that is the best fit for your research. If you are not sure, read the appropriate sections before making your selection.

☒ Life sciences　　　☐ Behavioural & social sciences　　　☐ Ecological, evolutionary & environmental sciences

For a reference copy of the document with all sections, see nature.com/documents/nr-reporting-summary-flat.pdf

# Life sciences study design

All studies must disclose on these points even when the disclosure is negative.

| | |
|---|---|
| Sample size | Statistical method was not used to predetermine sample size. We determined the sample size based on previous publications of our group and others and our experience. |
| Data exclusions | There was no inclusion/exclusion criteria for samples as no sample results were excluded |
| Replication | All experiments were performed at 1-3 times unless noted in the legend. All attempts at replication were successful. |
| Randomization | There was no randomization of the mice since post tamoxifen treatment we had to follow the excision of ATF4. No specific treatment was given to the mice so randomization was not possible. |
| Blinding | Tumor growth measurements in mice, quantification of immunofluorescence images and two photon microscopy, were blinded by use of coded subjects (mice). |

# Reporting for specific materials, systems and methods

We require information from authors about some types of materials, experimental systems and methods used in many studies. Here, indicate whether each material, system or method listed is relevant to your study. If you are not sure if a list item applies to your research, read the appropriate section before selecting a response.

## Materials & experimental systems

| n/a | Involved in the study |
|---|---|
| ☐ | ☒ Antibodies |
| ☐ | ☒ Eukaryotic cell lines |
| ☒ | ☐ Palaeontology and archaeology |
| ☐ | ☒ Animals and other organisms |
| ☐ | ☒ Human research participants |
| ☒ | ☐ Clinical data |
| ☒ | ☐ Dual use research of concern |

## Methods

| n/a | Involved in the study |
|---|---|
| ☒ | ☐ ChIP-seq |
| ☐ | ☒ Flow cytometry |
| ☒ | ☐ MRI-based neuroimaging |

## Antibodies

| | |
|---|---|
| Antibodies used | Rabbit monoclonal anti-ATF4 (Clone: D4B8, Cat#11815, RRID:AB_2616025), Rabbit monoclonal anti-PERK (Clone: C33E10, Cat#3192, RRID:AB_2095847), Rabbit monoclonal anti-p-eIF2a (Clone: D9G8, Cat#3398, RRID:AB_2096481), Rabbit polyclonal anti-eIF2a (Cat#9722, RRID:AB_2230924), Rabbit monoclonal anti-p-SMAD3 (Clone: C25A9, Cat#9520, RRID:AB_2193207), Rabbit monoclonal anti-SMAD2/3 (Clone: D7G7, Cat#8685, RRID:AB_10889933), Mouse monoclonal anti-b-actin (Clone: 8H10D10, Cat#3700, RRID:AB_2242334), Rabbit polyclonal anti-b-tubulin (Cat#2146, RRID:AB_2210545), Normal Rabbit IgG (Cat#2729S, RRID:AB_1031062) were purchased from Cell Signaling Technology. Rabbit polyclonal anti-Collagen Type I (Cat#AB765P, RRID:AB_92259), Mouse monoclonal anti-Acta2 (aSMA), Cy3 (Cat#C6198, RRID:AB_476856), Mouse monoclonal anti- Acta2, FITC (Clone: 1A4, Cat#F3777, RRID:AB_476977) were purchased from Millipore-Sigma. PE anti-mouse CD31 (Clone: MEC13.3, Cat#102508, RRID:AB_312915), Mouse monoclonal anti- CD34 (Clone: QBEnd/10, Cat#826401, RRID:AB_2564903) were purchased from BioLegend. Mouse monoclonal anti-PDGFRb (Clone: 42G12, Cat#ab69506, RRID:AB_1269704), Rabbit polyclonal anti-CD31 (Cat#ab28364, RRID:AB_726362), Rabbit polyclonal anti-ATF4 (Cat#ab31390) were purchased from Abcam. Rat monoclonal anti-NG2 (Clone: 546930, Cat#MA5-24247, AB_2606388), Mouse monoclonal anti-VEGF (Clone: JH121, Cat#MA5-13182, RRID:AB_10981661), Goat polyclonal anti-mouse IgG, Alexa Fluor 594 (Cat#A-11005, RRID:AB_141372), Donkey polyclonal anti-rabbit IgG, Alexa Fluor 488 (Cat#A-21206, RRID:AB_2535792), Donkey polyclonal anti-sheep IgG, Alexa Fluor 488 (Cat#A-11015, RRID:AB_2534082), Goat polyclonal anti-rat IgG, Alexa Fluor 488 (Cat#A-11006, RRID:AB_2534074), Donkey polyclonal anti-goat IgG, Alexa Fluor 488 (Cat#A-11055, RRID:AB_2534102), Goat polyclonal anti-Rabbit IgG, HRP (Cat#31460, RRID:AB_228341), Goat polyclonal anti-Mouse IgG, HRP (Cat# 31430, RRID:AB_228307) were purchased from ThermoFisher. Rat monoclonal anti-CD31 (Clone: MEC13.3, Cat#550274, RRID:AB_393571) and Mouse IgG2a, FITC (Cat#553456, RRID:AB_479604) were purchased from BD Biosciences. Sheep polyclonal anti-FAP (Cat#AF3715, RRID:AB_2102369), Mouse monoclonal CXCL12/SDF-1 (Clone: 79018, Cat#MAB-350, RRID:AB_2088149) were purchased from R&D systems. Goat anti-Collagen Type I (Cat#1310-01, RRID:AB_2753206) and Mouse IgG1 (Clone: 15H6, Cat#0102-01, RRID:AB_2793845) were purchased from Southern Biotech. Goat IgG (Cat#005-000-003, RRID:AB_2336985), Rabbit IgG (Cat#011-000-003, RRID:AB_2337118), Donkey polyclonal anti-goat IgG, HRP (Cat#705-035-147, |

RRID:AB_2313587) and Goat polyclonal anti-rabbit IgG, HRP (Cat#111-035-144, RRID:AB_2307391) were purchased from Jackson ImmunoResearch. Rabbit polyclonal anti-Ki67/MKI67 (Cat#NB500-170, RRID:AB_343263), Rabbit polyclonal anti-CD31 (Cat#NB100-2284, RRID:AB_10002513) were purchased from Novus. Antibodies were used for Western Blot, IF, IHC, ChIP, flow cytometry and multiple IF. Please refer to Supplementary Table 14 to check for their relevant application.

Validation | For most of the antibodies we validated using WT and knockout or knock down cells and different stimulations. Manufacturer's site also has validation for antibodies.

## Eukaryotic cell lines

Policy information about cell lines

Cell line source(s) | B16F10 were purchased from ATCC. MH6419 (6419c5) cells are a kind gift from Ben Stanger at The University of Pennsylvania. MH6419 cells originated from the KrasLSL-G12D/wt;Trp53fl/fl;Pdx1-Cre (KPC) model of spontaneous pancreatic cancer (PMID: 29958801).

Authentication | None of the cell lines were authenticated within 1 year. The MH6419 cell line was derived from tumor developed in a genetically modified mouse and has been characterized previously (PMID: 29958801).

Mycoplasma contamination | All cell lines were tested for mycoplasma and were found negative.

Commonly misidentified lines (See ICLAC register) | The cell lines used are not listed in the ICLAC register.

## Animals and other organisms

Policy information about studies involving animals; ARRIVE guidelines recommended for reporting animal research

Laboratory animals | Male and female mice 9-10 week old from the following genotypes (C57BL/6 background) were used in all in vivo experiments: Rosa26::CreERT2:Atf4wt/wt and Rosa26::CreERT2:Atf4fl/fl , Col1a1::CreERT2:Atf4wt/wt and Col1a1::CreERT2:Atf4fl/fl.

Wild animals | This study did not involve wild animals.

Field-collected samples | This study did not involve field collected samples.

Ethics oversight | All animal experiments have been approved by the University Laboratory Animal Resources (ULAR) and Institutional Animal Care and Use Committee (IACUC) of the University of Pennsylvania regulations.

Note that full information on the approval of the study protocol must also be provided in the manuscript.

## Human research participants

Policy information about studies involving human research participants

Population characteristics | Characteristics for malignant melanoma with normal skin tissue array: 90 males of 54.8±12.8 years old and 86 females of 53.3±12.9 years old (Supplementary Table 18) and pancreas cancer tissue array with adjacent normal pancreas tissue: 5 males of 59.4±9.6 years old and 5 females of 61.8±6.6 years old (Supplementary Table 19). Characteristics for the formalin-fixed paraffin-embedded human melanoma tumors are not available (Fig. 3k, Supplementary Table 20) since the samples were de-identified.

Recruitment | There is no available information for the recruitment of the patients.

Ethics oversight | Ethical considerations and protocols that are used in tissue collection of the tissue arrays (malignant melanoma and pancreatic cancer) are mentioned on the Biomax webpage (https://www.biomax.us/FAQs) under the FAQ10: "All tissue is collected under the highest ethical standards with the donor being informed completely and with their consent. We make sure we follow standard medical care and protect the donors' privacy. All human tissues are collected under HIPPA approved protocols. All samples have been tested negative for HIV and Hepatitis B or their counterparts in animals and approved for commercial product development".
Human melanoma tumor samples were obtained from patients resected at the Hospital of the University of Pennsylvania upon signing the informed consent in accordance with the Institutional Review Board (IRB) protocol No. 703001. There is no other information available for these patients since the samples were de-identified.

Note that full information on the approval of the study protocol must also be provided in the manuscript.

# Flow Cytometry

## Plots

Confirm that:

☒ The axis labels state the marker and fluorochrome used (e.g. CD4-FITC).

☒ The axis scales are clearly visible. Include numbers along axes only for bottom left plot of group (a 'group' is an analysis of identical markers).

☒ All plots are contour plots with outliers or pseudocolor plots.

☒ A numerical value for number of cells or percentage (with statistics) is provided.

## Methodology

| | |
|---|---|
| Sample preparation | Primary cultures of ATF4wt/wt lung endothelial cells and single cell suspension from B16F10 tumors grown in ATF4 WT and ATF4 KO mice. |
| Instrument | All data acquisition was done using a FACSCanto II (BD Biosciences). |
| Software | BD FACSCanto II used to collect the data and FlowJo v.10 to analyze them. |
| Cell population abundance | Lung endothelial cells were above 80% pure post beads isolation (positive selection) and determined by flow cytometry (CD31-PE+ cells). B16F10 tumors were analyzed for live/dead cells (AmCyan). |
| Gating strategy | For endothelial cells: Forward scatter (FSC) versus Side scatter (SSC) gating were used for the selection of population. CD31-PE versus Count gating were used to show the positivity for CD31.<br>For tumor cells: Forward scatter (FSC) versus Side scatter (SSC) gating were used for the selection of population. FSC-A versus FSC-H gating were used for the selection of single cells only. AmCyan-A versus SSC-A gating were used to determine the live/dead cells. |

☒ Tick this box to confirm that a figure exemplifying the gating strategy is provided in the Supplementary Information.

