## [Peer Review File · Nature Cell Biology]

Peer Review Information

Journal: Nature Cell Biology

Manuscript Title: A Stromal Integrated Stress Response Activates Perivascular Cancer-Associated Fibroblasts to Drive Angiogenesis and Tumor Progression

Corresponding author name(s): Dr Constantinos Koumenis

Reviewer Comments & Decisions:

Decision Letter, initial version:
--

Subject: Decision on Nature Cell Biology submission NCB-K45380

Message:

*Please delete the link to your author homepage if you wish to forward this email to co-authors.

Dear Dr Koumenis,

Your manuscript, "A Stromal Integrated Stress Response Activates Perivascular Cancer-Associated Fibroblasts to Drive Angiogenesis and Tumor Progression", has now been seen by 3 referees, who are experts in tumor angiogenesis (referee 1); stress response (referee 2); and tumor stroma (referee 3). As you will see from their comments (attached below) they find this work of potential interest, but have raised substantial concerns, which in our view would need to be addressed with considerable revisions before we can consider publication in Nature Cell Biology.

Nature Cell Biology editors discuss the referee reports in detail within the editorial team, including the chief editor, to identify key referee points that should be addressed with priority, and requests that are overruled as being beyond the scope of the current study. To guide the scope of the revisions, I have listed these points below. We are committed to providing a fair and constructive peer-review process, so please feel free to contact me if you would like to discuss any of the referee comments further.

In particular, it would be essential to:

A) Strengthen the evidence for the proposed role of CAFs as questioned by Reviewer 1:

"The authors should clearly demonstrate that the phenotype is a result of loss of Atf4 in fibroblasts and not tumor endothelial cells. Their observations in Col1a1-CreErt2 mice are encouraging, however it will be important to rule a potential ectopic activity of Cre in tumor vasculature, for example by RT-qPCR analysis of sorted tumor endothelial cells."

"The authors suggest that ATF4 main target cell type are perivascular CAFs – how are they different from vSMCs and pericytes? In human tissues "ATF4 is highly expressed (apart from the tumor cells) on CAFs (aSMA) that localized on the perivascular area (CD34) compared to other CAFs reside away from the blood vessels (Fig. 3i)." – however such perivascular ATF4 could as well be vSMCs, ATF4 expression and role in vSMCs has been reported."

"The authors propose that unlike wild type fibroblasts, Atf4 deficient fibroblasts do not support growth of B16F10 cells. It would be important to provide evidence that Atf4 deficient fibroblast survive upon such co-injection, it is possible that they are simply do not survive s.c. transplantation."

B) Provide further in vivo data as requested by reviewers:

Reviewer 1

"The authors show that one of the main effects of Atf4 deficiency is reduced tumor angiogenesis, and propose, based on their in vitro results, that this is a result of reduced production of pro-angiogenic factors, such as VEGF and SDF1, by CAFs. There is a missing link with in vivo observations – do Atf4 deficient fibroblasts have reduced levels of Vegfa mRNA or they display reduced Vegfa secretion and deposition? Do the authors observe reduced production of VEGF&other factors in vivo? If the main idea is that CAF secretome is affected, it would be important to confirm the results in independent experiments using Limunex or Elisa of tumor lysates but also for in vitro data, right now it looks like n=1 data."

Reviewer 3

"Authors show throughout the manuscript that ATF4 plays a remarkable role as a driver of collagen expression. In figure S4f and S4g, the authors show that vasculature has reduced permeability when host ATF4 is deleted. However, it is not clear whether the reduced permeability in tumor vasculature is due to reduced collagen production by ATF4 deficient perivascular CAFs or due to reduction of other ATF4-regulated factors."

"Considering the reduced number of metastases and the defective vasculature in tumors grown in ATF4 ko mice, it would be interesting to know whether the lack of host ATF4 affects intravasation/extravasation of tumor cells into the vasculature."

"They show that tumor growth is slowed in ATF ko mice after tail vein injection. This could be due to reduced growth as the authors suggest and/or reduced seeding into the lung. To address this they should delete ATF4 AFTER tail vein injection."

"The authors use the MH6419 pancreatic cancer cell line as a complementary model. However, MH6419 cells are injected subcutaneously and not orthotopically. It is well known that the tumor microenvironment can change drastically depending on the tumor site. High stromal content is a remarkable feature of pancreatic cancer, hence, it would increase the biological relevance if the MH6419 cells were orthotopically implanted. Without this the addition of the MH6419 doesn't add much to the manuscript and it would be better to supply an additional melanoma cell line."

"The authors argue that the lack of host ATF4 affects tumor initiation and tumor progression. The role of ATF4 in tumor progression is well addressed by using tumor cell implantation models. However, to better understand the role of host ATF4, more specific ATF4-expressing fibroblasts, in tumor initiation, it would be interesting to see how ATF4 knockout in fibroblasts (Col1a1-cre) would affect tumor generation in a GEMM model."

C) Improve analysis of the TME and tumors as asked by Reviewer 1:

"The authors should provide a more thorough analysis of the TME and tumors, including the analysis of mural coverage of vessels and proliferation and apoptosis of endothelial, CAFs and tumor cells."

D) All other referee concerns pertaining to strengthening existing data, providing controls, methodological details, clarifications and textual changes as applicable should also be addressed.

E) Finally please pay close attention to our guidelines on statistical and methodological reporting (listed below) as failure to do so may delay the reconsideration of the revised manuscript. In particular please provide:

We would be happy to consider a revised manuscript that would satisfactorily address these points, unless a similar paper is published elsewhere, or is accepted for publication in Nature Cell Biology in the meantime.

- ensure that it conforms to our format instructions and publication policies (see below and <https://www.nature.com/nature/for-authors>).
- provide a point-by-point rebuttal to the full referee reports verbatim, as provided at the end of this letter.
- provide the completed Reporting Summary (found here <https://www.nature.com/documents/nr-reporting-summary.pdf>). This is essential for reconsideration of the manuscript will be available to editors and referees in the event of peer review. For more information see <http://www.nature.com/authors/policies/availability.html> or contact me.

When submitting the revised version of your manuscript, please pay close attention to our [href="https://www.nature.com/nature-research/editorial-policies/image-integrity">Digital Image Integrity Guidelines](https://www.nature.com/nature-research/editorial-policies/image-integrity). and to the following points below:

Nature Cell Biology is committed to improving transparency in authorship. As part of our efforts in this direction, we are now requesting that all authors identified as 'corresponding author' on published papers create and link their Open Researcher and Contributor Identifier (ORCID) with their account on the Manuscript Tracking System (MTS), prior to acceptance. ORCID helps the scientific community achieve unambiguous attribution of all scholarly contributions. You can create and link your ORCID from the home page of the MTS by clicking on 'Modify my Springer Nature account'. For more information please visit www.springernature.com/orcid.

This journal strongly supports public availability of data. Please place the data used in your paper into a public data repository, or alternatively, present the data as Supplementary Information. If data can only be shared on request, please explain why in your Data Availability Statement, and also in the correspondence with your editor. Please note that for some data types, deposition in a public repository is mandatory - more information on our data deposition policies and available repositories appears below.

[REDACTED]

We would like to receive a revised submission within six months.

We hope that you will find our referees' comments, and editorial guidance helpful. Please do not hesitate to contact me if there is anything you would like to discuss.

Best wishes,
Zhe Wang

Zhe Wang, PhD
Senior Editor
Nature Cell Biology

Tel: +44 (0) 207 843 4924
email: zhe.wang@nature.com

Reviewers' Comments:

Reviewer #1:
Remarks to the Author:

The manuscript by Verginadis and colleagues investigates the role of stromal ATF4 in angiogenesis and tumor progression. Using mice with inducible global ATF4 inactivation the authors show that deletion of ATF4 delays growth, vascularization and ECM deposition in subcutaneously implanted B16F10 and MH6419 tumors. The effects of global ATF4 knockout were recapitulated in Col1a1-CreERT2;Atf4flox/flox mice in B16F10 model, suggesting that fibroblasts represent a major cell type targeted by Atf4 deficiency. Through scRNA seq analysis of small and large tumors B16F10, they report that ATF4 deletion mostly affects cancer-associated fibroblasts, leading to decreased expression of collagen 1 subunits, Pdgfrb and Acta2 in small tumors whereas the expression of collagen 1 is regained in large tumor, indicating the existence of compensatory mechanisms. The authors propose that a subset of CAFs with high levels of Pdgfrb and Acta2 represent (peri)vascular fibroblasts and that this cell type is most affected by lack of Atf4. Based on the in vitro studies of dermal Atg4 wt and knockout fibroblasts, they show that loss of Atf4 leads to decreased collagen I production and propose that this is a result of combined effect on both direct binding of Atf4 to collagen 1a1 promoter and on intracellular fluxes of glycine and proline. In another set of in vitro experiments, the authors show that conditional medium from ATF4 KO fibroblasts, which had reduced levels of angiogenic growth factor Vegfa and Sdf1, does not support endothelial tube formation in Matrigel unlike the conditioned medium from ATF4-expressing fibroblasts. Atf4 deficiency also reduced lung colonization by B16F10 cells and, especially, B16F10 metastasis after primary tumor resection. In human melanoma Col1a1 and ATF4 were also positively co-related and associated with poor prognosis. The authors thus propose that ATF4-expressing CAFs play a considerable role in creating a tumor-favorable TME by inducing angiogenesis and secreting ECM components such as collagen I.

Overall, it is a thorough and well written work, which establishes an important role of ATF4 in CAFs and tumor growth using a diverse array of models and tools. However, there are some important points that need to be addressed.

General comments

1. The authors should clearly demonstrate that the phenotype is a result of loss of Atf4 in fibroblasts and not tumor endothelial cells. Their observations in Col1a1-CreErt2 mice are encouraging, however it will be important to rule a potential ectopic activity of Cre in tumor vasculature, for example by RT-qPCR analysis of sorted tumor endothelial cells.
2. The authors show that one of the main effects of Atf4 deficiency is reduced tumor angiogenesis, and propose, based on their in vitro results, that this is a result of reduced production of pro-angiogenic factors, such as VEGF and Sdf1, by CAFs. There is a missing link with in vivo observations – do Atf4 deficient fibroblasts have reduced levels of Vegfa mRNA or they display reduced Vegfa secretion and deposition? Do the authors observe reduced production of VEGF&other factors in vivo? If the main idea is that CAF secretome is affected, it would be important to confirm the results in independent experiments using Limunex or Elisa of tumor lysates but also for in vitro data, right now it looks like n=1 data.
3. Related to 1, Figure 5 has several issues:
 - Endothelial cells seems to be in poor shape/dying (Fig 5b)

- no positive control is included
- use of Matrigel tube forming assay is a suboptimal way for modeling angiogenesis in vitro (PMID: 25931450), the authors should consider alternatives, such as endothelial spheroid sprouting assay.
- 4. The IHC analysis in Figure 3 shows loss of larger aSMA-covered tumor vessels, but not microvessels – such effect is opposite from what is expected upon loss of tumor angiogenesis signaling and indicates a more complex underlying mechanism than the one proposed by the authors. Analysis of tumor thick sections will be helpful to find out whether loss of Atf4 rather leads to increased tumor hypersprouting coupled with reduced lumen formation as observed upon inhibition of Notch signaling in other tumor models, such as in Dll4 +/- mice. This also leads to reduced tumor growth and perfusion. It may be also interesting to check whether there is loss of secretion/production of Notch ligands, such as Dll4 in endothelial cells.
- 5. The authors should provide a more thorough analysis of the TME and tumors, including the analysis of mural coverage of vessels and proliferation and apoptosis of endothelial, CAFs and tumor cells.
- 6. The authors suggest that ATF4 main target cell type are perivascular CAFs – how are they different from vSMCs and pericytes? In human tissues “ATF4 is highly expressed (apart from the tumor cells) on CAFs (aSMA) that localized on the perivascular area (CD34) compared to other CAFs reside away from the blood vessels (Fig. 3i).” – however such perivascular ATF4 could as well be vSMCs, ATF4 expression and role in vSMCs has been reported.
- 7. The authors propose that unlike wild type fibroblasts, Atf4 deficient fibroblasts do not support growth of B16F10 cells. It would be important to provide evidence that Atf4 deficient fibroblast survive upon such co-injection, it is possible that they are simply do not survive s.c. transplantation.
- 8. The authors should discuss in more detail potential mechanisms underlying a tumor escape from Atf4 inhibition.

Technical comments:

1. The authors state that they use orthotopic B16F10 model, however this is not the case as they do s.c. and not intradermal injections.
2. The authors mention that they identified a new population of melCAFs, with mixed melanoma and CAF identities - unusual “mixed” populations can be observed in scRNAseq data due to RNA transfer/uptake during sample preparation, especially in physically interacting cells. Please confirm out that this is not a sample preparation artifact by other means , e.g, costaining, or amend the analysis.
3. The authors sometimes use very small number of tumors for the analysis, please consider increasing the number of samples to make sure that the reported observation are indeed solid.

Minor comments

1. CD31 is a pan-endothelial marker and not specific for tumor endothelium
2. Please mention the percentage of CAFs found in large B16F10 tumors like in small tumors.
3. Regarding vessel permeability experiment, please comment on whether this is due to intrinsic (check endothelial gene signature from single cell RNA-seq) or extrinsic effect on endothelial cells.

4. supp Figure 4d, the merged figure look fuzzy -please replace
5. Supp 6d-e it would be easier to keep the naming of tumor and not use FB.
6. Fig 4b should be in supplemental data because it is literature and not a result of the paper.
7. Fig 4i please mention what the dots represent
8. Fig 5e mention the color code and the proteins in the legend

Reviewer #2:

Remarks to the Author:

This manuscript addresses the important role of ATF4 and the Integrated stress response (ISR) in the tumor environment during cancer progression. While ATF4 has previously been implicated in promoting tumor growth directly, the role of ATF4 in tumor stroma has not been previously investigated. The manuscript combined ATF4 knock-out mice in subcutaneous melanoma (orthotopic) and pancreatic mouse models of cancer with single cell transcriptome analyses and determined that a population of cancer-associated fibroblasts (termed vCAFs) require ATF4 to support neo-angiogenesis, vascularization, and collagen (ECM) expression to promote efficient tumor growth and metastasis. The key conclusion of the study is that ATF4 is a major driver of perivascular CAF function and plays a critical role in shaping the tumorigenic microenvironment. The animal experiments are elegantly designed and well-controlled, and the results support the major conclusions of the manuscript. The manuscript is clearly written and rigorous. The study adds significantly to our knowledge of the role of the ISR in the tumor microenvironment, and specifically implicates the ISR and ATF4 as critical regulators of CAF function, an important cell type implicated in the progression, growth, and metastasis of tumors. This manuscript has important implications for new cancer therapies, a point that could be expanded on further in the discussion. This reviewer is very enthusiastic about the manuscript. There are only minor concerns, which are listed below.

Minor concerns:

Line 177-178: The numbers of transcriptome analyses of small and large tumors could be more clearly explained in context of the WT and Atf4-delta mice.

Fig 2C and related figures: The x-axis is not clear here and should be labeled.

Line 267: This sentence was not clear: what was primarily restricted in the blood vessels?

Line 331: The uptake vs metabolic flux defect associated with Atf4 was not fully clear. The manuscript notes that uptake of M+3 serine and M+7 glutamine was similar in WT versus Atf4-deficient cells, suggesting that transport of serine and glutamine was not affected by ATF4 loss. Is this data shown in

the manuscript? Is the expression of genes involved in glycine or proline biosynthesis impacted in ATF4-deficient lung fibroblasts? What is the proposed mechanism for reduced biosynthesis of glycine and proline in ATF4-deficient cells?

Line 344: Secretory stress caused by high collagen production is proposed to cause activation of the eIF2 kinase PERK, resulting in increased expression of Atf4 (Fig. 4K). Interestingly, silencing of Col1a1 reduced PERK activation but led to an increase in ATF4 expression in the absence of TGF- β 1. Is there a change in eIF2 phosphorylation in Col1a1-silenced fibroblasts? Does this contribute to the secretory defect in ATF4-deficient fibroblasts?

Line 526: This sentence was not fully clear.

Discussion: There was no significant discussion of which eIF2 kinase may be the most prominent in driving CAF function. The manuscript mentions a GCN2 signature in ATF4-deficient CAFs (Fig. 2D) and demonstrates PERK activation in wild-type lung fibroblasts (Fig. 4K), but does not sufficiently revisit these points in the discussion. These would seem to be worthwhile ideas to discuss in the context of therapeutic strategies.

Reviewer #3:

Remarks to the Author:

In the present manuscript, Verginadis et al., addresses how the integrated stress response, which culminates in ATF4 activation in cancer-associated fibroblasts (CAFs) drives angiogenesis and collagen production to support tumor progression in melanoma and pancreatic cancer. This is a timely manuscript with novel data, however some points should be addressed. Please see comments below, not necessarily in order of importance.

1- Authors show throughout the manuscript that ATF4 plays a remarkable role as a driver of collagen expression. In figure S4f and S4g, the authors show that vasculature has reduced permeability when host ATF4 is deleted. However, it is not clear whether the reduced permeability in tumor vasculature is due to reduced collagen production by ATF4 deficient perivascular CAFs or due to reduction of other ATF4-regulated factors.

2- Considering the reduced number of metastases and the defective vasculature in tumors grown in ATF4 ko mice, it would be interesting to know whether the lack of host ATF4 affects intravasation/extravasation of tumor cells into the vasculature.

3- They show that tumor growth is slowed in ATF ko mice after tail vein injection. This could be due to reduced growth as the authors suggest and/or reduced seeding into the lung. To address this they should delete ATF4 AFTER tail vein injection.

- 4- The authors use the MH6419 pancreatic cancer cell line as a complementary model. However, MH6419 cells are injected subcutaneously and not orthotopically. It is well known that the tumor microenvironment can change drastically depending on the tumor site. High stromal content is a remarkable feature of pancreatic cancer, hence, it would increase the biological relevance if the MH6419 cells were orthotopically implanted. Without this the addition of the MH6419 doesn't add much to the manuscript and it would be better to supply an additional melanoma cell line.
- 5- The authors argue that the lack of host ATF4 affects tumor initiation and tumor progression. The role of ATF4 in tumor progression is well addressed by using tumor cell implantation models. However, to better understand the role of host ATF4, more specific ATF4-expressing fibroblasts, in tumor initiation, it would be interesting to see how ATF4 knockout in fibroblasts (Col1a1-cre) would affect tumor generation in a GEMM model.
- 6- The authors claim one of their CAF subpopulations consists of melCAFs, which retain some "traits" from melanoma cells. What does this mean and how do they rule out the possibility that eCAFs are in fact, melanoma cells?
- 7- The authors showed that larger tumors still have defective vascularization however collagen levels are not altered at this stage. Considering the perivascular localization of ATF4-expressing CAFs, are the reduced levels of collagen observed specific to the perivascular niche? CD31 (or other endothelial marker) costaining would help to identify whether collagen production is defective in perivascular areas of large tumors or whether collagen levels were normalized by an alternative pathway.
- 8- In line 286/287, the authors quote "DFBwt/wt injected into ATF4^{-/-} mice nearly completely reverse tumor growth inhibition observed in ATF4^{-/-} + DFB^{-/-} group" (fig 3k). Despite the fact that there is a clear reversal of the tumor growth inhibition by the lack of ATF4 in DFBs, as pointed by the authors and supported by this and other results, the role of ATF4 in other cell types within the tumor microenvironment can't be ruled out. This quote sounds like an overstatement.
- 9- In Fig. 4k, is TGF-beta1 stimulation supposed to increase collagen production? If so, the western blot picture does not support this conclusion. Further, the western would be strengthened by adding LFB ko cells treated with TGF-beta1, which should have high PERK levels regardless of TGF-beta1 treatment.
- 10- In Fig. 1b and other similar figures, it would increase clarity if the authors include in the Y axis the name of the gene they are showing mRNA expression levels.
- 11- The legend for Sup Fig. 2c and 3j needs to clarify what is being shown. The X axis states it is normalized...(WT/KO) but there is a bar for WT and for KO, should there not only be one bar given it is a ratio?? Please clarify.
- 12- Fig. 3i, they need to show more staining from more than one person. No way to assess if this is representative.
- 13- Fig. 3c/d and e/f show impressive loss of FAP and SMA in ATF ko tumors. Does this suggest that loss of ATF impacts all CAFs, not only vCAFs? This needs to be clarified/
- 14- Fig 7c shows Col1 and ATF staining on human TMAs. The staining in the stromal compartment versus tumor cells should be quantitated to establish that ATF4 is impacted in human CAFs.

Methods should be written concisely, but should contain all elements necessary to allow interpretation and replication of the results. As a guideline, Methods sections typically do not exceed 3,000 words. The Methods should be divided into subsections listing reagents and techniques. When citing previous methods, accurate references should be provided and any alterations should be noted. Information must be provided about: antibody dilutions, company names, catalogue numbers and clone numbers for monoclonal antibodies; sequences of RNAi and cDNA probes/primers or company names and catalogue numbers if reagents are commercial; cell line names, sources and information on cell line identity and authentication. Animal studies and experiments involving human subjects must be reported in detail, identifying the committees approving the protocols. For studies involving human subjects/samples, a statement must be included confirming that informed consent was obtained. Statistical analyses and

information on the reproducibility of experimental results should be provided in a section titled "Statistics and Reproducibility".

All Nature Cell Biology manuscripts submitted on or after March 21 2016 must include a Data availability statement as a separate section after Methods but before references, under the heading "Data Availability". For Springer Nature policies on data availability see <http://www.nature.com/authors/policies/availability.html>; for more information on this particular policy see <http://www.nature.com/authors/policies/data/data-availability-statements-data-citations.pdf>. The Data availability statement should include:

- Accession codes for primary datasets (generated during the study under consideration and designated as "primary accessions") and secondary datasets (published datasets reanalysed during the study under consideration, designated as "referenced accessions"). For primary accessions data should be made public to coincide with publication of the manuscript. A list of data types for which submission to community-endorsed public repositories is mandated (including sequence, structure, microarray, deep sequencing data) can be found here <http://www.nature.com/authors/policies/availability.html#data>.
- Unique identifiers (accession codes, DOIs or other unique persistent identifier) and hyperlinks for datasets deposited in an approved repository, but for which data deposition is not mandated (see here for details <http://www.nature.com/sdata/data-policies/repositories>).
- At a minimum, please include a statement confirming that all relevant data are available from the authors, and/or are included with the manuscript (e.g. as source data or supplementary information), listing which data are included (e.g. by figure panels and data types) and mentioning any restrictions on availability.
- If a dataset has a Digital Object Identifier (DOI) as its unique identifier, we strongly encourage including this in the Reference list and citing the dataset in the Methods.

We recommend that you upload the step-by-step protocols used in this manuscript to the Protocol Exchange. More details can found at www.nature.com/protocolexchange/about.

FIGURES – Colour figure publication costs \$600 for the first, and \$300 for each subsequent colour figure. All panels of a multi-panel figure must be logically connected and arranged as they would appear in the

final version. Unnecessary figures and figure panels should be avoided (e.g. data presented in small tables could be stated briefly in the text instead).

All imaging data should be accompanied by scale bars, which should be defined in the legend. Cropped images of gels/blots are acceptable, but need to be accompanied by size markers, and to retain visible background signal within the linear range (i.e. should not be saturated). The boundaries of panels with low background have to be demarked with black lines. Splicing of panels should only be considered if unavoidable, and must be clearly marked on the figure, and noted in the legend with a statement on whether the samples were obtained and processed simultaneously. Quantitative comparisons between samples on different gels/blots are discouraged; if this is unavoidable, it should only be performed for samples derived from the same experiment with gels/blots were processed in parallel, which needs to be stated in the legend.

- For line art, graphs, charts and schematics we prefer Adobe Illustrator (.AI), Encapsulated PostScript (.EPS) or Portable Document Format (.PDF). Files should be saved or exported as such directly from the application in which they were made, to allow us to restyle them according to our journal house style.
- We accept PowerPoint (.PPT) files if they are fully editable. However, please refrain from adding PowerPoint graphical effects to objects, as this results in them outputting poor quality raster art. Text used for PowerPoint figures should be Helvetica (preferred) or Arial.
- We do not recommend using Adobe Photoshop for designing figures, but we can accept Photoshop generated (.PSD or .TIFF) files only if each element included in the figure (text, labels, pictures, graphs, arrows and scale bars) are on separate layers. All text should be editable in 'type layers' and line-art such as graphs and other simple schematics should be preserved and embedded within 'vector smart objects' - not flattened raster/bitmap graphics.

The total number of Supplementary Figures (not including the “unprocessed scans” Supplementary Figure) should not exceed the number of main display items (figures and/or tables (see our Guide to Authors and March 2012 editorial <http://www.nature.com/ncb/authors/submit/index.html#supinfo>; <http://www.nature.com/ncb/journal/v14/n3/index.html#ed>). No restrictions apply to Supplementary Tables or Videos, but we advise authors to be selective in including supplemental data.

GUIDELINES FOR EXPERIMENTAL AND STATISTICAL REPORTING

REPORTING REQUIREMENTS – We are trying to improve the quality of methods and statistics reporting in our papers. To that end, we are now asking authors to complete a reporting summary that collects information on experimental design and reagents. The Reporting Summary can be found here <https://www.nature.com/documents/nr-reporting-summary.pdf> If you would like to reference the guidance text as you complete the template, please access these flattened versions at <http://www.nature.com/authors/policies/availability.html>.

STATISTICS – Wherever statistics have been derived the legend needs to provide the n number (i.e. the sample size used to derive statistics) as a precise value (not a range), and define what this value represents. Error bars need to be defined in the legends (e.g. SD, SEM) together with a measure of centre (e.g. mean, median). Box plots need to be defined in terms of minima, maxima, centre, and percentiles. Ranges are more appropriate than standard errors for small data sets. Wherever statistical significance has been derived, precise p values need to be provided and the statistical test used needs to be stated in the legend. Statistics such as error bars must not be derived from $n < 3$. For sample sizes of $n < 5$ please plot the individual data points rather than providing bar graphs. Deriving statistics from technical replicate samples, rather than biological replicates is strongly discouraged. Wherever statistical

significance has been derived, precise p values need to be provided and the statistical test stated in the legend.

Author Rebuttal to Initial comments

Reviewer #1:

General comments

1. The authors should clearly demonstrate that the phenotype is a result of loss of *Atf4* in fibroblasts and not tumor endothelial cells. Their observations in *Col1a1-CreERT2* mice are encouraging, however it will be important to rule a potential ectopic activity of Cre in tumor vasculature, for example by RT-qPCR analysis of sorted tumor endothelial cells.

We thank the reviewer for bringing out this important point. Ectopic activity in Cre recombinase has been previously reported¹. In order to test this scenario, we isolated tumor endothelial cells (positive selection using magnetic beads) from equal volume B16F10 tumors grown in *Col1a1;ATF4^{wt/wt}* (TEC^{wt/wt}) and *Col1a1;ATF4^{Δ/Δ}* (TEC^{Δ/Δ}) mice and carried out RT-qPCR for *Atf4*. To test the purity of isolated tumor endothelial cells, we included samples of LFB^{wt/wt} and LFB^{Δ/Δ} and additional markers, such as *Pecam-1* (specific marker of endothelial cells) and *Col1a1* and *Col1a2* (specific markers of fibroblasts). Indeed, the isolated TECs were characterized by high levels of *Pecam1* expression while exhibiting comparatively very low levels of *Col1a1* and *Col1a2*. More importantly, we found no significant difference in the expression of *Atf4* between TEC^{wt/wt}, TEC^{Δ/Δ} cells and LFB^{wt/wt}, indicating that there is no significant ectopic activity of Cre recombinase in tumor vasculature and that *Atf4* is specifically excised in the fibroblasts. The RT-qPCR graph has been added as **Supplementary Fig. 6i**.

2. The authors show that one of the main effects of *Atf4* deficiency is reduced tumor angiogenesis, and propose, based on their *in vitro* results, that this is a result of reduced production of pro-angiogenic factors, such as VEGF and SDF1, by CAFs. There is a missing link with *in vivo* observations – do *Atf4* deficient fibroblasts have reduced levels of *Vegfa* mRNA or they display reduced VEGfa secretion and deposition? Do the authors observe reduced production of VEGF&other factors *in vivo*? If the main idea is that CAF secretome is affected, it would be important to confirm the results in independent experiments using Limunex or Elisa of tumor lysates but also for *in vitro* data, right now it looks like n=1 data.

Thank you for bringing up this important point. We do believe that the *Atf4* deficient fibroblasts display reduced VEGF secretion and deposition. This notion is supported by the following observations: (a) we went back to our scRNA-seq analysis of melanoma tumors as well as microarray analysis of LFBs and whole lungs and we did not observe any significant difference in the *Vegf* mRNA expression levels in *Atf4^{wt/wt}* and *Atf4^{Δ/Δ}* cells or tissues, (b) we have

repeated the angiogenesis array using conditioned medium (CM) from LFB^{wt/wt}, LFB^{Δ/Δ} and LFB^{Δ/Δ}+AdmATF4 (see Replicate 2) and similar to the first experiment, we confirm that the levels of both VEGF and SDF-1 are significantly reduced in CM of LFB^{Δ/Δ} cells compared to the LFB^{wt/wt} cells. Re-expression of mATF4 in LFB^{Δ/Δ} cells nearly restored the levels of these cytokines (see also quantification graph from 2 biological replicates). The quantification graph has been updated (Fig. 5d).

To further probe the significance of this differential secretion of cytokines *in vivo*, we run the same angiogenesis array on tumor lysates from equal volume B16F10 tumors harvested from two *Atf4*^{wt/wt} and two *Atf4*^{Δ/Δ} mice. We found no significant difference in the VEGF and SDF-1 secreted levels between the tumors of different ATF4 host status. This may happen because of tumor heterogeneity and the fact that the fibroblasts are not the only source of the VEGF and SDF-1 secretion into the tumor microenvironment²⁻⁹. However, we surmised that even though **total** VEGF and SDF-1 levels may not be different, **local** levels in the vicinity of growing blood vessels may be different. Therefore, we analyzed the levels of these angiogenic factors in tumors from *Atf4*^{wt/wt} and *Atf4*^{Δ/Δ} mice and calculated the % colocalization between VEGF and SDF-1 with CD31 positive staining. Intriguingly, the levels of both angiogenic factors were significantly lower in the perivascular areas from tumors grown in *Atf4*^{Δ/Δ} mice. This finding further supports our model showing that ATF4-deficient perivascular CAFs present a defective source of secreted angiogenic factors leading to an abnormal angiogenesis and significant attenuation of tumor growth. Representative images from these experiments have been added as new Fig. 5e,f (for VEGF) and Fig. 5g,h (for SDF-1). The quantification graphs of total % VEGF and % SDF-1 positive areas have been added as Supplementary Fig. 8d and e, respectively. The methods and the antibodies list have been updated accordingly.

3. Related to 1, Figure 5 has several issues:

- Endothelial cells seems to be in poor shape/dying (Fig 5b)
- no positive control is included
- use of Matrigel tube forming assay is a suboptimal way for modeling angiogenesis *in vitro* (PMID: 25931450), the authors should consider alternatives, such as endothelial spheroid sprouting assay.

We agree with the reviewer's assessment. We have tweaked the conditions of the *in vitro* tube formation assay so that the EC^{wt/wt} cells were treated with CM from LFB^{wt/wt} and LFB^{Δ/Δ} cells for 24h (instead of 4h, as we have previously done). This longer incubation time allowed the ECs to be fully stimulated by the angiogenic factors that exist in the CM and as a result we have obtained better quality images. We found that the CM:LFB^{Δ/Δ}-treated cells presented significantly reduced tube formation and sprouting compared to the CM:LFB^{wt/wt}-treated cells. Human brain ECs were used as controls (as requested by the reviewer). Also, the quantification analysis was performed with ImageJ software using a code for the morphometric analysis of "Endothelial Tube Formation Assay"¹⁰. The representative images and the quantification graphs for the "number of tubes" and "number of junctions" have been added as **Fig. 5a,b**. The quantification graphs of "the number of branches" and "total branching length" have been added as **Supplementary Fig. 8c**. The methods and the manuscript have been updated accordingly.

The tube formation assay is a well-established and commonly accepted assay in the angiogenesis field. We agree with the reviewer that alternative assays such as endothelial spheroid sprouting assay would be informative as well. However, due to **(a)** the significant differences we observed in the tube formation assay and **(b)** further optimization needed (cell concentration, time points, etc.) for other protocols to be tested which also require many mice to be euthanized for the isolation of endothelial cells, we believe that the generated data from the tube formation assay strongly support our hypothesis.

4. The IHC analysis in Figure 3 shows loss of larger α SMA-covered tumor vessels, but not microvessels – such effect is opposite from what is expected upon loss of tumor angiogenesis signaling and indicates a more complex underlying mechanism than the one proposed by the authors. Analysis of tumor thick sections will be helpful to find out whether loss of *Atf4* rather leads to increased tumor hypersprouting coupled with reduced lumen formation as observed upon inhibition of Notch signaling in other tumor models, such as in *Dll4 +/-* mice. This also leads to reduced tumor growth and perfusion. It may be also interesting to check whether there is loss of secretion/production of Notch ligands, such as *Dll4* in endothelial cells.

We thank the reviewer for this comment. Based on our data (see Fig. 3c,d, Suppl. Fig. 5a-f), the α SMA expression is dramatically reduced (at both large vessels and microvessels) in both B16F10 and MH6419 tumors from *Atf4* ^{Δ/Δ} compared to *Atf4*^{wt/wt} mice. Also, our perfusion studies utilizing injected Texas Red-Dextran revealed significantly reduced vascular permeability in *Atf4* ^{Δ/Δ} (see Supplementary Fig. 4h,i), indicating disrupted microvessels. We agree that the mechanism for the disruption of tumor angiogenesis in the *Atf4* ^{Δ/Δ} mice is complex and may not be solely attributed to the non-functional vCAFs. Moreover, we acknowledge the importance of *Dll4*/Notch signaling in tumor angiogenesis. However, we feel that the *Dll4*/Notch signaling is out of the scope of this paper because we are focusing on the ISR pathway. Additionally, *Dll4* and Notch1 were not found in the list of DE genes from the scRNA-seq analysis of melanoma tumors. To study the effects of ATF4 loss on the tumor hypersprouting, we harvested B16F10 and MH6419 tumors from *Atf4*^{wt/wt} and *Atf4* ^{Δ/Δ} mice. The tumors were processed using the X-CLARITY™ Tissue Clearing System, thick sections were cut and stained for CD31/ α SMA. Unfortunately, we were not able to get good IF CD31 (or VE-cadherin) staining even though we tried several antibodies from different vendors. Interestingly, we were able to get good staining with α SMA. Representative images of α SMA IF staining in these thick sections validated the significant difference in α SMA levels between the two genotypes (data not included in the manuscript).

5. The authors should provide a more thorough analysis of the TME and tumors, including the analysis of mural coverage of vessels and proliferation and apoptosis of endothelial, CAFs and tumor cells.

This is an important point. The data from the scRNA-seq analysis showed that the vCAF gene signature is stronger than the mural (pericytes and vSMCs) gene signature (see detailed explanation below for comment #6). Although we still cannot exclude the possibility of some contribution from the pericyte compartment on some CAFs, we have shown that the % colocalization of the pericyte marker NG2⁺ with CD31⁺ was not different between the melanoma tumors grown in *Atf4*^{w^t/w^t and *Atf4*^{Δ/Δ} mice. To increase the statistical power of our observations, we repeated the NG2/CD31 IF staining on additional B16F10 tumors from 9 *Atf4*^{w^t/w^t and 7 *Atf4*^{Δ/Δ} mice, (Supplementary Fig. 5l,m). Again, the levels of NG2 levels were not different in tumors from the two *Atf4* genotypes.}}

Moreover, to analyze the proliferation status of the tumor, endothelial and CAFs cells in the tumor microenvironment, we stained B16F10 tumors from *Atf4*^{w^t/w^t and *Atf4*^{Δ/Δ} mice with anti-Ki67 antibody. We found that tumor cells in *Atf4*^{Δ/Δ} displayed significantly lower proliferation rates compared to the *Atf4*^{w^t/w^t, which is consistent with the significantly lower rates of tumor growth observed in *Atf4*^{Δ/Δ} mice (Supplementary Fig. 4o,p).}}

Similarly, significantly lower proliferation rates have been observed in both endothelial cells and CAFs in tumors grown in *Atf4*^{Δ/Δ} compared to *Atf4*^{w^t/w^t mice as shown by CD31/Ki67 and αSMA/Ki67 co-stains. Representative images and the quantification graphs have been added as Supplementary Fig. 4f,g (for CD31) and Supplementary Fig. 5j,k (for αSMA).}

To analyze the levels of apoptosis in the tumor microenvironment, we stained B16F10 tumor sections from *Atf4*^{wt/wt} and *Atf4*^{Δ/Δ} for TUNEL. We found that tumor cells in *Atf4*^{Δ/Δ} presented significantly higher apoptotic levels

compared to the *Atf4*^{wt/wt}, which is also consistent with the reduced tumor growth rates.

Representative images and the quantification graph have been added as **Supplementary Fig. 4m,n**.

Additionally, we found that the apoptosis levels of both the endothelial cells (% TUNEL⁺CD31⁺ cells) and CAFs (% TUNEL⁺αSMA⁺ cells) did not show any significant difference between the *Atf4*^{wt/wt} and *Atf4*^{Δ/Δ} mice, indicating that ATF4 ablation in these cell types results from a proliferation defect and not apoptosis. These data have been discussed in the manuscript but due to space limitations have not been included in the supplementary figures.

6. The authors suggest that ATF4 main target cell type are perivascular CAFs – how are they different from vSMCs and pericytes? In human tissues “ATF4 is highly expressed (apart from the tumor cells) on CAFs (αSMA) that localized on the perivascular area (CD34) compared to other CAFs reside away from the blood vessels (Fig. 3i).” – however such perivascular ATF4 could as well be vSMCs, ATF4 expression and role in vSMCs has been reported.

The reviewer brings up a valid point which we have carefully considered. To further delineate the cellular identity of the perivascular CAFs, we compiled a list of gene markers for mural cells (pericytes and vSMCs)¹¹. This gene signature comprises of 44 mural cell-specific genes that were commonly identified in four tissues (i.e., heart, colon, bladder and skeletal muscle). Furthermore, we created a list of perivascular CAFs gene markers¹². Both signatures can be found in Table below. Next, we visually/quantitatively inspected the signal strength of both gene signatures in the CAFs sub-cluster at question in both the small and large B16F10 tumor samples (please note that even though *Acta2*/αSMA is part of the list, we removed it from the mural cell signature because it is not unique to mural cells in the literature).

Figure. UMAP plots of B16F10 small (top row) and large (bottom row) tumors coupled with the expression signal of the vCAFs- and Mural cells-specific gene signatures.

As shown in the above figure, the signal strength of the vCAF gene signature is stronger in the sub-cluster at question (i.e., **sub-cluster 1** in small tumors and **sub-cluster 4** in large tumors) compared to the mural cell signature. This is supported by (a) the higher intensity of the color (orange) in the vCAFs signature UMAP plots compared to the mural cells plots and (b) the range of values (signal strength) is higher for the vCAF signature compared to the mural signature for both small B16F10 tumors (1.6 versus 0.75) and larger tumors (1.5 vs. 1.2).

Table. List of vCAFs- and mural cells-specific gene markers.

Gene signatures	
vCAFs	Sparcl1, Vtn, Col5a3, Nid2, Col4a1, Col4a2, Cd248, Kcnj8, Rbpms, Mef2c, Meox2, Pdlim1, Mcam, Notch3, Epas1, Col18a1, Nr2f2, Pdgfra, Angptl2, Il34, Des, Esam, Itga1, Pdgfrb
Mural cells	Des, Ppp1r12b, Lmod1, Mgst3, Rgs4, Pcp4l1, Stom, Cox4i2, Myl9, Pkig, Tpm2, Gja4, Tinagl1, Ppp1cb, Bcam, Uba2, Arhgef17, Mrvi1, Mob2, Sorbs2, Cnn1, Esam, Mcam, Tagln, Cspg4, Rasl12, Rbpms2, Tpm1, Ppp1r12a, Itga7, Atp1b2, S1pr3, Mef2c, Sncg, Ptp4a3, Myh11, Mylk, Filip1l, Notch3, Rcan2, Epas1, Cystm1, Rasgrp2, Flna

In further support of the notion that the main difference is in vCAFs and not mural cells, (a) *Cspg4* (the gene which encodes for NG2) was not present in the DE list from the scRNA-seq analysis. (b) Moreover, we stained additional B16F10 tumor samples for NG2/CD31 and confirmed that the NG2 levels remain unaffected in the tumors from both genotypes (see above response to **comment #5**).

7. The authors propose that unlike wild type fibroblasts, *Atf4* deficient fibroblasts do not support growth of B16F10 cells. It would be important to provide evidence that *Atf4* deficient fibroblast survive upon such co-injection, it is possible that they are simply do not survive s.c. transplantation.

This is an excellent suggestion. To test whether *Atf4*-deficient fibroblasts survive post-injection on the flank of the mice, we labeled the cytoplasmic membranes of both DFB^{wt/wt} and DFB^{Δ/Δ} cells using the non-cytotoxic Vybrant™ DiD Cell-Labeling dye right before s.c. injection. We then tracked the fluorescent signal from both DFB^{wt/wt} and DFB^{Δ/Δ} cells *in vivo* from **Day 0** (day of injection) to **Day 9** (when it is generally accepted that neovasculature is established), using a fluorescence machine (IVIS). Although there was a gradual loss of signal (indicating death of injected FBs), we found that both cell lines presented similar rates of decline over time. This strongly suggests that it is not the decline in numbers, but likely a functional deficiency of *Atf4*-deficient FBs which is responsible for the attenuated tumor growth. Similarly, when we co-inject DFB^{wt/wt} or DFB^{Δ/Δ} FBs in *ATF4*-deficient mice, only the former can rescue the tumor growth phenotype, due to activation of pro-angiogenic programs in the perivascular areas. Representative images and the quantification graph have been added as **Supplementary Fig. 6b-d**.

8. The authors should discuss in more detail potential mechanisms underlying a tumor escape from *Atf4* inhibition.

We would like to thank the reviewer for bringing out this point. In this study, we discuss the importance of host ATF4 in shaping CAF functionality to dictate extracellular matrix organization and angiogenesis to support tumor growth and progression. Even though host ATF4 ablation causes a significant delay in tumor growth in both melanoma and pancreatic tumor models (**Fig. 1d-g**), the tumors eventually grow

and reach the endpoint of the study when mice need to be euthanized. This indicates that other mechanisms may have been initiated to overcome the ATF4 ablation and accelerate the tumor growth. **(a)** We believe that the ATF4 excision may be incomplete in some cell types, including fibroblasts. As these cells are also proliferative (albeit more slowly compared to tumor cells), in turn, these nearly *Atf4*-proficient cells will acquire their potential pro-tumorigenic properties in contributing towards a tumor-promoting phenotype. This notion is supported by our scRNA-seq data which show a complete absence of *Atf4* mRNA signal in the CAFs of smaller tumors, but slightly elevated levels in CAFs of large tumors **(Supplementary Fig. 2b and 3c)**. **(b)** We have clearly shown in different mouse tumor models that host ATF4 ablation impacts the vascularization during all the stages of tumor initiation and progression. However, even small and “defective” blood vessels could continue to support, at a slower pace, the high demands of the tumor microenvironment, leading to a slow but progressively tumor growth. **(c)** We found that the vCAFs in tumors from *Atf4*^{Δ/Δ} mice present a defective angiogenesis-related secretome with the levels of both VEGF and SDF-1 to be significantly downregulated primarily in the perivascular area (Fig. 5e-h). However, the total levels of both angiogenic factors did not change between *Atf4*^{wt/wt} and *Atf4*^{Δ/Δ} mice, due to the tumor heterogeneity and the fact that the fibroblasts are not the sole source of the VEGF and SDF-1 secretion into the tumor microenvironment. The diffusion of these angiogenic factors towards the perivascular area/blood vessels may counterbalance the defective CAF-secretome and promote neo-angiogenesis. Nevertheless, *Atf4* excision post-establishment of palpable B16F10 tumors, still caused a significant inhibition in the tumor progression, which further underscores the critical role of ATF4 in the growing tumor and support the notion of ATF4 as a potential therapeutic target. We have amended our discussion to include a summary of these concepts.

Technical comments:

1. The authors state that they use orthotopic B16F10 model, however this is not the case as they do s.c. and not intradermal injections.

Thank you for this comment. We have deleted the word “orthotopic” when referring to the B16F10 tumors in the manuscript.

2. The authors mention that they identified a new population of melCAFs, with mixed melanoma and CAF identities - unusual “mixed” populations can be observed in scRNAseq data due to RNA transfer/uptake during sample preparation, especially in physically interacting cells. Please confirm out that this is not a sample preparation artifact by other means , e.g, costaining, or amend the analysis.

We would like to thank the reviewer for this mindful suggestion. Indeed, unusual mixed cell populations are commonly observed in scRNA-seq experiments. This could be the result of **(a)** RNAs transferred via extracellular vesicles between distant cells and/or via cell-to-cell direct contacts, **(b)** technical issues emerging during single-cell library preparation (e.g., occurrence of doublets, triplets etc.) or **(c)** extensive cellular plasticity in the tumor microenvironment¹³. In the first case, the observed concentration of the transferred RNA is reported to be significantly reduced compared to the concentration in the cells from which the RNA molecules originate from¹⁴. In the second case, doublets, triplets etc. (i.e., two or more cells inside the same bubble) are discarded before further downstream analysis. Finally, in the third case, RNA products of marker genes expressed primarily in specific cell-types are expected to be of close abundance.

Using three canonical CAF and three melanoma markers, we visualized the normalized expression of each gene marker across the melanoma cluster and CAF sub-clusters in both small and large B16F10 tumor samples (see figures below). In both figures, it is evident that **(a)** the expression levels of the three canonical CAF markers are of close abundance in all CAF sub-clusters (i.e., vCAFs, mCAFs, cCAFs, melCAFs), **(b)** the expression levels of the CAF markers are nearly undetectable in the melanoma cells cluster, **(c)** the expression levels of the three canonical melanoma markers are of close abundance in both melanoma cells and melCAFs sub-cluster and **(d)** the expression levels of the melanoma markers are nearly undetectable in the CAF sub-clusters, except for melCAFs (only in large tumors the *Mlana* gene presents minimal expression in mCAFs/cCAFs/vCAFs, while its distribution in melCAFs highly resembles that of melanoma cells). Collectively, these findings support the scenario that indeed melCAFs are a distinct population, comprising a distinct cell type, that presumably emerged due to the extensive cell plasticity that takes place in the tumor microenvironment^{15, 16}.

Figure. Normalized expression levels of three CAF and three melanoma markers in the melanoma cells cluster and the CAF sub-clusters in the small B16F10 tumor samples.

Figure. Normalized expression levels of three CAF and three melanoma markers in the melanoma cells cluster and the CAF sub-clusters in the large B16F10 tumor samples.

3. The authors sometimes use very small number of tumors for the analysis, please consider increasing the number of samples to make sure that the reported observation are indeed solid.

We thank the reviewer for this suggestion. To this end, we have added additional samples to the α SMA/CD31 co-stain (Fig. 3d and Supplementary Fig. 5a), to the NG2/CD31 co-stain (Supplementary Fig. 5l,m) and to the Dextran staining (Supplementary Fig. 4h). We believe that in most other cases, the magnitude of the biological phenotypes (e.g., Fig. 3a; Supplementary Fig. 4d,k; Supplementary 5b-d, g-i) is sufficiently large, and the statistical significance is very strong that lend ample support to the main conclusions. Please also note that we have used multiple tumor types (melanoma and pancreatic), and volumes (smaller and larger), with essentially comparable results.

Minor comments

1. CD31 is a pan-endothelial marker and not specific for tumor endothelium

We have corrected this statement in the manuscript.

2. Please mention the percentage of CAFs found in large B16F10 tumors like in small tumors.

We have added the percentage of CAFs found in large B16F10 tumors.

3. Regarding vessel permeability experiment, please comment on whether this is due to intrinsic (check endothelial gene signature from single cell RNA-seq) or extrinsic effect on endothelial cells.

Thank you for this comment. Due to the low number of endothelial cells captured for the scRNA-seq we did not identify many DE genes; thus, we cannot elucidate or exclude the possibility of an intrinsic effect. However, we have data supporting an extrinsic effect on endothelial cells that can also explain the changes in the blood vessel permeability: **a)** the collagen is primarily deposited in the perivascular area and was significantly reduced in the *Atf4*^{Δ/Δ} mice (see new Fig. 3i,j) and **b)** the levels of both VEGF and SDF-1 were significantly lower in the perivascular areas from tumors grown in the *Atf4*^{Δ/Δ} (see new Fig. 5e-h). We would like to refer the reviewer to our detailed response to reviewer 3 (comment #1).

4. supp Figure 4d, the merged figure look fuzzy -please replace

The figure has been replaced.

5. Supp 6d-e it would be easier to keep the naming of tumor and not use FB.

Thank you for this comment. We have replaced the FB with the “Col1a1” to indicate the fibroblast-specific ATF4 excision. The representative images and the graph have been updated (new Supplementary Fig. 6g-h).

6. Fig 4b should be in supplemental data because it is literature and not a result of the paper.

We agree with this suggestion. We moved this figure to supplementary data as Supplementary Fig. 7d.

7. Fig 4i please mention what the dots represent

We have updated the figure legend by adding “Each dot represents quantitative value from a 10x field.” Additionally, we updated the figure legend referring to Supplementary Fig. 7k.

8. Fig 5e mention the color code and the proteins in the legend

We have updated the figure legend (now as Fig. 5c) by adding the color code and the analyzed proteins.

Reviewer #2:

Minor concerns:

Line 177-178: The numbers of transcriptome analyses of small and large tumors could be more clearly explained in context of the WT and *Atf4*-delta mice.

We have updated the manuscript as per the reviewer's suggestion.

Fig 2C and related figures: The x-axis is not clear here and should be labeled.

We have updated these figures (Fig. 2c,f,h and Supplementary Fig. 3d,i), by moving the indicated gene from the top of each panel to below the x-axis.

Line 267: This sentence was not clear: what was primarily restricted in the blood vessels?

Thank you for this clarification. We have found that the α SMA expression was primarily expressed in the perivascular area of both B16F10 and MH6419 tumors. We have re-phrased this sentence accordingly.

Line 331: The uptake vs metabolic flux defect associated with *Atf4* was not fully clear. The manuscript notes that uptake of M+3 serine and M+7 glutamine was similar in WT versus *Atf4*-deficient cells, suggesting that transport of serine and glutamine was not affected by ATF4 loss. Is this data shown in the manuscript? Is the expression of genes involved in glycine or proline biosynthesis impacted in ATF4-deficient lung fibroblasts? What is the proposed mechanism for reduced biosynthesis of glycine and proline in ATF4-deficient cells?

We would like to thank the reviewer for this valuable comment. In the manuscript, we mention that "...the labeling fractions of M+3 serine and M+7 glutamine were similar in LFB^{wt/wt} and LFB ^{Δ/Δ} cells", which is supported by data in Supplementary Fig. 7g,h. In order to be clear, we cited these figures at the end of the above sentence.

Moreover, using NMR spectroscopy, we showed that the intracellular levels of glycine and proline were significantly reduced in ATF4 deficient cells (Fig. 4e). This can be explained by the regulation of the expression of genes involved in glycine or proline biosynthesis. We added schematic presentations to present the enzymes involved in glycine and proline biosynthesis (Supplementary Fig. 7e,f).

To delineate the impact of ATF4 on Gly and Pro biosynthetic pathways, we carried out RT-qPCR on LFB^{wt/wt} and LFB ^{Δ/Δ} . We found significantly reduced levels in enzymes involved in glycine (*Psat1*, *Shmt1* and *Shmt2*)¹⁷ and proline (*Aldh18a1* and *Pycr1*)¹⁸ biosynthesis in LFB ^{Δ/Δ} cells (Fig. 4d). We also observed that this difference was more pronounced in the enzymes involved in the proline biosynthesis, which was also validated by the significant downregulation of their expression levels in the microarray analysis of the LFBs (Fig. 4a). Also, the levels of *Aldh18a1* were found to be downregulated in the lungs of *Atf4* ^{Δ/Δ} compared to *Atf4*^{wt/wt} mice (Fig. 6a). Thus, these results support the notion that ATF4 regulates the expression of the genes involved in glycine and proline biosynthesis resulting in a significant reduction in the intracellular glycine and proline levels.

Line 344: Secretory stress caused by high collagen production is proposed to cause activation of the eIF2 kinase PERK, resulting in increased expression of Atf4 (Fig. 4K). Interestingly, silencing of Col1a1 reduced PERK activation but led to an increase in ATF4 expression in the absence of TGF-β1. Is there a change in eIF2 phosphorylation in Col1a1-silenced fibroblasts? Does this contribute to the secretory defect in ATF4-deficient fibroblasts?

We thank the reviewer for this mindful comment. We analyzed the p-eIF2α and total eIF2α levels in the western blot panel, along with the quantification under each band (Fig. 4k). Indeed, silencing of Col1a1 in the absence of TGF-β1 leads to a reduction in PERK activation. However, we do see an increase in the levels of p-eIF2α which is also followed by an increase in ATF4 expression. This can be explained by the activation of GCN2 kinase to compensate for the reduction in PERK activation. To expand these data, we repeated the

experiment at a later time point (24 h of TGF- β 1 treatment). Similarly, ATF4 expression remains in relatively high levels under the siCol1a1/no TGF- β 1 treatment, which however does not coincide with the lower levels of p-eIF2 α possibly due to differences in their kinetics (**Supplementary Fig. 7I**). Thus, we believe that both PERK and GCN2 kinases contribute to the p-eIF2 α /ATF4 pathway in a complementary and compensatory manner. This is an important point, so we added a comment in the discussion section (see also below, the response to your comment about the discussion).

Line 526: This sentence was not fully clear.

Thank you for the suggestion. We re-phrased the sentence as “Collectively, our work highlights the paramount importance of the ATF4 on regulating the functionality and activation of CAFs through the collagen I synthesis and TGF β /Smad3 pathways”.

Discussion: There was no significant discussion of which eIF2 kinase may be the most prominent in driving CAF function. The manuscript mentions a GCN2 signature in ATF4-deficient CAFs (Fig. 2D) and demonstrates PERK activation in wild-type lung fibroblasts (Fig. 4K), but does not sufficiently revisit these points in the discussion. These would seem to be worthwhile ideas to discuss in the context of therapeutic strategies.

We completely agree and acknowledge that there should be clearer recognition of this point at the discussion section. It is well established that TGF- β 1/Smad3 pathway is active in CAFs. Our *in vitro* data show that the high demands in collagen production after TGF- β 1 stimulation (secreted in the TME) cause ER stress followed by the phosphorylation and activation of PERK/p-eIF2 α /ATF4, underscoring the active role of PERK in CAFs response to collagen production. However, we noticed that the ATF4 levels post-siCol1a1 treatment remain elevated (as are those of p-eIF2 α), while PERK remains unphosphorylated. These data support the notion that another ISR kinase, (perhaps GCN2), becomes active under these *in vitro* conditions. The speculation of a GCN2 involvement is supported by the data from the scRNA-seq showing that the higher expression of genes in *Atf4*^{Δ/Δ} CAFs is associated mainly with ISR activation, including response of GCN2 to amino acid deficiency and eukaryotic translation initiation. These data can be explained by the activation of GCN2 kinase to compensate for the reduction in PERK activation to continue supporting CAF functionality.

Reviewer #3:

1- Authors show throughout the manuscript that ATF4 plays a remarkable role as a driver of collagen expression. In figure S4f and S4g, the authors show that vasculature has reduced permeability when host ATF4 is deleted. However, it is not clear whether the reduced permeability in tumor vasculature is due to reduced collagen production by ATF4 deficient perivascular CAFs or due to reduction of other ATF4-regulated factors.

We would like to thank the reviewer for bringing out this point. To better address this we performed a CD31/collagen IF co-stain on B16F10 tumors from *Atf4^{wt/wt}* and *Atf4^{Δ/Δ}* mice. We found that the % collagen positive area was significantly reduced in tumors from *Atf4^{Δ/Δ}* mice (new Fig. 3i,j), which confirms the data from Fig. 3g,h (collagen⁺ area from whole tumor sections). However, the most significant finding from this co-stain experiment is that the collagen deposition was primarily located in the perivascular area (CD31+), indicating that the perivascular CAFs (vCAFs) must be the primary source of the collagen deposition. Thus, the levels of collagen deposition in the perivascular area can, at least partially explain the changes in the blood vessel permeability.

Another factor likely to contribute to the changes in blood vessel permeability in our model is the CAF angiogenesis-related secretome. We found that fibroblasts deficient in ATF4 present with a defective secretome with reduced levels of angiogenic factors such as VEGF, SDF-1 etc. (Fig. 5c,d). It is well known that mainly VEGF, but SDF-1 as well, can increase the blood vessel permeability¹⁹⁻²³. Interestingly, in new experiments, we show that the perivascular levels of both VEGF and SDF-1 were significantly reduced in the B16F10 tumors from *Atf4^{Δ/Δ}* mice (new Fig. 5e-h, see also our response to general comment #2 from reviewer 1), indicating that these angiogenic factors can also explain the reduced permeability of vasculature in *Atf4^{Δ/Δ}* mice as shown by the Dextran staining.

2- Considering the reduced number of metastases and the defective vasculature in tumors grown in ATF4 ko mice, it would be interesting to know whether the lack of host ATF4 affects intravasation/extravasation of tumor cells into the vasculature.

We thank the reviewer for this comment. To address this, we performed a colony forming assay using circulating tumor cells isolated from the plasma of B16F10-GFP bearing *Atf4^{wt/wt}* and *Atf4^{Δ/Δ}* mice. After 15-20 days post plating, we were able to count 2 big colonies in only 1 out of 7 *Atf4^{wt/wt}* mice and 2 smaller colonies in 1 out of 8 *Atf4^{Δ/Δ}* mice, thus we cannot draw any conclusions. We need to acknowledge that this is a very challenging experiment with many limitations that needs optimization in multiple steps.

3- They show that tumor growth is slowed in ATF ko mice after tail vein injection. This could be due to reduced growth as the authors suggest and/or reduced seeding into the lung. To address this they should delete ATF4 AFTER tail vein injection.

We thank the reviewer for this mindful comment. We repeated the lung colonization experiment as suggested. Specifically, we followed our previous protocol (see new **Fig. 6d**) with the only difference being that the tamoxifen treatment (5 days) started at 3 days post tail vein injection of B16F10 cells. Interestingly, we found that both the number of lung metastases (macroscopic evaluation) and the % tumor area (based on the H&E staining) were significantly reduced in the $Atf4^{\Delta/\Delta}$ mice (**Supplementary Fig. 8i-l**). This indicates that ATF4 ablation renders the metastatic niche less permissive that reduces both the initial seeding and the growth of the tumor cells. The manuscript has been updated accordingly.

4- The authors use the MH6419 pancreatic cancer cell line as a complementary model. However, MH6419 cells are injected subcutaneously and not orthotopically. It is well known that the tumor microenvironment can change drastically depending on the tumor site. High stromal content is a remarkable feature of pancreatic cancer, hence, it would increase the biological relevance if the MH6419 cells were orthotopically implanted. Without this the addition of the MH6419 doesn't add much to the manuscript and it would be better to supply an additional melanoma cell line.

We thank the reviewer for bringing out this important comment. Indeed, the tumor microenvironment can be highly variable depending on the tumor site. Thus, we agree on the importance of the orthotopic pancreatic tumor model in the manuscript and we have spent considerable time developing and testing this model. We injected 5×10^4 MH6419 cells orthotopically on the tail of the pancreas of $Atf4^{wt/wt}$ and $Atf4^{\Delta/\Delta}$ mice. Three weeks post-injection, we recorded the bodyweight of the mice, and then we weighed and imaged the harvested pancreas (normal and tumor) and stored it in formalin. We found that growth of these orthotopic pancreatic tumors was significantly reduced in the $Atf4^{\Delta/\Delta}$ mice, as it is

shown (a) in the image of all harvested pancreas (Fig. 1h) (yellow and blue dotted lines indicate the tumor and normal area of the pancreas, respectively), (b) the quantification graph of % tumor normalized to body weight (Fig. 1i) and (c) from the H&E stained samples and its quantification (Supplementary Fig. 1i,j).

Moreover, to validate if our hypothesis model applies to the highly desmoplastic orthotopic pancreatic tumor model, we stained orthotopic tumors for α SMA and CD31. Indeed, we found that the perivascular α SMA⁺ area was significantly reduced in the *Atf4*^{Δ/Δ} mice while the blood vessels were smaller and sparse (new

Supplementary Fig 5e,f). This finding further validates the importance of ATF4 on the CAFs functionality and their contribution to angiogenesis and eventually tumor growth.

5- The authors argue that the lack of host ATF4 affects tumor initiation and tumor progression. The role of ATF4 in tumor progression is well addressed by using tumor cell implantation models. However, to better understand the role of host ATF4, more specific ATF4-expressing fibroblasts, in tumor initiation, it would be interesting to see how ATF4 knockout in fibroblasts (Col1a1-cre) would affect tumor generation in a GEMM model.

We agree that the distinction in the role of ATF4 in initiation vs. progression is important. We believe that we have partially addressed this by showing that even when host ATF4 was excised after the establishment of palpable B16F10 tumors (Fig. 1f), we demonstrate that ATF4 plays a role in progression as well. We acknowledge that a cross of the *Col1a1;Cre* mice with a GEMM model would be the ideal setting to test tumor initiation. However, such an experiment would be required very complex crossings (requiring at least 4 transgenes). Both GEMM melanoma models (e.g., *Tyr::NRAS^{Q61}/⁰; Tyr::Cre/⁰; Trp53^{-/-}*)²⁴ and PDAC (*LSL-Kras^{G12D}/+; LSL-Trp53^{R172H}/+; Pdx1-Cre*)²⁵ involve 3 different transgenes. Therefore, crossing to *Atf4^{fl/fl}* and *Col1a1;Cre* mice would take years of crosses and be prohibitively expensive. Please note that in addition to the models we presented in our original submission, we now provide data with orthotopically implanted syngeneic tumors. Nevertheless, we understand that the term “initiation” implies a series of genetically defined events including transformation and tumorigenesis; therefore, we have eliminated the term “tumor initiation” from the text.

6- The authors claim one of their CAF subpopulations consists of melCAFs, which retain some “traits” from melanoma cells. What does this mean and how to they rule out the possibility that melCAFs are in fact, melanoma cells?

We thank the reviewer for the comment. To keep our responses as concise as possible, we would like to refer the reviewer to our detailed response to reviewer 1 (**technical comments, answer #2**).

7- The authors showed that larger tumors still have defective vascularization however collagen levels are not altered at this stage. Considering the perivascular localization of ATF4-expressing CAFs, are the reduced levels of collagen observed specific to the perivascular niche? CD31 (or other endothelial marker) costaining would help to identify whether collagen production is defective in perivascular areas of large tumors or whether collagen levels were normalized by an alternative pathway.

Thank the reviewer for this suggestion. We performed the collagen/CD31 co-stain on large B16F10 tumors, and we found that **(a)** the collagen deposition was primarily restricted in the perivascular area and **(b)** the % collagen positive area was significantly reduced in tumors from *Atf4^{Δ/Δ}* mice (**Fig. 3i,j**). Please also refer to our detailed response in **comment #1**, for the representative images and quantification graphs.

8- In line 286/287, the authors quote “DFBwt/wt injected into ATF4Δ/Δ mice nearly completely reverse tumor growth inhibition observed in ATF4Δ/Δ + DFBΔ/Δ group” (fig 3k). Despite the fact that there is a clear reversal of the tumor growth inhibition by the lack of ATF4 in DFBs, as pointed by the authors and supported by this and other results, the role of ATF4 in other cell types within the tumor microenvironment can’t be ruled out. This quote sounds like an overstatement.

Thank you for the comment. We agree that the ATF4 excision may affect other cell types within the tumor microenvironment to drive this inhibitory phenotype. Thus, we have re-phrased the sentence as “*DFB^{wt/wt}* injected into *Atf4^{Δ/Δ}* mice significantly reversed the tumor growth inhibition observed in *Atf4^{Δ/Δ}* + *DFB^{Δ/Δ}* group, while *DFB^{Δ/Δ}* injected into *Atf4^{wt/wt}* mice, caused a delay in tumor growth compared to the *Atf4^{wt/wt}* + *DFB^{wt/wt}* group (**Fig. 3m**)”.

General comments

1. We changed the font color in the LFB and lung microarray analyses to indicate the downregulated (in blue) and upregulated (in red) genes.
2. In the quantification graphs of α SMA/CD31, PDGFR β /CD31, NG2/CD31 co-stains we updated the y-axes as “% colocalization of α SMA⁺ with CD31⁺” etc.
3. In the western blots, we added the ratio of the expression of p-eIF2 α , ATF4 and p-SMAD3 normalized to the expression of loading control (β -actin or β -tubulin) or T-eIF2 α .
4. The “qRT-PCR” has been replaced with “RT-qPCR” in the manuscript, methods and figure legends.
5. We have corrected the “Atf4^{wt/wt} and Atf4 ^{Δ/Δ} ” to “Atf4^{wt/wt} and Atf4 ^{Δ/Δ} ” in the manuscript and figure legends.
6. We have added statistics in Supplementary Fig. 2d.
7. Figure legends have been updated.
8. Statistical analysis section has been updated.
9. We have updated the References.
10. We have updated the Acknowledgments section.

References

1. Wu, D., Huang, Q., Orban, P.C. & Levings, M.K. Ectopic germline recombination activity of the widely used Foxp3-YFP-Cre mouse: a case report. *Immunology* **159**, 231-241 (2020).
2. Aad, G. *et al.* Measurement of Z boson production in Pb-Pb collisions at $\sqrt{s(NN)}=2.76$ TeV with the ATLAS detector. *Phys Rev Lett* **110**, 022301 (2013).
3. Folkins, C. *et al.* Glioma tumor stem-like cells promote tumor angiogenesis and vasculogenesis via vascular endothelial growth factor and stromal-derived factor 1. *Cancer research* **69**, 7243-7251 (2009).
4. Ponti, D. *et al.* Isolation and in vitro propagation of tumorigenic breast cancer cells with stem/progenitor cell properties. *Cancer research* **65**, 5506-5511 (2005).
5. Beckermann, B.M. *et al.* VEGF expression by mesenchymal stem cells contributes to angiogenesis in pancreatic carcinoma. *Br J Cancer* **99**, 622-631 (2008).
6. Hughes, R. *et al.* Perivascular M2 Macrophages Stimulate Tumor Relapse after Chemotherapy. *Cancer research* **75**, 3479-3491 (2015).
7. Stockmann, C. *et al.* Deletion of vascular endothelial growth factor in myeloid cells accelerates tumorigenesis. *Nature* **456**, 814-818 (2008).
8. Facciabene, A., Motz, G.T. & Coukos, G. T-regulatory cells: key players in tumor immune escape and angiogenesis. *Cancer research* **72**, 2162-2171 (2012).
9. Facciabene, A. *et al.* Tumour hypoxia promotes tolerance and angiogenesis via CCL28 and T(reg) cells. *Nature* **475**, 226-230 (2011).
10. Carpentier, G. *et al.* Angiogenesis Analyzer for ImageJ - A comparative morphometric analysis of "Endothelial Tube Formation Assay" and "Fibrin Bead Assay". *Sci Rep* **10**, 11568 (2020).
11. Muhl, L. *et al.* Single-cell analysis uncovers fibroblast heterogeneity and criteria for fibroblast and mural cell identification and discrimination. *Nature communications* **11**, 3953 (2020).
12. Bartoschek, M. *et al.* Spatially and functionally distinct subclasses of breast cancer-associated fibroblasts revealed by single cell RNA sequencing. *Nature communications* **9**, 5150 (2018).
13. Yuan, S., Norgard, R.J. & Stanger, B.Z. Cellular Plasticity in Cancer. *Cancer Discov* **9**, 837-851 (2019).
14. Haimovich, G. *et al.* Intercellular mRNA trafficking via membrane nanotube-like extensions in mammalian cells. *Proceedings of the National Academy of Sciences of the United States of America* **114**, E9873-E9882 (2017).
15. Sahai, E. *et al.* A framework for advancing our understanding of cancer-associated fibroblasts. *Nat Rev Cancer* **20**, 174-186 (2020).
16. Ohlund, D., Elyada, E. & Tuveson, D. Fibroblast heterogeneity in the cancer wound. *J Exp Med* **211**, 1503-1523 (2014).
17. Selvarajah, B. *et al.* mTORC1 amplifies the ATF4-dependent de novo serine-glycine pathway to supply glycine during TGF-beta1-induced collagen biosynthesis. *Sci Signal* **12** (2019).
18. D'Aniello, C. *et al.* A novel autoregulatory loop between the Gcn2-Atf4 pathway and (L)-Proline [corrected] metabolism controls stem cell identity. *Cell Death Differ* **22**, 1094-1105 (2015).
19. Sirois, M.G. & Edelman, E.R. VEGF effect on vascular permeability is mediated by synthesis of platelet-activating factor. *Am J Physiol* **272**, H2746-2756 (1997).
20. Bates, D.O. Vascular endothelial growth factors and vascular permeability. *Cardiovasc Res* **87**, 262-271 (2010).
21. Dvorak, H.F. Reconciling VEGF With VPF: The Importance of Increased Vascular Permeability for Stroma Formation in Tumors, Healing Wounds, and Chronic Inflammation. *Front Cell Dev Biol* **9**, 660609 (2021).
22. Azzi, S., Hebda, J.K. & Gavard, J. Vascular permeability and drug delivery in cancers. *Frontiers in oncology* **3**, 211 (2013).

23. Butler, J.M. *et al.* SDF-1 is both necessary and sufficient to promote proliferative retinopathy. *The Journal of clinical investigation* **115**, 86-93 (2005).
24. Gembarska, A. *et al.* MDM4 is a key therapeutic target in cutaneous melanoma. *Nat Med* **18**, 1239-1247 (2012).
25. Hingorani, S.R. *et al.* Trp53R172H and KrasG12D cooperate to promote chromosomal instability and widely metastatic pancreatic ductal adenocarcinoma in mice. *Cancer Cell* **7**, 469-483 (2005).

Decision Letter, first revision:

Subject: Decision on Nature Cell Biology submission NCB-K45380A
Message:

*Please delete the link to your author homepage if you wish to forward this email to co-authors.

Dear Dr Koumenis,

Your manuscript, "A Stromal Integrated Stress Response Activates Perivascular Cancer-Associated Fibroblasts to Drive Angiogenesis and Tumor Progression", has now been seen by the original Reviewer 2 and 3. Please note that as the original Reviewer 1 was not available to re-review, a new reviewer (Reviewer 4) was recruited to cross-comment on your response to Reviewer 1. As you will see from their comments (attached below) they find this work of interest, but have raised some important points. Although we are also very interested in this study, we believe that their concerns should be addressed before we can consider publication in Nature Cell Biology.

Nature Cell Biology editors discuss the referee reports in detail within the editorial team, including the chief editor, to identify key referee points that should be addressed with priority, and requests that are overruled as being beyond the scope of the current study. To guide the scope of the revisions, I have listed these points below. We are committed to providing a fair and constructive peer-review process, so please feel free to contact me if you would like to discuss any of the referee comments further.

In particular, it would be essential to:

A) Address the persisting concerns as questioned by Reviewer 4;

B) All other referee concerns pertaining to strengthening existing data, providing controls, methodological details, clarifications and textual changes as applicable should also be addressed.

C) Finally please pay close attention to our guidelines on statistical and methodological reporting (listed below) as failure to do so may delay the reconsideration of the revised manuscript. In particular please provide:

We therefore invite you to take these points into account when revising the manuscript. In addition, when preparing the revision please:

- ensure that it conforms to our format instructions and publication policies (see below and <https://www.nature.com/nature/for-authors>).

- provide a point-by-point rebuttal to the full referee reports verbatim, as provided at the end of this letter.

- provide the completed Reporting Summary (found here <https://www.nature.com/documents/nr-reporting-summary.pdf>). This is essential for reconsideration of the manuscript and will be available to editors and referees in the event of peer review. For more information see <http://www.nature.com/authors/policies/availability.html> or contact me.

When submitting the revised version of your manuscript, please pay close attention to our [href="https://www.nature.com/nature-research/editorial-policies/image-integrity">Digital Image Integrity Guidelines](https://www.nature.com/nature-research/editorial-policies/image-integrity). and to the following points below:

- that unprocessed scans are clearly labelled and match the gels and western blots presented in figures.
- that control panels for gels and western blots are appropriately described as loading on sample processing controls

-- all images in the paper are checked for duplication of panels and for splicing of gel lanes.

Nature Cell Biology is committed to improving transparency in authorship. As part of our efforts in this direction, we are now requesting that all authors identified as 'corresponding author' on published papers create and link their Open Researcher and Contributor Identifier (ORCID) with their account on the Manuscript Tracking System (MTS), prior to acceptance. ORCID helps the scientific community achieve unambiguous attribution of all scholarly contributions. You can create and link your ORCID from the home page of the MTS by clicking on 'Modify my Springer Nature account'. For more information please visit www.springernature.com/orcid.

This journal strongly supports public availability of data. Please place the data used in your paper into a public data repository, or alternatively, present the data as Supplementary Information. If data can only be shared on request, please explain why in your Data Availability Statement, and also in the correspondence with your editor. Please note that for some data types, deposition in a public repository is mandatory - more information on our data deposition policies and available repositories appears below.

[REDACTED]

We would like to receive the revision within four weeks. If submitted within this time period, reconsideration of the revised manuscript will not be affected by related studies published elsewhere, or accepted for publication in Nature Cell Biology in the meantime. We would be happy to consider a revision even after this timeframe, but in that case we will consider the published literature at the time of resubmission when assessing the file.

We hope that you will find our referees' comments, and editorial guidance helpful. Please do not hesitate to contact me if there is anything you would like to discuss.

Best wishes,

Zhe Wang

Zhe Wang, PhD
Senior Editor
Nature Cell Biology

Tel: +44 (0) 207 843 4924
email: zhe.wang@nature.com

Reviewers' Comments:

Reviewer #2:

Remarks to the Author:

This revised manuscript addresses the important role of ATF4 and the Integrated stress response in the tumor environment during cancer progression. While ATF4 has previously been implicated in promoting tumor growth directly, the role of ATF4 in tumor stroma had not been previously investigated. The study combined ATF4 knock-out mice in subcutaneous melanoma (orthotopic) and pancreatic mouse models of cancer with single cell transcriptome analyses and determined that a population of cancer-associated fibroblasts require ATF4 to support neo-angiogenesis, vascularization, and collagen (ECM) expression to promote efficient tumor growth and metastasis. The major conclusion is that ATF4 is a major driver of perivascular CAF function and plays a critical role in shaping the tumorigenic microenvironment. The animal experiments are elegantly designed and well-controlled, and the results support the major conclusions of the manuscript. The manuscript text and figures are clear and succinct.

Overall, the manuscript adds significantly to our knowledge of the role of the Integrated stress response (ISR) in the tumor microenvironment, and specifically implicates the ISR and ATF4 as critical regulators of CAF function, an important cell type implicated in the progression, growth, and metastasis of tumors. Prior reviewer concerns were thoroughly addressed. This manuscript has important implications for new anticancer therapies and would be of broad interest.

Reviewer #3:

Remarks to the Author:

The authors have addressed my major concerns.

Reviewer #4:

Remarks to the Author:

I report below the original comments of Rev 1 and my feedback to the authors' response to each point ("Response").

General comments

1. The authors should clearly demonstrate that the phenotype is a result of loss of Atf4 in fibroblasts and not tumor endothelial cells. Their observations in Col1a1-CreERT2 mice are encouraging, however it will be important to rule a potential ectopic activity of Cre in tumor vasculature, for example by RT-qPCR analysis of sorted tumor endothelial cells.

Response: This point has been convincingly addressed by measuring Atf4 mRNA in tumor endothelial cells (TECs) with or without floxed Atf4 alleles (no differences were found). The best positive control would be CAFs isolated from the same tumors as TECs, rather than lung fibroblasts (LFB). Yet, what matters most is that no differences were found in TECs.

2. The authors show that one of the main effects of Atf4 deficiency is reduced tumor angiogenesis, and propose, based on their in vitro results, that this is a result of reduced production of pro-angiogenic factors, such as VEGF and SDF1, by CAFs. There is a missing link with in vivo observations – do Atf4 deficient fibroblasts have reduced levels of Vegfa mRNA or they display reduced Vegfa secretion and deposition? Do the authors observe reduced production of VEGF&other factors in vivo? If the main idea is that CAF secretome is affected, it would be important to confirm the results in independent experiments using Limunex or Elisa of tumor lysates but also for in vitro data, right now it looks like n=1 data.

Response: This point has been experimentally addressed. The new data are not yet fully compelling, even though they are interesting and somewhat support the authors' conclusions. But definitive proof that perivascular CAFs secrete VEGFA and CXCL12 (correct name for SDF1) in an ATF4-dependent manner is still lacking. The authors are right that VEGFA and CXCL12 protein levels are not expected to drop in tumors with conditional Atf4 deletion, as many cells other than CAFs express those proteins in the TME. Yet, the data in Fig 5e-h (and S8) are not fully convincing, as there is no proof that the relative

decrease of VEGFA and CXCL12 protein levels in the perivascular space is in fact due to cell-autonomous effects in the CAFs. The perivascular space is populated by pericytes, macrophages and other cells that also express functional VEGFA and CXCL12. I propose the following:

- Fig 5e-h should be improved and include CAF, pericyte and macrophage-specific markers, along with VEGFA and CXCL12, with better confocal resolution and relative quantification. Currently, it is unclear which cell types express VEGFA and CXCL12 in the peri-endothelial space.
- The Luminex data, shown in the rebuttal, that indicate no difference in total VEGFA and CXCL12 levels in Atf4 wt and DD tumors should be included in the ms and described.

3. Related to 1, Figure 5 has several issues:

- Endothelial cells seems to be in poor shape/dying (Fig 5b)
- no positive control is included
- use of Matrigel tube forming assay is a suboptimal way for modeling angiogenesis in vitro (PMID: 25931450), the authors should consider alternatives, such as endothelial spheroid sprouting assay.

Response: These points have been experimentally addressed. I agree with the Reviewer that Matrigel tube forming assay is not an adequate model to study angiogenesis in vitro. Yet, the authors' new data and response may be acceptable at this stage.

4. The IHC analysis in Figure 3 shows loss of larger aSMA-covered tumor vessels, but not microvessels – such effect is opposite from what is expected upon loss of tumor angiogenesis signaling and indicates a more complex underlying mechanism than the one proposed by the authors. Analysis of tumor thick sections will be helpful to find out whether loss of Atf4 rather leads to increased tumor hypersprouting coupled with reduced lumen formation as observed upon inhibition of Notch signaling in other tumor models, such as in Dll4 +/- mice. This also leads to reduced tumor growth and perfusion. It may be also interesting to check whether there is loss of secretion/production of Notch ligands, such as Dll4 in endothelial cells.

Response: The Reviewer argues that the effects of Atf4 inactivation on the angiogenic tumor vasculature were not thoroughly characterized. I agree with this concern, which remains valid. While I agree with the authors that looking at Notch/DLL4 signaling may be beyond the scope of the study, it is felt that better characterization of the tumor vasculature should be provided, at least by addressing the question on sprouting and other vascular parameters using thick sections. The quality of CD31 staining in Fig S4 and other figures is often suboptimal and should be refined (eg, is co-localization with ki67 reliable?).

5. The authors should provide a more thorough analysis of the TME and tumors, including the analysis of mural coverage of vessels and proliferation and apoptosis of endothelial, CAFs and tumor cells.

Response: The authors have added substantial new data. They should include all the data shown in the rebuttal letter in the ms (including data on TUNEL+ ECs/CAFs). Some reservations remain on the quality of vascular staining (see point 4 above), but overall this point has been satisfactorily addressed.

6. The authors suggest that ATF4 main target cell type are perivascular CAFs – how are they different from vSMCs and pericytes? In human tissues “ATF4 is highly expressed (apart from the tumor cells) on CAFs (αSMA) that localized on the perivascular area (CD34) compared to other CAFs reside away from the blood vessels (Fig. 3i).” – however such perivascular ATF4 could as well be vSMCs, ATF4 expression and role in vSMCs has been reported.

Response: I agree with the Review and feel that the authors’ response is not convincing. The new data shown in the rebuttal do not make a compelling case that vCAFs differ from mural cells/pericytes (are we splitting the hair with gene expression data?). In fact, vCAFs should be regarded as (a subset of) “mural cells”, unless differential origins or phenotypes are clearly identified, which is not the case here. I recommend the data be included in the ms and any claim be toned down and critically reassessed.

7. The authors propose that unlike wild type fibroblasts, Atf4 deficient fibroblasts do not support growth of B16F10 cells. It would be important to provide evidence that Atf4 deficient fibroblast survive upon such co-injection, it is possible that they are simply do not survive s.c. transplantation.

Response: Response and new data are compelling.

8. The authors should discuss in more detail potential mechanisms underlying a tumor escape from Atf4 inhibition.

Response: Response and discussion are adequate.

Technical comments:

1. The authors state that they use orthotopic B16F10 model, however this is not the case as they do s.c. and not intradermal injections.

2. The authors mention that they identified a new population of melCAFs, with mixed melanoma and CAF identities - unusual “mixed” populations can be observed in scRNAseq data due to RNA transfer/uptake during sample preparation, especially in physically interacting cells. Please confirm out that this is not a sample preparation artifact by other means, e.g, costaining, or amend the analysis.

3. The authors sometimes use very small number of tumors for the analysis, please consider increasing the number of samples to make sure that the reported observation are indeed solid.

Response: Response to the 3 technical comments is adequate. All data shown in the rebuttal should be included in the ms and addressed with text revisions (point 2).

Minor comments

1. CD31 is a pan-endothelial marker and not specific for tumor endothelium
2. Please mention the percentage of CAFs found in large B16F10 tumors like in small tumors.
3. Regarding vessel permeability experiment, please comment on whether this is due to intrinsic (check endothelial gene signature from single cell RNA-seq) or extrinsic effect on endothelial cells.
4. supp Figure 4d, the merged figure look fuzzy -please replace
5. Supp 6d-e it would be easier to keep the naming of tumor and not use FB.
6. Fig 4b should be in supplemental data because it is literature and not a result of the paper.
7. Fig 4i please mention what the dots represent
8. Fig 5e mention the color code and the proteins in the legend

Response: Response to all minor comments is adequate.

GUIDELINES FOR SUBMISSION OF NATURE CELL BIOLOGY ARTICLES

ARTICLE FORMAT

ABSTRACT – should not exceed 150 words and should be unreferenced. This paragraph is the most visible part of the paper and should briefly outline the background and rationale for the work, and accurately summarize the main results and conclusions. Key genes, proteins and organisms should be specified to ensure discoverability of the paper in online searches.

TEXT – the main text consists of the Introduction, Results, and Discussion sections and must not exceed 3500 words including the abstract. The Introduction should expand on the background relating to the work. The Results should be divided in subsections with subheadings, and should provide a concise and accurate description of the experimental findings. The Discussion should expand on the findings and their implications. All relevant primary literature should be cited, in particular when discussing the background and specific findings.

REFERENCES – are limited to a total of 70 in the main text and Methods combined,. They must be numbered sequentially as they appear in the main text, tables and figure legends and Methods and must follow the precise style of Nature Cell Biology references. References only cited in the Methods should be numbered consecutively following the last reference cited in the main text. References only associated with Supplementary Information (e.g. in supplementary legends) do not count toward the total reference limit and do not need to be cited in numerical continuity with references in the main

text. Only published papers can be cited, and each publication cited should be included in the numbered reference list, which should include the manuscript titles. Footnotes are not permitted.

Methods should be written concisely, but should contain all elements necessary to allow interpretation and replication of the results. As a guideline, Methods sections typically do not exceed 3,000 words. The Methods should be divided into subsections listing reagents and techniques. When citing previous methods, accurate references should be provided and any alterations should be noted. Information must be provided about: antibody dilutions, company names, catalogue numbers and clone numbers for monoclonal antibodies; sequences of RNAi and cDNA probes/primers or company names and catalogue numbers if reagents are commercial; cell line names, sources and information on cell line identity and authentication. Animal studies and experiments involving human subjects must be reported in detail, identifying the committees approving the protocols. For studies involving human subjects/samples, a statement must be included confirming that informed consent was obtained. Statistical analyses and information on the reproducibility of experimental results should be provided in a section titled “Statistics and Reproducibility”.

All Nature Cell Biology manuscripts submitted on or after March 21 2016, must include a Data availability statement as a separate section after Methods but before references, under the heading “Data Availability”. For Springer Nature policies on data availability see <http://www.nature.com/authors/policies/availability.html>; for more information on this particular policy see <http://www.nature.com/authors/policies/data/data-availability-statements-data-citations.pdf>. The Data availability statement should include:

- Accession codes for primary datasets (generated during the study under consideration and designated as “primary accessions”) and secondary datasets (published datasets reanalysed during the study under consideration, designated as “referenced accessions”). For primary accessions data should be made public to coincide with publication of the manuscript. A list of data types for which submission to community-endorsed public repositories is mandated (including sequence, structure, microarray, deep sequencing data) can be found here <http://www.nature.com/authors/policies/availability.html#data>.
- Unique identifiers (accession codes, DOIs or other unique persistent identifier) and hyperlinks for datasets deposited in an approved repository, but for which data deposition is not mandated (see here for details <http://www.nature.com/sdata/data-policies/repositories>).

- At a minimum, please include a statement confirming that all relevant data are available from the authors, and/or are included with the manuscript (e.g. as source data or supplementary information), listing which data are included (e.g. by figure panels and data types) and mentioning any restrictions on availability.
- If a dataset has a Digital Object Identifier (DOI) as its unique identifier, we strongly encourage including this in the Reference list and citing the dataset in the Methods.

We recommend that you upload the step-by-step protocols used in this manuscript to the Protocol Exchange. More details can found at www.nature.com/protocolexchange/about.

DISPLAY ITEMS – main display items are limited to 6-8 main figures and/or main tables. For Supplementary Information see below.

FIGURES – Colour figure publication costs \$395 per colour figure. All panels of a multi-panel figure must be logically connected and arranged as they would appear in the final version. Unnecessary figures and figure panels should be avoided (e.g. data presented in small tables could be stated briefly in the text instead).

All imaging data should be accompanied by scale bars, which should be defined in the legend. Cropped images of gels/blots are acceptable, but need to be accompanied by size markers, and to retain visible background signal within the linear range (i.e. should not be saturated). The boundaries of panels with low background have to be demarked with black lines. Splicing of panels should only be considered if unavoidable, and must be clearly marked on the figure, and noted in the legend with a statement on whether the samples were obtained and processed simultaneously. Quantitative comparisons between samples on different gels/blots are discouraged; if this is unavoidable, it has to be performed for samples derived from the same experiment with gels/blots were processed in parallel, which needs to be stated in the legend.

Regardless of format, all figures must be vector graphic compatible files, not supplied in a flattened raster/bitmap graphics format, but should be fully editable, allowing us to highlight/copy/paste all text and move individual parts of the figures (i.e. arrows, lines, x and y axes, graphs, tick marks, scale bars etc). The only parts of the figure that should be in pixel raster/bitmap format are photographic images or 3D rendered graphics/complex technical illustrations.

Unprocessed scans of all key data generated through electrophoretic separation techniques need to be presented in a supplementary figure that should be labeled and numbered as the final supplementary figure, and should be mentioned in every relevant figure legend. This figure does not count towards the total number of figures and is the only figure that can be displayed over multiple pages, but should be provided as a single file, in PDF or TIFF format. Data in this figure can be displayed in a relatively informal style, but size markers and the figures panels corresponding to the presented data must be indicated.

The total number of Supplementary Figures (not including the “unprocessed scans” Supplementary Figure) should not exceed the number of main display items (figures and/or tables (see our Guide to Authors and March 2012 editorial <http://www.nature.com/ncb/authors/submit/index.html#supinfo>; <http://www.nature.com/ncb/journal/v14/n3/index.html#ed>). No restrictions apply to Supplementary Tables or Videos, but we advise authors to be selective in including supplemental data.

GUIDELINES FOR EXPERIMENTAL AND STATISTICAL REPORTING

REPORTING REQUIREMENTS – We ask authors to complete a Reporting Summary that collects information on experimental design and reagents. We hope this will aid in your evaluation of the paper. The Reporting Summary can be found here <https://www.nature.com/documents/nr-reporting-summary.pdf>) Please note that these forms are dynamic ‘smart pdfs’ and must therefore be downloaded and completed in Adobe Reader. We will then flatten them for ease of use. If you would like to reference the guidance text as you complete the template, please access these flattened versions at <http://www.nature.com/authors/policies/availability.html>.

We strongly recommend the presentation of source data for graphical and statistical analyses as a separate Supplementary Table, and request that source data for all independent repeats are provided when representative experiments of multiple independent repeats, or averages of two independent experiments are presented. This supplementary table should be in Excel format, with data for different figures provided as different sheets within a single Excel file. It should be labelled and numbered as one of the supplementary tables, titled “Statistics Source Data”, and mentioned in all relevant figure legends.

Author Rebuttal, first revision:

We thank reviewer #4 for commenting on former Rev #1's remarks. Below please find our response (in blue font) to the comments of Reviewer #4.

Remarks to the Author:

I report below the original comments of Rev 1 and my feedback to the authors' response to each point ("Response").

General comments

1. The authors should clearly demonstrate that the phenotype is a result of loss of Atf4 in fibroblasts and not tumor endothelial cells. Their observations in Col1a1-CreERT2 mice are encouraging, however it will be important to rule a potential ectopic activity of Cre in tumor vasculature, for example by RT-qPCR analysis of sorted tumor endothelial cells.

Rev#4 Response: This point has been convincingly addressed by measuring Atf4 mRNA in tumor endothelial cells (TECs) with or without floxed Atf4 alleles (no differences were found). The best positive control would be CAFs isolated from the same tumors as TECs, rather than lung fibroblasts (LFB). Yet, what matters most is that no differences were found in TECs.

We thank the reviewer for that response.

2. The authors show that one of the main effects of Atf4 deficiency is reduced tumor angiogenesis, and propose, based on their in vitro results, that this is a result of reduced production of pro-angiogenic factors, such as VEGF and Sdf1, by CAFs. There is a missing link with in vivo observations – do Atf4 deficient fibroblasts have reduced levels of Vegfa mRNA or they display reduced Vegfa secretion and deposition? Do the authors observe reduced production of VEGF&other factors in vivo? If the main idea is that CAF secretome is affected, it would be important to confirm the results in independent experiments using Limunex or Elisa of tumor lysates but also for in vitro data, right now it looks like n=1 data.

Rev#4 Response: This point has been experimentally addressed. The new data are not yet fully compelling, even though they are interesting and somewhat support the authors' conclusions. But definitive proof that perivascular CAFs secrete VEGFA and CXCL12 (correct name for SDF1) in an ATF4-dependent manner is still lacking. The authors are right that VEGFA and CXCL12 protein levels are not expected to drop in tumors with conditional Atf4 deletion, as many cells other than CAFs express those proteins in the TME. Yet, the data in Fig 5e-h (and S8) are not fully convincing, as there is no proof that the relative decrease of VEGFA and CXCL12 protein levels in the perivascular space is in fact due to cell-autonomous effects in the CAFs. The perivascular space is populated by pericytes, macrophages and other cells that also express functional VEGFA and CXCL12. I propose the following:

- Fig 5e-h should be improved and include CAF, pericyte and macrophage-specific markers, along with VEGFA and CXCL12, with better confocal resolution and relative quantification. Currently, it is unclear which cell types express VEGFA and CXCL12 in the peri-endothelial space.

Reviewer #4 brings up a valid point. Indeed, both angiogenic factors can be secreted by different cell types that reside in the perivascular area (i.e., CAF, pericytes, macrophages, etc.). However, we would like to point out that these secreted factors are immediately diffused in the proximal vicinity, so tracking their source by co-staining with cell-specific markers would be quite challenging and might not lead to conclusive results. Instead, we decided to analyze the levels of these angiogenic factors in B16F10 tumors grown in Col1a1Cre;Atf4^{wt/wt} and Col1a1Cre;Atf4^{Δ/Δ} mice (fibroblast-specific Atf4 excision) and calculated the % colocalization between VEGF and CXCL12 with CD31 positive staining. Intriguingly, the levels of both angiogenic factors were significantly lower in the perivascular areas from tumors grown in Col1a1Cre;Atf4^{Δ/Δ} compared to Col1a1Cre;Atf4^{wt/wt} littermates. Since Col1a1 is almost exclusively expressed in fibroblasts (and osteoclasts which are not present here), these data further indicate that the decrease in these cytokines is primarily due to defects in the vCAFs. However, some levels of VEGF and CXCL12 are still present in the tumors grown in FB-specific Atf4 knockouts and we cannot exclude some additional contribution from other sources. We have modified the results section to indicate this possibility. Representative images and quantification graphs have been added as new **Supplementary Fig. 8f,g** (for VEGF) and **Supplementary Fig. 8h,i** (for CXCL12). Also, SDF-1 has been corrected to CXCL12 in both manuscript and figures.

- The Luminex data, shown in the rebuttal, that indicate no difference in total VEGFA and CXCL12 levels in *Atf4* wt and DD tumors should be included in the ms and described.

The figure of the angiogenesis array analyzing tumor lysates has been added as new **Fig. 5g**. The manuscript and figure legends have also been updated.

3. Related to 1, Figure 5 has several issues:

- Endothelial cells seems to be in poor shape/dying (Fig 5b)
- no positive control is included
- use of Matrigel tube forming assay is a suboptimal way for modeling angiogenesis in vitro (PMID: 25931450), the authors should consider alternatives, such as endothelial spheroid sprouting assay.

Rev#4 Response: These points have been experimentally addressed. I agree with the Reviewer that Matrigel tube forming assay is not an adequate model to study angiogenesis in vitro. Yet, the authors' new data and response may be acceptable at this stage.

We thank the reviewer for this assessment.

4. The IHC analysis in Figure 3 shows loss of larger aSMA-covered tumor vessels, but not microvessels – such effect is opposite from what is expected upon loss of tumor angiogenesis signaling and indicates a more complex underlying mechanism than the one proposed by the authors. Analysis of tumor thick sections will be helpful to find out whether loss of *Atf4* rather leads to increased tumor hypersprouting coupled with reduced lumen formation as observed upon inhibition of Notch signaling in other tumor models, such as in *Dll4* +/- mice. This also leads to reduced tumor growth and perfusion. It may be also interesting to check whether there is loss of secretion/production of Notch ligands, such as *Dll4* in endothelial cells.

Rev#4 Response: The Reviewer argues that the effects of *Atf4* inactivation on the angiogenic tumor vasculature were not thoroughly characterized. I agree with this concern, which remains valid. While I agree with the authors that looking at Notch/DLL4 signaling may be beyond the scope of the study, it is felt that better characterization of the tumor vasculature should be provided, at least by addressing the question on sprouting and other vascular parameters using thick sections. The quality of CD31 staining in Fig S4 and other figures is often suboptimal and should be refined (eg, is co-localization with ki67 reliable?).

We agree with the need to probe in more detail the impact of ATF4 deficiency in vascular architecture and sprouting, to complement our *in vitro* data. To do this, we performed *ex vivo* tumor vasculature imaging using multiphoton and confocal microscopy on B16F10 tumors of equal volume excised from *Atf4*^{wt/wt} and *Atf4*^{Δ/Δ} mice. These experiments

confirmed our immunofluorescence data, again finding significantly fewer number of blood vessels in the B16F10 tumors grown in $Atf4^{\Delta/\Delta}$ mice. However, this approach also revealed significantly less sprouting/branching in the tumors grown in the $Atf4^{\Delta/\Delta}$ mice compared to the tumors grown in $Atf4^{wt/wt}$ littermates. This finding further supports our hypothesis of an abnormal vascularization phenotype following $Atf4$ excision. Representative images and videos and quantification data have been added as new **Fig. 5a,b** and **Supplementary Video 1-4**.

Concerning the co-staining of CD31 with Ki67 and TUNEL, we have increased the brightness of the CD31 staining in the originally captured images to better demonstrate the co-localization. See Supplementary Fig. 4f.

5. The authors should provide a more thorough analysis of the TME and tumors, including the analysis of mural coverage of vessels and proliferation and apoptosis of endothelial, CAFs and tumor cells.

Rev#4 Response: *The authors have added substantial new data. They should include all the data shown in the rebuttal letter in the ms (including data on TUNEL+ ECs/CAFs). Some reservations remain on the quality of vascular staining (see point 4 above), but overall this point has been satisfactorily addressed.*

The figures and quantification data from the TUNEL/CD31 and TUNEL/CAFs co-staining have been added as new **Supplementary Fig. 4h,i** and **Supplementary Fig. 5l,m**, respectively.

6. The authors suggest that ATF4 main target cell type are perivascular CAFs – how are they different from vSMCs and pericytes? In human tissues “ATF4 is highly expressed (apart from the tumor cells) on CAFs (aSMA) that localized on the perivascular area (CD34) compared to other CAFs reside away from the blood vessels (Fig. 3i).” –

however such perivascular ATF4 could as well be vSMCs, ATF4 expression and role in vSMCs has been reported.

Rev#4 Response: I agree with the Review and feel that the authors' response is not convincing. The new data shown in the rebuttal do not make a compelling case that vCAFs differ from mural cells/pericytes (are we splitting the hair with gene expression data?). In fact, vCAFs should be regarded as (a subset of) "mural cells", unless differential origins or phenotypes are clearly identified, which is not the case here. I recommend the data be included in the ms and any claim be toned down and critically reassessed.

The issue of the plasticity of CAFs and mural cells is an important one and we have spent considerable effort to critically analyze our tumor scRNA-seq data. We would like to point out that CAFs represent the cell type with the highest level of plasticity amongst the TME¹⁻³. When our results in Fig. 3g-j, Fig. 3l-n and Supplementary Fig. 5p,q are taken collectively, we believe that there is a strong case that the major defect in terms of collagen synthesis following *Atf4* excision amongst all the TME cells lies in cells with a high level of genes associated with perivascular CAFs. We would like to draw the attention of Rev. #4 to Supplementary Fig. 3f, where it can be seen that the strength of the vCAFs gene signature⁴ in this cell cluster, is at least twice as high as the established signature of mural cells⁵. However, we do recognize that in the larger size tumors, this difference in the levels of the gene signature is diminished. This suggests that there is a trend towards a more "mural phenotype" for these vCAFs. Thus, we have amended the text in the results to underscore this underlying plasticity in cell phenotype and the possibility that at least in large tumors, the cell population we've identified has features of both CAFs and mural cells. See **Supplementary Fig. 3f and Supplementary Table 7** (please note that even though *Acta2/αSMA* is part of the list, we removed it from the mural cell signature because it is not unique to mural cells in the literature). Also, we have addressed this point extensively in the manuscript.

7. The authors propose that unlike wild type fibroblasts, *Atf4* deficient fibroblasts do not support growth of B16F10 cells. It would be important to provide evidence that *Atf4* deficient fibroblast survive upon such co-injection, it is possible that they are simply do not survive s.c. transplantation.

Rev#4 Response: Response and new data are compelling.
We thank the reviewer for that response.

8. The authors should discuss in more detail potential mechanisms underlying a tumor escape from *Atf4* inhibition.

Rev#4 Response: Response and discussion are adequate.
We thank the reviewer for that response.

Technical comments:

1. The authors state that they use orthotopic B16F10 model, however this is not the case as they do s.c. and not intradermal injections.
2. The authors mention that they identified a new population of melCAFs, with mixed melanoma and CAF identities - unusual "mixed" populations can be observed in scRNAseq data due to RNA transfer/uptake during sample preparation, especially in physically interacting cells. Please confirm out that this is not a sample preparation artifact by other means , e.g, costaining, or amend the analysis.
3. The authors sometimes use very small number of tumors for the analysis, please consider increasing the number of samples to make sure that the reported observation are indeed solid.

Rev#4 Response: Response to the 3 technical comments is adequate. All data shown in the rebuttal should be included in the ms and addressed with text revisions (point 2).

We have included the violin plots from both small and large sized B16F10 tumors showing that melCAFs are a distinct population, comprising a distinct cell type, that presumably emerged due to the extensive cell plasticity that takes place in the tumor microenvironment. The new figures are **Supplementary Fig. 2e** (small tumors) and **Supplementary Fig. 3b** (large tumors).

Minor comments

1. CD31 is a pan-endothelial marker and not specific for tumor endothelium
2. Please mention the percentage of CAFs found in large B16F10 tumors like in small tumors.
3. Regarding vessel permeability experiment, please comment on whether this is due to intrinsic (check endothelial gene signature from single cell RNA-seq) or extrinsic effect on endothelial cells.
4. supp Figure 4d, the merged figure look fuzzy -please replace
5. Supp 6d-e it would be easier to keep the naming of tumor and not use FB.
6. Fig 4b should be in supplemental data because it is literature and not a result of the paper.
7. Fig 4i please mention what the dots represent
8. Fig 5e mention the color code and the proteins in the legend

Rev#4 Response: Response to all minor comments is adequate.

We thank the reviewer for that response.

Additional comments

1. We have corrected the "SDF-1" to "CXCL12" based on the reviewer's suggestion.
2. Figure legends have been updated.
3. Methods and Supplementary tables have been updated.
4. Mr. Duo Zhang has been included as an author.
5. We have updated the References.

References

1. Sahai, E. *et al.* A framework for advancing our understanding of cancer-associated fibroblasts. *Nat Rev Cancer* **20**, 174-186 (2020).
2. Chen, X. & Song, E. Turning foes to friends: targeting cancer-associated fibroblasts. *Nat Rev Drug Discov* **18**, 99-115 (2019).
3. Yoshida, G.J. Regulation of heterogeneous cancer-associated fibroblasts: the molecular pathology of activated signaling pathways. *J Exp Clin Cancer Res* **39**, 112 (2020).
4. Bartoschek, M. *et al.* Spatially and functionally distinct subclasses of breast cancer-associated fibroblasts revealed by single cell RNA sequencing. *Nature communications* **9**, 5150 (2018).
5. Muhl, L. *et al.* Single-cell analysis uncovers fibroblast heterogeneity and criteria for fibroblast and mural cell identification and discrimination. *Nature communications* **11**, 3953 (2020).

Decision Letter, second revision:

Subject: Your manuscript, NCB-K45380B

Message:

Our ref: NCB-K45380B

23rd February 2022

Dear Dr. Koumenis,

Thank you for submitting your revised manuscript "A Stromal Integrated Stress Response Activates Perivascular Cancer-Associated Fibroblasts to Drive Angiogenesis and Tumor Progression" (NCB-K45380B). It has now been seen by the original referees and their comments are below. The reviewers find that the paper has improved in revision, and therefore we'll be happy in principle to publish it in

Nature Cell Biology, pending minor revisions to satisfy the referees' final requests and to comply with our editorial and formatting guidelines.

Thank you again for your interest in Nature Cell Biology Please do not hesitate to contact me if you have any questions.

Sincerely,

Zhe Wang, PhD
Senior Editor
Nature Cell Biology

Tel: +44 (0) 207 843 4924
email: zhe.wang@nature.com

Reviewer #4 (Remarks to the Author):

The authors have addressed the remaining questions, also by including compelling new data.

Minor revision:

- The background color in the the two large panels of Figure 8 may be change from salmon to white.

Decision letter, final requests:

Subject: NCB: Your manuscript, NCB-K45380B

Message:

Our ref: NCB-K45380B

17th March 2022

Dear Dr. Koumenis,

Thank you for your patience as we've prepared the guidelines for final submission of your Nature Cell Biology manuscript, "A Stromal Integrated Stress Response Activates Perivascular Cancer-Associated Fibroblasts to Drive Angiogenesis and Tumor Progression" (NCB-K45380B). Please carefully follow the step-by-step instructions provided in the attached file, and add a response in each row of the table to indicate the changes that you have made. Ensuring that each point is addressed will help to ensure that your revised manuscript can be swiftly handed over to our production team.

We would like to start working on your revised paper, with all of the requested files and forms, as soon as possible (preferably within one week). Please get in contact with us if you anticipate delays.

In recognition of the time and expertise our reviewers provide to Nature Cell Biology's editorial process, we would like to formally acknowledge their contribution to the external peer review of your manuscript entitled "A Stromal Integrated Stress Response Activates Perivascular Cancer-Associated Fibroblasts to Drive Angiogenesis and Tumor Progression". For those reviewers who give their assent, we will be publishing their names alongside the published article.

Nature Cell Biology offers a Transparent Peer Review option for new original research manuscripts submitted after December 1st, 2019. As part of this initiative, we encourage our authors to support increased transparency into the peer review process by agreeing to have the reviewer comments, author rebuttal letters, and editorial decision letters published as a Supplementary item. When you submit your final files please clearly state in your cover letter whether or not you would like to participate in this initiative. Please note that failure to state your preference will result in delays in accepting your manuscript for publication.

Cover suggestions

As you prepare your final files we encourage you to consider whether you have any images or illustrations that may be appropriate for use on the cover of Nature Cell Biology.

Nature Cell Biology has now transitioned to a unified Rights Collection system which will allow our Author Services team to quickly and easily collect the rights and permissions required to publish your work. Approximately 10 days after your paper is formally accepted, you will receive an email in providing you with a link to complete the grant of rights. If your paper is eligible for Open Access, our Author Services team will also be in touch regarding any additional information that may be required to arrange payment for your article.

Please note that Nature Cell Biology is a Transformative Journal (TJ). Authors may publish their research with us through the traditional subscription access route or make their paper immediately open access through payment of an article-processing charge (APC). Authors will not be required to make a final decision about access to their article until it has been accepted. Find out more about Transformative Journals

Authors may need to take specific actions to achieve compliance with funder and institutional open access mandates. If your research is supported by a funder that requires immediate open access (e.g. according to Plan S principles) then you should select the gold OA route, and we will direct you to the compliant route where possible. For authors selecting the subscription publication route, the journal's standard licensing terms will need to be accepted, including self-archiving policies. Those licensing terms will supersede any other terms that the author or any third party may assert apply to any version of the manuscript.

For information regarding our different publishing models please see our Transformative Journals page. If you have any questions about costs, Open Access requirements, or our legal forms, please contact ASJournals@springernature.com.

[REDACTED]

If you have any further questions, please feel free to contact us. Thank you!

Best regards,

Ziqian Li
Editorial Assistant
Nature Cell Biology

On behalf of

Zhe Wang, PhD
Senior Editor
Nature Cell Biology

Tel: +44 (0) 207 843 4924
email: zhe.wang@nature.com

Reviewer #4:

Remarks to the Author:

The authors have addressed the remaining questions, also by including compelling new data.

Minor revision:

- The background color in the the two large panels of Figure 8 may be change from salmon to white.

Author Rebuttal, second revision:

We thank reviewer #4 for the quick turnaround of this revised version. Below please find our response (in blue font) to the comment of Reviewer #4.

Remarks to the Author:

The authors have addressed the remaining questions, also by including compelling new data.

Minor revision:

- The background color in the two large panels of Figure 8 may be change from salmon to white.

We thank the reviewer for this suggestion. The illustration in Figure 8 has been moved to Extended Data as Figure 10, based on editor's suggestion. Moreover, we feel that will be more appropriate to keep the salmon color as background in both large panels as we use this color to denote the intracellular space.

Final Decision Letter:

Subject: Decision on Nature Cell Biology submission NCB-K45380C

Message:

Dear Dr Koumenis,

I am pleased to inform you that your manuscript, "A Stromal Integrated Stress Response Activates Perivascular Cancer-Associated Fibroblasts to Drive Angiogenesis and Tumor Progression", has now been accepted for publication in Nature Cell Biology.

Please note that Nature Cell Biology is a Transformative Journal (TJ). Authors may publish their research with us through the traditional subscription access route or make their paper immediately open access through payment of an article-processing charge (APC). Authors will not be required to make a final decision about access to their article until it has been accepted. Find out more about Transformative Journals

Authors may need to take specific actions to achieve compliance with funder and institutional open access mandates. If your research is supported by a funder that requires immediate open access (e.g. according to Plan S principles) then you should select the gold OA route, and we will direct you to the compliant route where possible. For authors selecting the subscription publication route, the journal's standard licensing terms will need to be accepted, including self-archiving policies. Those licensing terms will supersede any other terms that the author or any third party may assert apply to any version of the manuscript.

To assist our authors in disseminating their research to the broader community, our SharedIt initiative provides you with a unique shareable link that will allow anyone (with or without a subscription) to read

the published article. Recipients of the link with a subscription will also be able to download and print the PDF.

If you have not already done so, we strongly recommend that you upload the step-by-step protocols used in this manuscript to the Protocol Exchange (www.nature.com/protocolexchange), an open online resource established by Nature Protocols that allows researchers to share their detailed experimental know-how. All uploaded protocols are made freely available, assigned DOIs for ease of citation and are fully searchable through nature.com. Protocols and Nature Portfolio journal papers in which they are used can be linked to one another, and this link is clearly and prominently visible in the online versions of both papers. Authors who performed the specific experiments can act as primary authors for the Protocol as they will be best placed to share the methodology details, but the Corresponding Author of the present research paper should be included as one of the authors. By uploading your Protocols to Protocol Exchange, you are enabling researchers to more readily reproduce or adapt the methodology you use, as well as increasing the visibility of your protocols and papers. You can also establish a dedicated page to collect your lab Protocols. Further information can be found at www.nature.com/protocolexchange/about

With kind regards,

Zhe Wang, PhD
Senior Editor
Nature Cell Biology

Tel: +44 (0) 207 843 4924
email: zhe.wang@nature.com